# Revisiting Zeroth-Order Hessian Approximation: A Single-Step Policy Optimization Lens

**Junbin Qiu** [1]  **Zhaowei Hong** [1]  **Renzhe Xu** [2]  **Yao Shu** [1]

## Abstract

Accurate Zeroth-Order (ZO) Hessian estimation is a cornerstone of derivative-free methods, essential for tasks such as bilevel optimization, Bayesian inference, and uncertainty quantification. However, obtaining a complete suite of low-variance estimators for the Hessian and its inverse in high-dimensional settings remains a significant challenge. To address this, we propose a unified framework that reinterprets ZO Hessian approximation through the lens of single-step Policy Optimization (PO). This perspective establishes a theoretical equivalence between general ZO Hessian estimators and the Hessian of a smoothed PO objective, unifying distinct classical randomized estimators as specific instances of baseline selection. Building on this foundation, we introduce ZoVH, a comprehensive suite of variance-reduced estimators for the full Hessian matrix, its regularized inverse, and the bias-corrected inverse Hessian-gradient product. ZoVH leverages two key techniques: *(1)* a unique optimal baseline derived to provably minimize variance, and *(2)* a query reuse strategy that incorporates historical function queries to enhance sample efficiency without inflating costs. Our rigorous theoretical analysis confirms the unbiasedness of the Hessian estimator, validates the variance optimality of our baseline, provides error bounds for the entire ZoVH suite, and establishes convergence guarantees for the resulting curvature-aware ZO algorithm. Extensive empirical results validate our theoretical findings, demonstrating that ZoVH achieves superior estimation accuracy and convergence performance in real-world applications. Code is available at https://github.com/Qjbtiger/ZoVH.

[1]Hong Kong University of Science and Technology (Guangzhou) [2]Shanghai University of Finance and Economics. Correspondence to: Yao Shu <yaoshu@hkust-gz.edu.cn>.

*Proceedings of the 43 rd International Conference on Machine Learning*, Seoul, South Korea. PMLR 306, 2026. Copyright 2026 by the author(s).

## 1. Introduction

Zeroth-Order (ZO) Hessian approximation for black-box objectives $F(\boldsymbol{\theta}) \triangleq \mathbb{E}_{\xi}[f(\boldsymbol{\theta}; \xi)]$, which are accessible solely through noisy function evaluations, represents a fundamental challenge in derivative-free optimization (Coope & Tappenden, 2020; Balasubramanian & Ghadimi, 2022; Roy et al., 2022; Feng & Wang, 2023). Estimating accurate curvature information is crucial for various applications, including derivative-free bilevel optimization (Aghasi & Ghadimi, 2025; Yang et al., 2026), uncertainty quantification (Kalmikov & Heimbach, 2014), and curvature-aware fine-tuning of Large Language Models (LLMs) (Zhao et al., 2025). While naive finite-difference methods typically necessitate $\mathcal{O}(d^2)$ function queries (Fabian, 1971), randomized rank-one estimators can reduce the query complexity to a constant number of evaluations, independent of the dimension $d$ (e.g., 4 queries for 2SPSA (Spall, 2000) and 3 for RDSA (L. A. et al., 2017)). We provide a discussion of related work about ZO Hessian approximation in Appx. A.

Despite its practical significance, ZO Hessian approximation suffers from high variance, particularly in noisy, high-dimensional regimes (Zhu, 2022; Feng & Wang, 2023). While a recent study by (Qiu et al., 2025) bridged ZO gradient estimation with single-step Policy Optimization (PO) methods to establish a rigorous theoretical foundation and variance reduction techniques, such connections remain unexplored for ZO Hessian approximation. Moreover, prior studies have primarily focused on estimating the Hessian $\nabla^2 F(\boldsymbol{\theta})$ itself. In contrast, the estimation of regularized inverse Hessian $(\nabla^2 F(\boldsymbol{\theta}) + \lambda \mathbf{I}_d)^{-1}$ and its product with the gradient $(\nabla^2 F(\boldsymbol{\theta}) + \lambda \mathbf{I}_d)^{-1} \nabla F(\boldsymbol{\theta})$ are essential for curvature-aware zeroth-order optimization (Zhu, 2022; Zhao et al., 2025), but have received limited attention.

Our **first** contribution is to demonstrate a unified framework for ZO Hessian approximation from the PO perspective. Within this framework, ZO Hessian approximation is reinterpreted as the Hessian of a smoothed objective $\nabla^2 F_\mu(\boldsymbol{\theta})$ under a parameterized sampling policy (Sec. 3.1). Specifically, for Gaussian sampling, this perspective leads to explicit Hessian identities (Props. 3.2 and 3.3) incorporating baseline terms. By selecting different baselines, these identities recover classical Hessian estimators from (Balasub-

ramanian & Ghadimi, 2022) as special cases (Sec. 3.2). Furthermore, this framework extends naturally to general sampling distributions via importance sampling (Sec. 3.3).

Building on this foundation, our **second** contribution is ZoVH, a novel *Zeroth-Order Variance-reduced Hessian approximation* that minimizes estimator variance through optimal baseline selection and query reuse (Sec. 4). ZoVH constructs a Hessian estimator leveraging two techniques: *(a)* an averaged baseline that provably minimizes variance among all constant baselines (Thm. 4.7); and *(b)* a query reuse strategy that estimates the Hessian from a history buffer of past function queries, increasing the sample size without incurring additional query costs, thereby further reducing variance (Sec. 4.1). To assess the efficacy of ZoVH, we provide a comprehensive theoretical analysis in Sec. 4.2, covering its unbiasedness (Thm. 4.6) and bias-variance trade-off (Thm. 4.8). We then present an empirical error analysis in Sec. 6.1, demonstrating that ZoVH achieves significantly lower estimation error compared to classical Hessian estimators across various synthetic functions and neural networks.

Our **third** contribution lies in the development of a practical framework that extends ZoVH to an efficient curvature-aware Zeroth-Order Optimization (ZOO) algorithm. Specifically, we develop a stable ZO inverse Hessian approximation by incorporating ridge regularization into the ZoVH estimator (Sec. 5.1), and a bias-corrected inverse Hessian-gradient product that mitigates the bias introduced by reusing function queries for both Hessian and gradient estimation (Sec. 5.2). These approximations form the basis of a curvature-aware ZOO algorithm (Alg. 1) that efficiently computes second-order updates using minimal function queries. Theoretically, we prove the convergence guarantee of this approach (Thm. 5.5), and empirically, we validate its effectiveness on three representative applications (Sec. 6.2), showing substantial improvements over existing ZO methods.

## 2. Preliminaries

**Problem Setup.** We study ZO approximation of the Hessian $\nabla^2 F(\boldsymbol{\theta}) \in \mathbb{R}^{d \times d}$ for the stochastic objective:

$$F(\boldsymbol{\theta}) \triangleq \mathbb{E}_\xi[f(\boldsymbol{\theta}; \xi)], \tag{1}$$

where $F$ is accessible only through noisy function evaluations of $f(\boldsymbol{\theta}; \xi)$, and $\xi$ represents the inherent stochasticity in the function evaluation. Our goal is to construct a Hessian estimator $\widehat{\mathbf{H}}(\boldsymbol{\theta})$ using a small number of function queries, thereby avoiding the prohibitive $\mathcal{O}(d^2)$ complexity of full finite-difference schemes (Fabian, 1971), while maintaining low estimation variance.

**Classical ZO Hessian Estimators.** We adopt a unified perspective in which Hessian information is recovered from finite differences of noisy function evaluations along random probing directions. Several classical ZO Hessian estimators (Balasubramanian & Ghadimi, 2022; Zhu, 2022) can be expressed using a common template. Let $\{\mathbf{u}_k\}_{k=1}^K \overset{\text{i.i.d.}}{\sim} \mathcal{N}(\mathbf{0}, \mathbf{I}_d)$ be standard Gaussian probing directions and $\mu > 0$ be a smoothing parameter. The One-Point Stein estimator combines a rank-one term with an identity correction derived from Stein's identity (Stein, 1972):

$$\widehat{\mathbf{H}}_{S_1} \triangleq \frac{1}{K} \sum_{k=1}^K \frac{f(\boldsymbol{\theta} + \mu\mathbf{u}_k; \xi)}{\mu^2} \left(\mathbf{u}_k\mathbf{u}_k^\top - \mathbf{I}_d\right). \tag{2}$$

Following the same principle, the Two-Point Stein estimator replaces the one-point measurement with a one-sided finite difference:

$$\widehat{\mathbf{H}}_{S_2} \triangleq \frac{1}{K} \sum_{k=1}^K \frac{f(\boldsymbol{\theta} + \mu\mathbf{u}_k; \xi) - f(\boldsymbol{\theta}; \xi)}{\mu^2} \left(\mathbf{u}_k\mathbf{u}_k^\top - \mathbf{I}_d\right), \tag{3}$$

and the Three-Point Stein estimator further adopts a symmetric central difference, which typically improves numerical stability:

$$\widehat{\mathbf{H}}_{S_3} \triangleq \frac{1}{2K} \sum_{k=1}^K \frac{f(\boldsymbol{\theta}_k^+; \xi) - 2f(\boldsymbol{\theta}; \xi) + f(\boldsymbol{\theta}_k^-; \xi)}{\mu^2} \\ \times \left(\mathbf{u}_k\mathbf{u}_k^\top - \mathbf{I}_d\right). \tag{4}$$

where $\boldsymbol{\theta}_k^\pm \triangleq \boldsymbol{\theta} \pm \mu\mathbf{u}_k$.

In contrast to the Stein-type estimators, (Balasubramanian & Ghadimi, 2022; Zhao et al., 2025) consider a randomized Central-Difference (CD) estimator that employs a rank-one weighted average without the identity correction:

$$\widehat{\mathbf{H}}_{CD}(\boldsymbol{\theta}) \triangleq \frac{1}{2K} \sum_{k=1}^K \frac{f(\boldsymbol{\theta}_k^+; \xi) - 2f(\boldsymbol{\theta}; \xi) + f(\boldsymbol{\theta}_k^-; \xi)}{\mu^2} \mathbf{u}_k\mathbf{u}_k^\top. \tag{5}$$

In Sec. 3, we reinterpret these estimators through the lens of PO, demonstrating that they emerge as special cases of our generalized Hessian estimation framework.

**ZO Gradient Estimator.** In practical ZOO algorithms, Hessian approximation is typically paired with a ZO gradient estimator to enable second-order updates (Zhu, 2022; Zhao et al., 2025). We adopt the Gaussian-smoothed gradient estimator from (Qiu et al., 2025), which approximates $\nabla F(\boldsymbol{\theta})$ via finite differences along Gaussian probing directions $\{\mathbf{u}_k\}_{k=1}^K \overset{\text{i.i.d.}}{\sim} \mathcal{N}(\mathbf{0}, \mathbf{I}_d)$:

$$\hat{\nabla} F(\boldsymbol{\theta}) \triangleq \frac{1}{K-1} \sum_{k=1}^K \frac{f(\boldsymbol{\theta} + \mu\mathbf{u}_k; \xi) - b}{\mu} \mathbf{u}_k, \tag{6}$$

where the averaged baseline $b \triangleq \frac{1}{K} \sum_{k=1}^{K} f(\boldsymbol{\theta} + \mu \mathbf{u}_k; \xi)$ is used to reduce the estimator variance.

**Notations.** Throughout this paper, we denote $\mathbb{E}_{\mathbf{u}}[\cdot] \triangleq \mathbb{E}_{\mathbf{u} \sim \mathcal{N}(0, \mathbf{I}_d)}[\cdot]$, and $\mathbb{E}[\cdot] \triangleq \mathbb{E}_{\xi} \mathbb{E}_{\mathbf{u}}[\cdot]$ as the full expectation over both the stochasticity $\xi$ and the standard Gaussian distribution. Furthermore, for a matrix $\mathbf{A}$, we use $\|\mathbf{A}\|$ and $\|\mathbf{A}\|_F$ to denote its spectral norm and Frobenius norm.

## 3. Rethinking ZO Hessian Approximation

In this section, we reframe Zeroth-Order (ZO) Hessian approximation through the lens of Policy Optimization (PO). By interpreting the smoothed objective as an expectation over a sampling policy, we derive a unified Hessian estimator that generalizes classical finite-difference methods (Sec. 3.1). This perspective offers a rigorous derivation of existing estimators (Sec. 3.2), extends naturally to importance sampling (Sec. 3.3), and highlights a potential pathway for developing improved approximation schemes.

### 3.1. The Single-Step Policy Optimization Perspective

Following (Qiu et al., 2025), we adopt the smoothed objective function $F_\mu(\boldsymbol{\theta})$ defined over a smoothing policy $\pi_{\boldsymbol{\theta}}(\mathbf{x})$. We reformulate this objective within a single-step policy optimization framework by treating $\pi_{\boldsymbol{\theta}}(\mathbf{x})$ as a parameterized policy (where $\mathbf{x} = \boldsymbol{\theta} + \mu \mathbf{u}$):

$$F_\mu(\boldsymbol{\theta}) = \mathbb{E}_{\mathbf{x} \sim \pi_{\boldsymbol{\theta}}(\mathbf{x})}[\mathbb{E}_{\xi}[f(\mathbf{x}; \xi)]] . \tag{7}$$

The Hessian of the PO objective $F_\mu(\boldsymbol{\theta})$ is derived by differentiating the policy gradient:

**Lemma 3.1** (Hessian of PO objective (7)). *The Hessian of the smoothed objective $F_\mu$ in (7) is given by:*

$$\nabla^2 F_\mu(\boldsymbol{\theta}) = \mathbb{E}_{\mathbf{x} \sim \pi_{\boldsymbol{\theta}}(\mathbf{x})}[\mathbf{H}_{\pi_{\boldsymbol{\theta}}} \cdot \mathbb{E}_{\xi}[f(\mathbf{x}; \xi)]] ,$$

*where $\mathbf{H}_{\pi_{\boldsymbol{\theta}}} \triangleq (\nabla \ln \pi_{\boldsymbol{\theta}}(\mathbf{x}))(\nabla \ln \pi_{\boldsymbol{\theta}}(\mathbf{x}))^\top + \nabla^2 \ln \pi_{\boldsymbol{\theta}}(\mathbf{x})$ denotes the Hessian of the policy $\pi_{\boldsymbol{\theta}}$.*

**Remark.** The proof is provided in Appx. C.2. Lem. 3.1 yields a Hessian estimator for the PO objective (7) applicable to any differentiable smoothing policy $\pi_{\boldsymbol{\theta}}$. This generality facilitates the derivation of Hessian estimators for diverse smoothing distributions, provided their explicit forms are known.

We now consider the canonical case of Gaussian policy, which yields a general Gaussian smoothed Hessian estimators:

**Proposition 3.2** (Gaussian Smoothed Hessian Estimators). *Let the smoothing policy be Gaussian, i.e., $\pi_{\boldsymbol{\theta}}(\mathbf{x}) = \mathcal{N}(\boldsymbol{\theta}, \mu^2 \mathbf{I}_d)$. The Hessian of the smoothed objective $F_\mu$*

*in (7) reduces to:*

$$\nabla^2 F_\mu(\boldsymbol{\theta}) = \mathbb{E}_{\mathbf{u}}\left[\frac{\mathbb{E}_{\xi}[f(\boldsymbol{\theta} + \mu \mathbf{u}; \xi)] - b}{\mu^2}\left(\mathbf{u}\mathbf{u}^\top - \mathbf{I}_d\right)\right] ,$$

*where the constant baseline $b$ is an arbitrary term independent of the standard Gaussian random direction $\mathbf{u} \sim \mathcal{N}(\mathbf{0}, \mathbf{I}_d)$.*

**Remark.** The proof is provided in Appx. C.3. Prop. 3.2 is derived by substituting the Gaussian policy $\pi_{\boldsymbol{\theta}}(\mathbf{x}) = \mathcal{N}(\boldsymbol{\theta}, \mu^2 \mathbf{I}_d)$ into Lem. 3.1. This derivation does not require explicit invocation of Stein's identity, as the identity matrix $\mathbf{I}_d$ naturally arises from the term $\nabla^2 \ln \pi_{\boldsymbol{\theta}}(\mathbf{x}) = -\frac{1}{\mu^2}\mathbf{I}_d$. Furthermore, since $\mathbb{E}_{\mathbf{u}}[\mathbf{u}\mathbf{u}^\top - \mathbf{I}_d] = 0$, the choice of baseline $b$ is flexible and does not introduce bias. In Sec. 3.2, we demonstrate that specific choices of $b$ recover various classical Stein-type Hessian estimators introduced in Sec. 2.

Since $\mathbb{E}_{\mathbf{u}}[\mathbf{u}\mathbf{u}^\top - \mathbf{I}_d] = 0$, the identity matrix in Prop. 3.2 can be effectively eliminated by absorbing it into the expectation via baseline subtraction. This observation leads to the following statistically equivalent Hessian estimator:

**Proposition 3.3** (Rank-One Hessian Estimator). *Let the smoothing policy be Gaussian, i.e., $\pi_{\boldsymbol{\theta}}(\mathbf{x}) = \mathcal{N}(\boldsymbol{\theta}, \mu^2 \mathbf{I}_d)$. The Hessian of the smoothed objective $F_\mu$ in (7) reduces to:*

$$\nabla^2 F_\mu(\boldsymbol{\theta}) = \mathbb{E}_{\mathbf{u}}\left[\frac{\mathbb{E}_{\xi}[f(\boldsymbol{\theta} + \mu \mathbf{u}; \xi)] - b}{\mu^2}\mathbf{u}\mathbf{u}^\top\right]$$

*with the averaged baseline:*

$$b \triangleq \mathbb{E}_{\mathbf{u}}[\mathbb{E}_{\xi}[f(\boldsymbol{\theta} + \mu \mathbf{u}; \xi)]] , \tag{8}$$

*where $\mathbf{u} \triangleq (\mathbf{x} - \boldsymbol{\theta})/\mu$ is a standard Gaussian random direction.*

**Remark.** The proof is provided in Appx. C.4. Unlike the baseline in the gradient estimator (6), which is often manually introduced for gradient variance reduction, the averaged baseline in Prop. 3.3 emerges naturally from the Hessian derivation. Specifically, the identity term $\mathbf{I}_d$ is absorbed into the expectation using the identity $\mathbb{E}[\mathbf{u}\mathbf{u}^\top] = \mathbf{I}_d$, yielding an estimator structured as an average of rank-one matrices. In contrast to the flexibility of the baseline choice in Prop. 3.2, the averaged baseline in Prop. 3.3 is the unique choice that preserves unbiasedness (see Lem. 3.5).

### 3.2. Recovery of Classical ZO Hessian Estimators

The choice of the baseline $b$ in Prop. 3.2 plays a pivotal role in determining the structure of the Hessian estimator. By selecting distinct baselines such as the zero baseline ($b = 0$) or the anchor baseline ($b = f(\boldsymbol{\theta}; \xi)$), we can derive the classical Hessian estimators described in Sec. 2:

**Corollary 3.4** (Recovery of Stein-Type Estimators). *The general estimator in Prop. 3.2 recovers the following classical unbiased ZO Hessian estimators as special cases:*

*(I) **One-Point Stein Estimator:** Setting zero baseline $b = 0$ yields the estimator defined in (2).*

*(II) **Two-Point Stein Estimator:** Setting anchor baseline $b = f(\boldsymbol{\theta}; \xi)$ yields the estimator defined in (3).*

*(III) **Three-Point Stein Estimator:** Applying antithetic sampling $\{\mathbf{u}, -\mathbf{u}\}$ with the anchor baseline $b = f(\boldsymbol{\theta}; \xi)$ yields the estimator defined in (4).*

**Remark.** Since Prop. 3.2 holds for any baseline $b$ independent of $\mathbf{u}$, all Stein-type Hessian estimators recovered above are unbiased estimators of $\nabla^2 F_\mu(\boldsymbol{\theta})$. While the averaged baseline (8) can also be applied within the framework of Prop. 3.2, the resulting estimator is mathematically equivalent to the formulation in Def. 4.1 (derived from Prop. 3.3). We therefore omit a separate derivation here for brevity (see Appx. D.1 for further discussion).

In contrast to the flexibility of Prop. 3.2, the estimator in Prop. 3.3 imposes stricter constraints on the baseline to maintain validity. Specifically, to preserve unbiasedness, the averaged baseline (8) is the unique admissible choice:

**Lemma 3.5** (Uniqueness of Averaged Baseline). *For the Hessian estimator in Prop. 3.3, the averaged baseline in (8) is the unique choice that preserves unbiasedness, i.e., $\mathbb{E}[\widehat{\mathbf{H}}(\boldsymbol{\theta})] = \nabla^2 F_\mu(\boldsymbol{\theta})$.*

**Remark.** The proof is provided in Appx. C.5. This result highlights the critical role of the averaged baseline in ensuring the unbiasedness of the estimator in Prop. 3.3. Furthermore, in Thm. 4.7, we show that the averaged baseline is also optimal for variance minimization.

It is natural to inquire whether antithetic sampling can be applied to Prop. 3.3 to construct an unbiased central-difference estimator using the averaged baseline. We find that this is not feasible. Consider one finite sample of the averaged baseline using an antithetic pair $\{\mathbf{u}, -\mathbf{u}\}$, given by $b = \frac{1}{2}(f(\boldsymbol{\theta}^+; \xi) + f(\boldsymbol{\theta}^-; \xi))$. Substituting this into the estimator yields a zero coefficient, $f(\boldsymbol{\theta}^+; \xi) + f(\boldsymbol{\theta}^-; \xi) - 2b = 0$, causing the estimator to degenerate.

However, if we relax the requirement for unbiasedness with respect to $\nabla^2 F_\mu(\boldsymbol{\theta})$ and instead employ the anchor baseline $b = f(\boldsymbol{\theta}; \xi)$ with antithetic sampling, we recover the classical central-difference estimator:

**Corollary 3.6** (Recovery of Randomized Central-Difference Estimator (5)). *By applying antithetic sampling $\{\mathbf{u}, -\mathbf{u}\}$ with the anchor baseline $b = f(\boldsymbol{\theta}; \xi)$, Prop. 3.3 reduces to the classical central difference Hessian estimator (5).*

### 3.3. Generalized Form Through Importance Sampling

While Gaussian smoothing offers analytical tractability, optimal performance in ZOO often relies on the geometry of the smoothing distribution, such as uniform distributions on

the sphere for gradient bounding. To extend our framework to arbitrary sampling distributions, we leverage Importance Sampling (IS). This approach facilitates the estimation of the Hessian for a target smoothed objective $F_\mu$ defined by a target distribution $\pi_{\boldsymbol{\theta}}$, while drawing samples from a distinct proposal distribution $\rho$.

**Proposition 3.7** (General Hessian Estimator via Importance Sampling). *Let $\pi_{\boldsymbol{\theta}} = \mathcal{N}(\boldsymbol{\theta}, \mu^2 \mathbf{I}_d)$ be the target smoothing distribution, $\rho$ be a fixed sampling distribution, and $b$ be the averaged baseline in (8). Assume $supp(\pi_{\boldsymbol{\theta}}) \subseteq supp(\rho)$. The Hessian of the smoothed objective $F_\mu(\boldsymbol{\theta})$ in (7) can be evaluated by sampling $\mathbf{u} \sim \rho$ via:*

$$\nabla^2 F_\mu(\boldsymbol{\theta}) = \mathbb{E}_{\mathbf{u} \sim \rho} \left[ \frac{\mathbb{E}_\xi[f(\boldsymbol{\theta} + \mu\mathbf{u}; \xi)] - b}{\mu^2} \mathbf{u}\mathbf{u}^\top \cdot \mathcal{W}_{\pi/\rho}(\mathbf{u}) \right] ,$$

*with the averaged baseline via IS:*

$$b \triangleq \mathbb{E}_{\mathbf{u} \sim \rho} \left[ \mathbb{E}_\xi \left[ f(\boldsymbol{\theta} + \mu\mathbf{u}; \xi) \right] \cdot \mathcal{W}_{\pi/\rho}(\mathbf{u}) \right] ,$$

*where the IS ratio $\mathcal{W}_{\pi/\rho}(\mathbf{u}) \triangleq \frac{\pi_{\boldsymbol{\theta}}(\boldsymbol{\theta} + \mu\mathbf{u})}{\rho(\mathbf{u})}$.*

**Remark.** The proof is in Appx. C.6. This result generalizes Prop. 3.3 to arbitrary proposal distributions $\rho$. By incorporating the IS ratio $\mathcal{W}_{\pi/\rho}(\mathbf{u})$, we can correct for the discrepancy between the target smoothing distribution $\pi_{\boldsymbol{\theta}}$ and the sampling distribution $\rho$. This flexibility allows us to select proposal distributions that align better with the problem geometry, potentially improving estimator efficiency and reducing variance.

## 4. Variance-Reduced Hessian Approximation

Building on the foundation in Sec. 3, we introduce ZoVH, a practical framework for variance-reduced ZO Hessian approximation. This method integrates the *Averaged Baseline* (8) with a *Query Reuse* mechanism to enhance estimation efficiency (Sec. 4.1). We then provide a theoretical analysis of its bias and variance properties in Sec. 4.2.

### 4.1. Hessian Approximation

While Prop. 3.3 characterizes the Hessian as an expectation over Gaussian perturbations, a practical implementation necessitates a finite-sample Monte Carlo approximation. Moreover, we incorporate the averaged baseline and query reuse techniques to reduce variance, as detailed below.

**Averaged Baseline.** As established in Lem. 3.5 and Thm. 4.6, the averaged baseline defined in (8) is the unique choice that ensures the Hessian estimator remains unbiased. Consequently, we employ the following estimator, constructed from a finite batch of queries using the averaged baseline:

**Definition 4.1** (Hessian Approximation). We propose a Hessian approximation $\widehat{\mathbf{H}}(\boldsymbol{\theta})$ to approximate $\nabla^2 F_\mu(\boldsymbol{\theta})$ in Prop. 3.3 by aggregating a set of $K$ queries $\{f(\mathbf{x}_k; \xi)\}_{k=1}^K$ with Gaussian perturbations $\mathbf{x}_k = \boldsymbol{\theta} + \mu\mathbf{u}_k$ and $\mathbf{u}_k \sim \mathcal{N}(\mathbf{0}, \mathbf{I}_d)$ via importance sampling:

$$\widehat{\mathbf{H}}(\boldsymbol{\theta}) \triangleq \frac{1}{K-1} \sum_{k=1}^K \frac{f(\mathbf{x}_k; \xi) - \hat{b}}{\mu^2} \mathbf{u}_k \mathbf{u}_k^\top \cdot \mathcal{W}(\mathbf{x}_k), \quad (9)$$

where the baseline is $\hat{b} \triangleq \frac{1}{K} \sum_{k=1}^K \mathcal{W}(\mathbf{x}_k) f(\mathbf{x}_k; \xi)$, and the importance sampling ratio is $\mathcal{W}(\mathbf{x}_k) \triangleq \frac{\pi_{\boldsymbol{\theta}}(\mathbf{x}_k)}{\rho(\mathbf{u}_k)}$.

**Remark.** The normalization factor $\frac{1}{K-1}$ applies a bias correction to account for the use of the sample mean $\hat{b}$ (see Thm. 4.6). Additionally, self-normalized importance sampling is utilized to enhance numerical stability and robustness. In Thm. 4.7, we further demonstrate that the averaged baseline is optimal for variance minimization under common sampling distributions.

**Query Reuse.** Adopting the strategy from (Qiu et al., 2025), the query reuse technique significantly reduces estimator variance by aggregating historical queries from recent optimization steps. Specifically, we maintain a history buffer $\mathcal{H}_t$ that stores the most recent $N \times K$ queries, comprising the random directions $\{\mathbf{u}_{t-i,k}\}_{i,k=1}^{N,K}$ and their corresponding function values $\{y_{t-i,k} \triangleq f(\boldsymbol{\theta}_{t-i} + \mu\mathbf{u}_{t-i,k}; \xi_{t-i})\}_{i,k=1}^{N,K}$. Notably, the memory overhead introduced by this mechanism is negligible in practice, as the history buffer requires storing only scalar function values and random seeds, rather than high-dimensional random vectors (see Appx. E.1).

By integrating the averaged baseline with query reuse technique, we define the complete ZoVH estimator as follows:

**Definition 4.2** (ZoVH (Zeroth-Order Variance-Reduced Hessian Estimator)). At iteration $t$, given the history buffer $\mathcal{H}_t$ containing the most recent $N \times K$ queries, the Hessian approximation is defined as:

$$\widehat{\mathbf{H}} \triangleq \frac{1}{NK-1} \sum_{i,k=1}^{N,K} \frac{y_{t-i,k} - \hat{b}_t}{\mu^2} \mathbf{u}_{t-i,k} \mathbf{u}_{t-i,k}^\top, \quad (10)$$

where the averaged baseline is $\hat{b}_t \triangleq \frac{1}{NK} \sum_{i,k=1}^{N,K} y_{t-i,k}$.

**Remark.** From the perspective of policy optimization in reinforcement learning, reusing past queries can be interpreted as a form of off-policy learning, where historical data is leveraged to refine current estimates. This perspective aligns with the importance sampling framework established in Prop. 3.7, wherein the proposal distribution $\rho$ corresponds to the off-policy distribution induced by historical perturbations, distinct from the target distribution $\pi_{\boldsymbol{\theta}}$ (which remains a Gaussian distribution centered at the current parameter

$\boldsymbol{\theta}$). Consequently, Def. 4.2 can be viewed as a special case of Def. 4.1, where the importance sampling ratio $\mathcal{W}(\mathbf{x})$ is implicitly integrated into the historical queries.

## 4.2. Theoretical Analysis

To facilitate the theoretical analysis, we focus on the simplified setting of $N = 1$ (i.e., without query reuse) in Def. 4.2. This choice is motivated by the observation that in local optimization, parameters typically exhibit slow evolution over short time horizons. Consequently, the proposal distribution $\rho$ closely approximates the target distribution $\pi_{\boldsymbol{\theta}}$, implying that the importance sampling ratio $\mathcal{W}(\mathbf{x}) \approx 1$ and that historical queries are approximately drawn from the target (on-policy) distribution. We also provide a theoretical analysis for the general case of $N > 1$ in Appx. B, which follows analogous reasoning.

We first adopt the following standard assumptions regarding the objective function $f(\boldsymbol{\theta}; \xi)$:

**Assumption 4.3** (Lipschitz Continuity). $\forall \boldsymbol{\theta}, \boldsymbol{\theta}' \in \mathbb{R}^d$,

$$\begin{aligned} |f(\boldsymbol{\theta}; \xi) - f(\boldsymbol{\theta}'; \xi)| &\leq L_0 \|\boldsymbol{\theta} - \boldsymbol{\theta}'\| \\ \|\nabla f(\boldsymbol{\theta}; \xi) - \nabla f(\boldsymbol{\theta}'; \xi)\| &\leq L_1 \|\boldsymbol{\theta} - \boldsymbol{\theta}'\| \\ \|\nabla^2 f(\boldsymbol{\theta}; \xi) - \nabla^2 f(\boldsymbol{\theta}'; \xi)\|_F &\leq L_2 \|\boldsymbol{\theta} - \boldsymbol{\theta}'\| . \end{aligned} \quad (11)$$

**Assumption 4.4** (Bounded Variance). $\forall \boldsymbol{\theta} \in \mathbb{R}^d$

$$\mathbb{E}_\xi \left[ |f(\boldsymbol{\theta}; \xi) - F(\boldsymbol{\theta})|^2 \right] \leq \sigma_\xi^2 \quad (12)$$

We then restate the gradient and Hessian of the Gaussian smoothed objective (7) to support our subsequent analysis:

**Lemma 4.5** (Smoothed Gradient and Hessian). *For a smoothing parameter $\mu > 0$, the gradient and Hessian of the smoothed objective $F_\mu(\boldsymbol{\theta})$ with Gaussian smoothing distribution $\mathcal{N}(\mathbf{0}, \mathbf{I}_d)$ satisfy:*

$$\nabla F_\mu(\boldsymbol{\theta}) = \frac{1}{\mu} \mathbb{E}_{\mathbf{u} \sim \mathcal{N}(\mathbf{0}, \mathbf{I}_d)}[F(\boldsymbol{\theta} + \mu\mathbf{u})\mathbf{u}]$$

$$\nabla^2 F_\mu(\boldsymbol{\theta}) = \frac{1}{\mu^2} \mathbb{E}_{\mathbf{u} \sim \mathcal{N}(\mathbf{0}, \mathbf{I}_d)} \left[ (\mathbf{u}\mathbf{u}^\top - \mathbf{I}_d)F(\boldsymbol{\theta} + \mu\mathbf{u}) \right].$$

Based on Lem. 4.5, we prove the unbiasedness of the Hessian estimator (9):

**Theorem 4.6** (Bias of Hessian Estimator). *The Hessian estimator $\widehat{\mathbf{H}}(\boldsymbol{\theta})$ in (9) is an unbiased estimator of the Hessian of the smoothed objective (4.5):*

$$\mathbb{E}[\widehat{\mathbf{H}}(\boldsymbol{\theta})] = \nabla^2 F_\mu(\boldsymbol{\theta}) .$$

**Remark.** The proof is provided in Appx. C.8. Thm. 4.6 extends Lem. 3.5 by demonstrating the unbiasedness of the proposed Hessian estimator (9) with respect to $\nabla^2 F_\mu(\boldsymbol{\theta})$ when employing the averaged baseline. Notably, the bias

relative to the true Hessian $\nabla^2 F(\boldsymbol{\theta})$ persists, stemming from the inherent discrepancy between $\nabla^2 F_\mu(\boldsymbol{\theta})$ and $\nabla^2 F(\boldsymbol{\theta})$ (see Thm. 4.8 and Appx. D.2).

**Theorem 4.7** (Optimal Baseline). *The optimal baseline $b^*$ that minimizes the variance of the Hessian estimator* $\mathrm{Var}\left(\widehat{\mathbf{H}}(\boldsymbol{\theta})\right) \triangleq \mathbb{E}\left[\left\|\widehat{\mathbf{H}}(\boldsymbol{\theta}) - \nabla^2 F_\mu(\boldsymbol{\theta})\right\|_F^2\right]$ *is:*

$$b^* = \frac{\mathbb{E}_{\mathbf{u}}\left[F(\boldsymbol{\theta} + \mu\mathbf{u})\left\|\left(\mathbf{u}_k\mathbf{u}_k^\top - \mathbf{I}_d\right)\right\|_F^2\right]}{\mathbb{E}_{\mathbf{u}}\left[\left\|\left(\mathbf{u}_k\mathbf{u}_k^\top - \mathbf{I}_d\right)\right\|_F^2\right]} \ . \quad (13)$$

*In particular, when $\mathbf{u} \sim \mathcal{N}(0, \mathbf{I}_d)$ as $d \to \infty$, such that $\left\|\left(\mathbf{u}_k\mathbf{u}_k^\top - \mathbf{I}_d\right)\right\|_F^2$ is effectively constant, the optimal baseline reduces to:*

$$b^* = \mathbb{E}_{\mathbf{u}}[F(\boldsymbol{\theta} + \mu\mathbf{u})] = F_\mu(\boldsymbol{\theta}) \ .$$

**Remark.** The proof is provided in Appx. C.9. Thm. 4.7 provides theoretical justification for using the averaged baseline in our Hessian estimator. It presents the explicit form of the optimal baseline that minimizes estimator variance without making assumptions about the distribution of random directions. If the random directions are sampled from standard Gaussian distribution, $\left\|\left(\mathbf{u}_k\mathbf{u}_k^\top - \mathbf{I}_d\right)\right\|_F^2$ concentrates around the constant $d(d+1)$ for large dimensions $d$. Thus, the averaged baseline remains a robust approximation of the optimal baseline. Besides, the optimal baseline is also related to control-variate view (detailed in Appx. D.3).

**Theorem 4.8** (Bias-Variance Decomposition of Hessian Estimator). *Under Assump. 4.3 and 4.4, with the optimal baseline $b = F_\mu(\boldsymbol{\theta})$ (as in Thm. 4.7) and $\mathbf{u} \sim \mathcal{N}(\mathbf{0}, \mathbf{I}_d)$, the expected Frobenius norm error of the Hessian estimator $\widehat{\mathbf{H}}(\boldsymbol{\theta})$ with respect to the true Hessian $\nabla^2 F(\boldsymbol{\theta})$ is bounded by:*

$$\mathbb{E}\left[\left\|\widehat{\mathbf{H}}(\boldsymbol{\theta}) - \nabla^2 F(\boldsymbol{\theta})\right\|_F^2\right] \leq \underbrace{\frac{Kd(d+2)V}{(K-1)^2\mu^4}}_{Variance} + \underbrace{L_2^2\mu^2 d}_{Squared\ Bias} \ ,$$

*where $V \triangleq \sigma_\xi^2 + 4L_0^2\mu^2(d+2)$.*

**Remark.** The proof is provided in Appx. C.10. Thm. 4.8 characterizes the bias-variance trade-off of the Hessian estimator. The variance term captures stochastic fluctuations $\sigma_\xi^2$ and the Gaussian smoothing effect (the second term in $V$), while the squared bias term quantifies the discrepancy between the smoothed and true Hessians. Notably, the variance decreases with a larger smoothing radius $\mu$ and increased random directions $K$, whereas the bias increases with $\mu$. Consequently, selecting an appropriate smoothing radius $\mu$ is critical for minimizing the overall error.

# 5. Efficient Inverse Hessian Approximations

Although the Hessian estimator in Def. 4.1 is valuable, practical curvature-aware ZO algorithms typically require the inverse Hessian or its product with the gradient. In this section, we extend ZoVH to derive efficient approximations for both the inverse Hessian (Sec. 5.1) and the inverse Hessian-gradient product (Sec. 5.2) by leveraging the structure of our estimator.

To simplify notation, we omit the explicit dependence on the iteration $t$ throughout this section. Furthermore, when query reuse is employed, $K$ denotes the total number of queries, encompassing both current and historical samples.

## 5.1. Inverse Hessian Approximation

By utilizing the Woodbury matrix identity (Woodbury, 1950), we can efficiently compute the inverse of the Hessian estimator in Def. 4.1 without explicitly inverting a dense matrix:

**Definition 5.1** (Inverse Hessian Approximation). Let $\widehat{\mathbf{H}}(\boldsymbol{\theta})$ be the Hessian estimator and $\hat{b}$ be the averaged baseline (8). Define the scalar $\nu_k \triangleq \mathcal{W}(\mathbf{x}_k)\left(\frac{f(\mathbf{x}_k;\xi)-\hat{b}}{\mu^2}\right)$, the matrix $\mathbf{U} \triangleq [\mathbf{u}_1, \ldots, \mathbf{u}_K]$, and $\mathbf{D} \triangleq \mathrm{diag}\left(\frac{\nu_1}{K-1}, \ldots, \frac{\nu_K}{K-1}\right)$. Then for any regularization parameter $\lambda > 0$, the exact inverse of the regularized Hessian estimator is:

$$\left(\widehat{\mathbf{H}}(\boldsymbol{\theta}) + \lambda\mathbf{I}_d\right)^{-1} = \frac{1}{\lambda}\mathbf{I}_d - \frac{1}{\lambda^2}\mathbf{U}\left(\mathbf{D}^{-1} + \frac{1}{\lambda}\mathbf{U}^\top\mathbf{U}\right)^{-1}\mathbf{U}^\top \ . \quad (14)$$

Further approximating $\mathbf{U}^\top\mathbf{U}$ by $\mathrm{diag}(\mathbf{U}^\top\mathbf{U})$ yields the approximation of $\left(\widehat{\mathbf{H}}(\boldsymbol{\theta}) + \lambda\mathbf{I}_d\right)^{-1}$ as below:

$$\widetilde{\mathbf{H}}^{-1}(\boldsymbol{\theta}) \triangleq \frac{1}{\lambda}\mathbf{I}_d - \sum_{k=1}^K \frac{\nu_k}{\lambda^2(K-1) + \lambda\nu_k\|\mathbf{u}_k\|^2}\mathbf{u}_k\mathbf{u}_k^\top \ . \quad (15)$$

Note that this approximation becomes exact if the random directions $\{\mathbf{u}_k\}_{k=1}^K$ are pairwise orthogonal.

**Remark.** The detailed derivation is provided in Appx. C.11. Since $\widehat{\mathbf{H}}(\boldsymbol{\theta})$ is rank-$K$ (where $K \ll d$), the regularization term $\lambda\mathbf{I}_d$ is essential for invertibility. Strict equality in (15) holds when the random directions are pairwise orthogonal. This condition is naturally satisfied by uniform spherical or coordinate-basis sampling and is approximately satisfied by standard Gaussian sampling when the dimension $d$ is large. In Appx. D.4, we further quantify the gap between the exact inverse (14) and the approximation (15) under standard Gaussian sampling, showing that the theoretical approximation error is $\mathcal{O}_p(K/\sqrt{d})$ and empirically decays as $d^{-1/2}$ for fixed small $K$.

We then analyze the error of the inverse Hessian approximation $\widetilde{\mathbf{H}}^{-1}(\boldsymbol{\theta})$ defined in (15) with respect to the inverse of

the true Hessian $\nabla^2 F(\boldsymbol{\theta})$:

**Theorem 5.2** (Bias of Inverse Hessian Estimator). *Assume that there exists a positive constant $\rho \in (0, 1]$ such that $|\nu_k \|\mathbf{u}_k\|^2 / \lambda(K-1) + 1| \geq \rho$ for all $k \in [K]$. Under Assump. 4.3 and the orthogonality of random directions $\{\mathbf{u}_k\}_{k=1}^K$, the bias of the inverse Hessian approximation in (15) with respect to the inverse of the true Hessian is bounded by:*

$$\mathbb{E}\left[\left\|\widetilde{\mathbf{H}}^{-1}(\boldsymbol{\theta}) - \left(\nabla^2 F(\boldsymbol{\theta}) + \lambda \mathbf{I}_d\right)^{-1}\right\|_F^2\right]$$

$$\leq \frac{d}{\lambda^2 \rho^2 (\sigma_{\min} + \lambda)^2} \left(\frac{K d(d+2)V}{(K-1)^2 \mu^4} + L_2^2 \mu^2 d\right),$$

*where $\sigma_{\min}$ denotes the minimum eigenvalue of the true Hessian $\nabla^2 F(\boldsymbol{\theta})$.*

**Remark.** The proof is provided in Appx. C.13. Thm. 5.2 bounds the approximation error, showing it depends on the Hessian estimation error (Thm. 4.8) scaled by three factors: *(1)* The regularization parameter $\lambda$, ensuring the invertibility and stability. A larger $\lambda$ reduces the bound but may over-smooth curvature, necessitating a balanced choice. *(2)* The minimum eigenvalue $\sigma_{\min}$ of the true Hessian. A larger $\sigma_{\min}$ leads to a smaller error bound. *(3)* The constant $\rho$, ensuring non-singularity in the Woodbury update. In high dimensions where $\mathbb{E}[\|\mathbf{u}\|^2] \approx d$, the term $|\nu_k \|\mathbf{u}_k\|^2 / \lambda(K-1) + 1|$ is typically sufficient to guarantee a reasonable $\rho$.

### 5.2. Inverse Hessian-Gradient Product Approximation

Combining the ZO gradient estimator (6) with the inverse Hessian approximation (15), we compute the product $\widetilde{\mathbf{H}}^{-1}(\boldsymbol{\theta})\hat{\nabla}F(\boldsymbol{\theta})$. We consider two sampling strategies based on whether the gradient and Hessian estimators share the same set of random directions:

**Regime 1: Decoupled Sampling.** When the Hessian and gradient estimators use independent sets of random directions, they are statistically uncorrelated, allowing direct multiplication of gradient estimator (6) and Hessian estimator (15). However, this doubles the query complexity to $2K$ evaluations per iteration, which may be impractical.

**Regime 2: Shared Sampling.** Using the same directions $\{\mathbf{u}_k\}_{k=1}^K$ for both estimators reduces the cost to $K$ evaluations but introduces correlation bias. To mitigate this while maintaining efficiency, we propose a bias-corrected estimator:

**Definition 5.3** (Bias-Corrected Shared Product). Assume $\widetilde{\mathbf{H}}^{-1}(\boldsymbol{\theta})$ and $\hat{\nabla}F(\boldsymbol{\theta})$ share the same standard Gaussian directions $\{\mathbf{u}_k\}_{k=1}^K$. The bias-corrected product approxima-

tion is defined as:

$$\widetilde{\mathbf{H}}^{-1}(\boldsymbol{\theta})\hat{\nabla}F(\boldsymbol{\theta}) \triangleq \sum_{k=1}^K \mu \nu_k \left(\frac{1}{\lambda(K-1)}\right.$$

$$\left. - \frac{\mathbf{u}_k^\top \left(\sum_{k'=1, k' \neq k}^K \nu_{k'} \mathbf{u}_{k'}\right)/(K-2)}{\lambda^2(K-1) + \lambda \nu_k \|\mathbf{u}_k\|^2}\right) \mathbf{u}_k. \quad (16)$$

**Remark.** The derivation is given in Appx. C.12. We use a leave-one-out strategy to remove self-interaction terms $\nu_k \mathbf{u}_k$, significantly reducing bias. Defining the global weighted sum $\mathbf{s} \triangleq \sum_{j=1}^K \nu_j \mathbf{u}_j$, the interaction term can be efficiently computed as $\mathbf{u}_k^\top (\sum_{j \neq k} \nu_j \mathbf{u}_j) = \mathbf{u}_k^\top \mathbf{s} - \nu_k \|\mathbf{u}_k\|^2$. This allows computation in $\mathcal{O}(Kd)$ time, avoiding the $\mathcal{O}(K^2 d)$ cost of pairwise operations. Additionally, since the directions are reused, no extra function evaluations are needed.

**Advantages.** Def. 5.3 provides an efficient framework for curvature-aware zeroth-order optimization with minimal query overhead, as outlined in Alg. 1. This approach offers several advantages: *(a)* it preserves efficiency by sharing query directions for both gradient and Hessian estimation, requiring only $K$ function evaluations per iteration, and is identical to the cost of (Zhu, 2022; Zhao et al., 2025); *(b)* it benefits from variance reduction via an averaged baseline, and can be further enhanced by integrating a query reuse mechanism, as detailed in Appx. D.5; and *(c)* it is easy to implement, requiring only a straightforward modification to standard ZO optimizers to incorporate the product approximation in Def. 5.3.

---

**Algorithm 1** ZoVH for Curvature-Aware Zeroth-Order Optimization

**Require:** Initial parameter $\boldsymbol{\theta}_0$, smoothing parameter $\mu$, regularization parameter $\lambda$, learning rate $\eta$, history size $N$, number of random directions $K$, total iterations $T$.

1: Initialize history buffer $\mathcal{H} \leftarrow \emptyset$.
2: **for** $t = 0$ to $T - 1$ **do**
3:      Sample $K$ random directions $\{\mathbf{u}_{t,k}\}_{k=1}^K$.
4:      Evaluate values $\{y_{t,k} \triangleq f(\boldsymbol{\theta}_t + \mu \mathbf{u}_{t,k}; \xi_t)\}_{k=1}^K$.
5:      Update history buffer: $\mathcal{H} \leftarrow \mathcal{H} \cup \{(\mathbf{u}_{t,k}, y_{t,k})\}_{k=1}^K$.
6:      Maintain buffer size: if $|\mathcal{H}| > NK$, remove the oldest $K$ entries.
7:      Compute $\widetilde{\mathbf{H}}^{-1}(\boldsymbol{\theta}_t)\hat{\nabla}F(\boldsymbol{\theta}_t)$ using $\mathcal{H}$ via (16) (when $N = 1$) or (122) (when $N > 1$).
8:      Update parameters: $\boldsymbol{\theta}_{t+1} = \boldsymbol{\theta}_t - \eta \widetilde{\mathbf{H}}^{-1}(\boldsymbol{\theta}_t)\hat{\nabla}F(\boldsymbol{\theta}_t)$.
9: **end for**
**Ensure:** $\boldsymbol{\theta}_T$

---

We then analyze the error of the inverse Hessian-gradient product approximation $\widetilde{\mathbf{H}}^{-1}(\boldsymbol{\theta})\hat{\nabla}F(\boldsymbol{\theta})$ from (16) relative to the true product $\left(\nabla^2 F(\boldsymbol{\theta}) + \lambda \mathbf{I}_d\right)^{-1} \nabla F(\boldsymbol{\theta})$:

**Theorem 5.4** (Bias of Inverse Hessian-Gradient Product Estimator (Informal)). *Assume $f(\boldsymbol{\theta}; \xi)$ is bounded. Under Assump. 4.3 and 4.4, and assuming orthogonality of $\{\mathbf{u}_k\}_{k=1}^{K}$, the bias of the estimator (16) is bounded by:*

$$\mathbb{E}\left[\left\|\widetilde{\mathbf{H}}^{-1}(\boldsymbol{\theta})\hat{\nabla}F(\boldsymbol{\theta}) - \left(\nabla^2 F(\boldsymbol{\theta}) + \lambda \mathbf{I}_d\right)^{-1}\nabla F(\boldsymbol{\theta})\right\|^2\right]$$

$$\leq \frac{Kd^2\left(B_1\sigma_\xi^2 + 4B_2L_0^2\mu^2\right)}{\lambda^2(\sigma_{\min} + \lambda)^2(K-1)^2},$$

*where $B_1$, $B_2$ are some constants, and $\sigma_{\min}$ denotes the minimum eigenvalue of the true Hessian $\nabla^2 F(\boldsymbol{\theta})$.*

**Remark.** The formal statement is provided in Thm. C.8 and proof are available in Appx. C.14. Thm. 5.4 shows that the error bound of the inverse Hessian-gradient product approximation shares the same structure as that of the inverse Hessian approximation in Thm. 5.2, differing only by constant factors $B_1$ and $B_2$, since the biased-corrected multiplication in (16) mitigates the correlation-induced bias.

We provide an convergence guarantee for Alg. 1 below:

**Theorem 5.5** (Convergence of Alg. 1 (Informal)). *Under Assump. 4.3 and 4.4, employing the optimal baseline $b_t$ from Thm. 4.7, with $\eta \sim \mathcal{O}(\epsilon^2)$ and $T \sim \mathcal{O}(\epsilon^{-4})$, we have:*

$$\frac{1}{T}\sum_{t=0}^{T-1}\mathbb{E}[\|\nabla F(\boldsymbol{\theta}_t)\|] \leq \epsilon + \frac{\mu L_1 d}{2}.$$

**Remark.** The formal statement is provided in Thm. C.9 and detailed proof are available in Appx. C.15. Thm. 5.5 establishes the convergence of ZoVH to a stationary point, up to a bias term induced by the Gaussian smoothing. By appropriately tuning the learning rate $\eta$, iteration count $T$, and smoothing parameter $\mu$, the algorithm can achieve convergence to an $\epsilon$-stationary point.

# 6. Experiments

We evaluate the performance of ZoVH through empirical Hessian error analysis on synthetic functions and a neural network (Sec. 6.1), convergence benchmarks on synthetic optimization (Sec. 6.2) and black-box adversarial attack (Sec. 6.3). Additional experiments on curvature-aware zeroth-order LLM fine-tuning and ablation study of the averaged baseline and query reuse techniques are detailed in Appx. E.

## 6.1. Empirical Error Analysis of Hessian Approximation

We first compare the empirical Frobenius norm error of various Hessian estimators against the ground-truth Hessian. We assess the two-point (3) and three-point (4) Stein Estimators, the randomized central-difference estimator (5),

and ZoVH (both with and without query reuse technique). The one-point Stein estimator (2) is excluded due to its significantly higher error than other methods.

**Synthetic Functions.** We employ three synthetic benchmarks: Quadratic, Rosenbrock, and Styblinski-Tang functions. For each, we initialize $\boldsymbol{\theta}_0 \in \mathbb{R}^d$ ($d = 5000$) and perform gradient descent steps, collecting 25 test points along the trajectory (detailed in Appx. F.1). Fig. 2 demonstrates that ZoVH with $N = 6$ (utilizing query reuse) consistently yields the lowest estimation error. Even ZoVH with $N = 1$ (without query reuse), outperforms all baselines, achieving an $8\times$ accuracy improvement over the central-difference estimator (5) on Quadratic and Rosenbrock functions, and $3.4\times$ on Styblinski-Tang. The reduced error bars for ZoVH with $N = 6$ further confirm the variance reduction efficacy of query reuse.

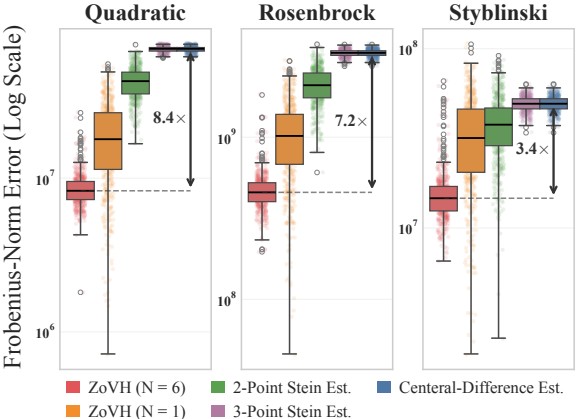

*Figure 2.* Frobenius norm Hessian error on three synthetic functions. Results are averaged over 20 random initializations and 25 test points along each optimization trajectory. The outliers are defined as points outside 1.5 times the interquartile range.

**Neural Network Hessian.** We extend our evaluation to a Convolutional Neural Network (CNN) trained on MNIST (LeCun et al., 1998). The network comprises two convolutional and two fully connected layers. We collect 1875 test points along a complete SGD training trajectory (detailed in Appx. F.1). The results in Fig. 3 reveals that ZoVH with $N = 4$ (utilizing query reuse) consistently achieves the lowest estimation error across all layers. Notably, for the second fully connected layer (*Fc2.weight*), ZoVH with $N = 4$ achieves a $9\times$ error reduction compared to the central-difference estimator.

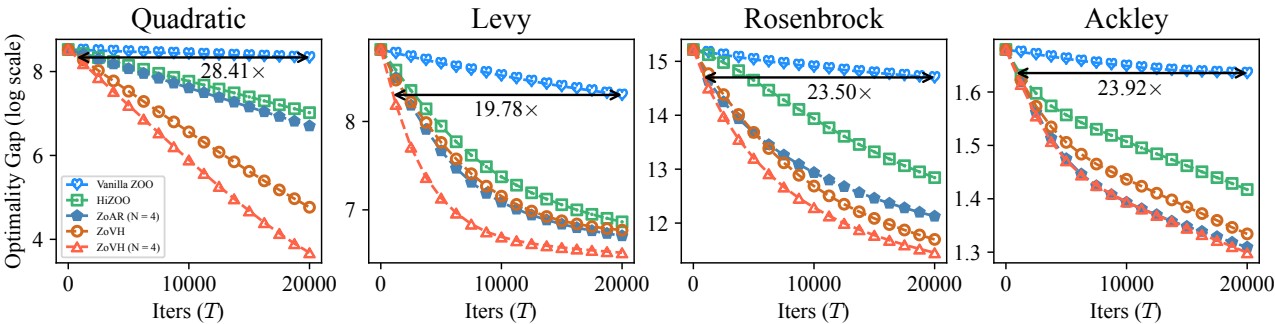

*Figure 1.* Comparison of convergence among different ZO Hessian optimization algorithms on four synthetic functions. All curves are averaged over 10 independent runs.

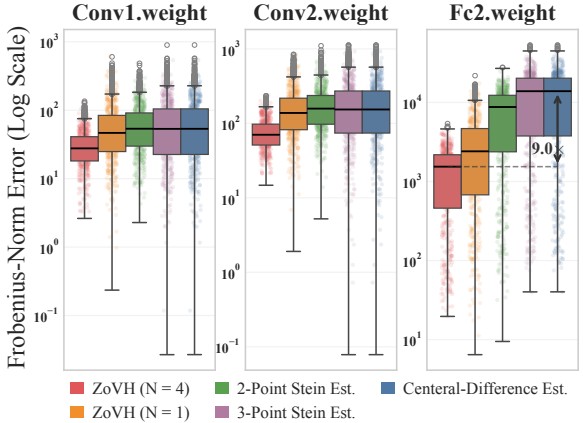

*Figure 3.* Frobenius norm Hessian error across CNN layers. Results averaged over 3 independent runs and 1875 test points per training trajectory. The outliers are defined as points outside 1.5 times the interquartile range.

### 6.2. Synthetic Function Optimization

We then evaluate the convergence of ZoVH against Vanilla ZOO (Nesterov & Spokoiny, 2017), HiZOO (Zhao et al., 2025), and ZoAR (Qiu et al., 2025) on four synthetic functions: Quadratic, Levy, Rosenbrock, and Ackley (detailed in Appx. F.2). As shown in Fig. 1, ZoVH with $N = 4$ (utilizing query reuse) significantly outperforms baselines in both convergence speed and final accuracy. Notably, ZoVH achieves an average speedup of $22\times$ over Vanilla ZOO, underscoring its efficiency in high-dimensional ZOO.

### 6.3. Black-Box Adversarial Attack

We further evaluate the performance of ZoVH in the domain of black-box adversarial attacks, a prominent application of zeroth-order optimization (Zhu, 2022; Shu et al., 2023). In this scenario, the objective is to identify an optimal perturbation $\delta$ for a given input image $x$ such that a target black-box model misclassifies $x + \delta$. The attack is performed using a convolutional neural network (CNN) trained on the MNIST

dataset (LeCun et al., 1998). The compared methods are the same as synthetic experiments in Sec. 6.2 (detailed in Appx. F.3). To evaluate the efficiency of the algorithms, we measure the minimum number of iterations to achieve a successful attack. The results in Tab. 1 show that ZoVH with $N = 4$ (utilizing query reuse) consistently requires fewer iterations compared to other baseline methods, indicating its superior efficiency in generating adversarial examples. Specifically, ZoVH with $N = 1$ (w/o query reuse) achieves a $3\times$ speedup over Vanilla ZOO, while ZoVH with $N = 4$ (utilizing query reuse) further achieves a $4\times$ speedup.

*Table 1.* Comparison of the minimal number of iterations to achieve a successful attack for different ZO Hessian methods. Results are averaged over 10 runs. The speedup is compared against the Vanilla ZOO.

| Method | # Iters | Speedup |
|---|---|---|
| Vanilla ZOO | $1629 \pm 610$ | $1.00\times$ |
| HiZOO | $1045 \pm 305$ | $1.56\times$ |
| ZoAR ($N = 4$) | $538 \pm 113$ | $3.03\times$ |
| ZoVH ($N = 1$) | $536 \pm 133$ | $3.04\times$ |
| **ZoVH ($N = 4$)** | $\mathbf{403 \pm 76}$ | $\mathbf{4.04\times}$ |

## 7. Conclusion

We introduced a unified framework connecting zeroth-order Hessian approximation with single-step policy optimization to derive ZoVH, a variance-reduced Hessian estimator. Leveraging an optimal baseline and query reuse, ZoVH minimizes estimation error efficiently without increasing the query budget. We further developed stable inverse Hessian approximations for a practical curvature-aware zeroth-order algorithm with proven convergence guarantees. Experiments across synthetic benchmarks, adversarial attacks, and LLM fine-tuning confirm that ZoVH achieves superior convergence speed and accuracy, providing a robust foundation for curvature-aware derivative-free optimization.

## Impact Statement

This paper presents a novel perspective on zeroth-order Hessian approximation framed through single-step policy optimization. Our findings contribute to the theoretical foundations of derivative-free optimization and have the potential to enhance the efficiency of algorithms used in black-box optimization settings. We do not anticipate any specific negative ethical implications or adverse societal impacts resulting from this work.

## Acknowledgements

This work was supported in part by the Youth S&T Talent Support Programme of Guangdong Provincial Association for Science and Technology (Grant No. SKXRC2025466).

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

# A. Related Works

**Derivative-Free Hessian Approximation.** Early efforts on derivative-free Hessian approximation date back to coordinate-wise perturbation schemes that form second-order updates by probing each coordinate direction, which typically requires on the order of $\mathcal{O}(d^2)$ function evaluations per iteration for an $d$-dimensional problem (Fabian, 1971). To reduce this query cost, subsequent work moved from coordinate perturbations to random perturbations for curvature estimation. A representative milestone is Spall's second-order SPSA (2SPSA, Spall (2000)), which estimates the Hessian using only four function evaluations per update, independent of the dimension, yielding a substantial improvement over earlier $\mathcal{O}(d^2)$ query constructions. In a similar spirit, randomized finite difference methods were developed for estimating Hessian. One notable line of work is Random Directions Stochastic Approximation (RDSA, L. A. et al. (2017)), which employs central sampling to construct Hessian estimates with uniform or asymmetrical Bernoulli distributed directions. Furthermore, Balasubramanian & Ghadimi (2022) extended the central sampling approach to Gaussian perturbations, deriving a family of unbiased ZO Hessian estimators via Stein's identity. On the application side, ZO Hessian approximations have been leveraged to improve query efficiency and optimization performance in black-box adversarial attacks (Ye et al., 2025) and in curvature-aware ZO fine-tuning of large language models (Zhao et al., 2025).

**Variance-Reduced Zeroth-Order Optimization.** Zeroth-Order Optimization (ZOO) aims to minimize black-box objectives using only function evaluations, and has been extensively studied due to its broad applicability when derivatives are unavailable. A classical line of work constructs ZO gradient estimators via randomized smoothing and finite differences, including Gaussian smoothing (Nesterov & Spokoiny, 2017), uniform sampling on the unit sphere (Flaxman et al., 2005), and coordinate-wise perturbations (Lian et al., 2016). While these estimators are simple and widely used, they typically suffer from high variance in noisy and high-dimensional regimes, leading to slow convergence and substantial query complexity. To mitigate this issue, recent studies develop variance-reduced ZO methods by leveraging past queries to reduce the variance of ZO gradient estimates (Shu et al., 2023; Wang et al., 2024; Qiu et al., 2025). Another complementary direction proposes new optimization paradigms that are variance-efficient by design, thereby improving ZO optimization convergence rate and final performance (Shu et al., 2025). Our work is most closely related to variance reduction in ZOO, but differs in that we focus on the approximation of second-order information, i.e., the Hessian matrix.

# B. Theoretical Analysis for the General Case $N > 1$

We now extend the theoretical analysis to the query-reuse setting with $N > 1$. In this case, the estimator uses a history buffer of $NK$ function queries rather than only the $K$ queries from the current iteration. This larger sample size reduces the variance of the Hessian estimator, while the use of historical queries introduces an additional error because those queries were generated around previous iterates. The analysis below makes this bias-variance trade-off explicit. We first state the resulting Hessian estimation bound (Thm. B.1) and then explain how the same bound affects the inverse Hessian approximation, the inverse Hessian-gradient product, and the convergence guarantee.

**Theorem B.1** (Bias-Variance Decomposition for $N > 1$). *Under Assump. 4.3 and 4.4, with the optimal baseline $b = F_\mu(\boldsymbol{\theta})$ and $\mathbf{u} \sim \mathcal{N}(\mathbf{0}, \mathbf{I}_d)$. Let $N > 1$, the expected Frobenius norm error of the query-reuse Hessian estimator $\widehat{\mathbf{H}}_t(\boldsymbol{\theta}_t)$ at iteration $t$ defined in Def. 4.2 is bounded by:*

$$\mathbb{E}\left[\left\|\widehat{\mathbf{H}}(\boldsymbol{\theta}) - \nabla^2 F(\boldsymbol{\theta})\right\|_F^2\right] \le \frac{NKd(d+2)\lambda^2 V}{\lambda^2(NK-1)^2\mu^4 - L_0^2\eta^2\mu^2K(N^2-1)d(d+2)/3} + L_2^2\mu^2 d \, ,$$

*where $V \triangleq \sigma_\xi^2 + 4L_0^2\mu^2(d+2)$.*

**Remark.** The proof is provided in Appx. C.16. Thm. B.1 extends the bias-variance decomposition in Thm. 4.8 to the $N > 1$ (query reuse) setting. Compared with the $N = 1$ case, reusing queries across $N$ iterations increases the effective sample size from $K$ to $NK$, reducing the leading variance term through the factor $(NK-1)^2$. This gain is offset by the additional drift term $L_0^2\eta^2\mu^2K(N^2-1)d(d+2)/3$ in the denominator, which captures the error induced by reusing historical queries generated at previous iterates. When $N = 1$, this drift term vanishes and the bound recovers Thm. 4.8. Therefore, for a small learning rate $\eta$ and a moderate history length $N$, query reuse can reduce the Hessian estimation error and improve the constants in the convergence bound of Thm. 5.5, while an overly large $N$ or rapid parameter updates may offset the variance reduction.

**Extension to Inverse Hessian Approximation.** The same resolvent-identity argument used in Appx. C.13 gives the corresponding inverse Hessian bound. Let the $N > 1$ Hessian estimation error bound be

$$\mathcal{E}_{H,N} \triangleq \frac{NKd(d+2)\lambda^2 V}{\lambda^2(NK-1)^2\mu^4 - L_0^2\eta^2\mu^2 K(N^2-1)d(d+2)/3} + L_2^2\mu^2 d. \tag{17}$$

With this notation, the inverse Hessian approximation satisfies

$$\mathbb{E}\left[\left\|\widetilde{\mathbf{H}}_N^{-1}(\boldsymbol{\theta}) - \left(\nabla^2 F(\boldsymbol{\theta}) + \lambda\mathbf{I}_d\right)^{-1}\right\|_F^2\right] \leq \frac{d}{\lambda^2\rho^2(\sigma_{\min}+\lambda)^2}\mathcal{E}_{H,N}. \tag{18}$$

Relative to Thm. 5.2, query reuse mainly replaces the $K$-sample Hessian error by the $NK$-sample error in $\mathcal{E}_{H,N}$. The regularization factor $\lambda$, the non-singularity constant $\rho$, and the spectral factor $\sigma_{\min} + \lambda$ play the same stability roles as in the $N = 1$ analysis. When the history buffer is moderate and the iterate drift is small, $\mathcal{E}_{H,N}$ can be smaller than the $N = 1$ counterpart. If $N$ or $\eta$ is too large, the drift term in the denominator may offset this benefit.

**Extension to Inverse Hessian-Gradient Product Approximation.** The same reasoning applies to the bias-corrected inverse Hessian-gradient product in Def. D.3. The leave-one-out correction is computed over the whole history buffer, so each current or historical query $(i, k)$ excludes all other pairs $(j, k') \neq (i, k)$ rather than only the other directions in the current iteration. Query reuse therefore increases the effective sample size in both the Hessian estimate and the shared inverse-Hessian-gradient product. Historical samples are still generated around previous iterates, so rapid parameter changes or an overly large buffer can make the reused samples stale. This is the same off-policy drift captured in Thm. B.1. Query reuse can improve the estimator when $N$ and $\eta$ are moderate, but the benefit can be offset if the history distribution moves too far from the current target distribution.

**Implication for Convergence.** This extension also clarifies the convergence guarantee in Thm. 5.5. The convergence bound depends on the estimation error of the inverse Hessian-gradient product, which is in turn controlled by the quality of the Hessian estimator and the variance bound in Thm. B.1. Improving the Hessian estimate through the optimal baseline and query reuse therefore improves the constants in the convergence bound. The resulting iteration complexity remains the standard $\mathcal{O}(\epsilon^{-4})$ zeroth-order rate when $N$ and $K$ are fixed. The theoretical role of query reuse is to explain the practical speedup through better estimator quality and smaller constants, rather than through a different asymptotic order.

## C. Proofs

### C.1. Useful Lemmas

**Lemma C.1** (Lipschitz Continuity for Objective). $\forall\boldsymbol{\theta}_1, \boldsymbol{\theta}_2 \in \mathbb{R}^d$, the objective (1) satisfies:

$$\begin{aligned}
|F(\boldsymbol{\theta}_1) - F(\boldsymbol{\theta}_2)| &\leq L_0\|\boldsymbol{\theta}_1 - \boldsymbol{\theta}_2\|, \\
\|\nabla F(\boldsymbol{\theta}_1) - \nabla F(\boldsymbol{\theta}_2)\| &\leq L_1\|\boldsymbol{\theta}_1 - \boldsymbol{\theta}_2\|, \\
\|\nabla^2 F(\boldsymbol{\theta}_1) - \nabla^2 F(\boldsymbol{\theta}_2)\| &\leq L_2\|\boldsymbol{\theta}_1 - \boldsymbol{\theta}_2\|.
\end{aligned} \tag{19}$$

*Proof.* Since $F(\boldsymbol{\theta}) = \mathbb{E}_\xi[f(\boldsymbol{\theta}; \xi)]$, the Lipschitz continuity of $F(\boldsymbol{\theta})$ follows:

$$|F(\boldsymbol{\theta}_1) - F(\boldsymbol{\theta}_2)| = |\mathbb{E}_\xi[f(\boldsymbol{\theta}_1; \xi) - f(\boldsymbol{\theta}_2; \xi)]| \overset{(a)}{\leq} \mathbb{E}_\xi[|f(\boldsymbol{\theta}_1; \xi) - f(\boldsymbol{\theta}_2; \xi)|] \overset{(b)}{\leq} L_0\|\boldsymbol{\theta}_1 - \boldsymbol{\theta}_2\|, \tag{20}$$

where $(a)$ follows from Jensen's inequality, and $(b)$ comes from Lem. C.1.

The Lipschitz continuity for the gradient and Hessian of $F(\boldsymbol{\theta})$ can be proved similarly. $\square$

**Lemma C.2** (Bias of Gradient Estimator (6)). Let smoothed objective $F_\mu(\boldsymbol{\theta})$ be (7), and the gradient estimator $\hat{\nabla}F(\boldsymbol{\theta})$ be (6), we have:

$$\mathbb{E}[\hat{\nabla}F(\boldsymbol{\theta})] = \nabla F_\mu(\boldsymbol{\theta}), \quad \text{and} \quad \left\|\hat{\nabla}F(\boldsymbol{\theta}) - \nabla F(\boldsymbol{\theta})\right\|^2 \leq \frac{\mu^2 L_1^2 d^2}{4}. \tag{21}$$

*Proof.* The unbiasedness of the gradient estimator $\hat{\nabla} F(\boldsymbol{\theta})$ follows from Stein's identity:

$$
\begin{aligned}
\mathbb{E}[\hat{\nabla} F(\boldsymbol{\theta})] =& \mathbb{E}\left[\frac{1}{K-1}\sum_{k=1}^{K}\frac{f(\boldsymbol{\theta}+\mu\mathbf{u}_k;\xi) - \frac{1}{K}\sum_{k'=1}^{K}f(\boldsymbol{\theta}+\mu\mathbf{u}_{k'};\xi)}{\mu}\mathbf{u}_k\right] \\
=& \mathbb{E}\left[\frac{1}{K-1}\sum_{k=1}^{K}\frac{\frac{K-1}{K}f(\boldsymbol{\theta}+\mu\mathbf{u}_k;\xi) - \frac{1}{K}\sum_{k'=1,k'\neq k}^{K}f(\boldsymbol{\theta}+\mu\mathbf{u}_{k'};\xi)}{\mu}\mathbf{u}_k\right] \\
=& \mathbb{E}\left[\frac{1}{K}\sum_{k=1}^{K}\frac{f(\boldsymbol{\theta}+\mu\mathbf{u}_k;\xi)}{\mu}\mathbf{u}_k\right] - \mathbb{E}\left[\frac{1}{K(K-1)}\sum_{k=1}^{K}\sum_{k'=1,k'\neq k}^{K}\frac{f(\boldsymbol{\theta}+\mu\mathbf{u}_{k'};\xi)}{\mu}\mathbf{u}_k\right] \\
\overset{(a)}{=}& \mathbb{E}_{\mathbf{u}}\left[\frac{F(\boldsymbol{\theta}+\mu\mathbf{u})}{\mu}\mathbf{u}\right] \overset{(b)}{=} \nabla F_\mu(\boldsymbol{\theta}),
\end{aligned}
\tag{22}
$$

where $(a)$ comes from the dependence of different random directions $\{\mathbf{u}_k\}_{k=1}^{K}$, and $(b)$ follows from Stein's identity.

For the bias between the smoothed gradient and the true gradient, we have:

$$
\|\nabla F_\mu(\boldsymbol{\theta}) - \nabla F(\boldsymbol{\theta})\|^2 \overset{(a)}{\leq} \mathbb{E}_{\mathbf{u}}\left[\|\nabla F(\boldsymbol{\theta}+\mu\mathbf{u}) - \nabla F(\boldsymbol{\theta})\|^2\right] \overset{(b)}{\leq} L_1^2\mu^2\mathbb{E}_{\mathbf{u}}\left[\|\mathbf{u}\|^2\right] \overset{(c)}{=} L_1^2\mu^2 d,
\tag{23}
$$

where $(a)$ follows from Jensen's inequality, $(b)$ comes from Lem. C.1, and $(c)$ results from the fact that $\mathbb{E}_{\mathbf{u}}[\|\mathbf{u}\|^2] = d$ for $\mathbf{u} \sim \mathcal{N}(0, \mathbf{I}_d)$.

Moreover, since the gradient estimator $\hat{\nabla} F(\boldsymbol{\theta})$ is constructed from $K$ dependent random directions, we have the variance bound:

$$
\begin{aligned}
\mathbb{E}\left[\left\|\hat{\nabla} F(\boldsymbol{\theta}) - \nabla F_\mu(\boldsymbol{\theta})\right\|^2\right] &\overset{(a)}{=} \mathbb{E}\left[\left\|\hat{\nabla} F(\boldsymbol{\theta})\right\|^2\right] - \|\nabla F_\mu(\boldsymbol{\theta})\|^2 \overset{(b)}{\leq} \mathbb{E}\left[\left\|\frac{1}{K-1}\sum_{k=1}^{K}\frac{f(\boldsymbol{\theta}+\mu\mathbf{u}_k;\xi) - F_\mu(\boldsymbol{\theta})}{\mu}\mathbf{u}_k\right\|^2\right] \\
&\overset{(c)}{\leq} \frac{K}{(K-1)^2\mu^2}\mathbb{E}\left[|f(\boldsymbol{\theta}+\mu\mathbf{u};\xi) - F_\mu(\boldsymbol{\theta})|^2\|\mathbf{u}\|^2\right] \overset{(d)}{\leq} \frac{Kd\left(\sigma_\xi^2 + 4L_0^2\mu^2(d+1)\right)}{(K-1)^2\mu^2},
\end{aligned}
\tag{24}
$$

where $(a)$ follows from (22), $(b)$ comes from getting rid of $\|F_\mu(\boldsymbol{\theta})\|^2$, $(c)$ follows from the dependence of random directions $\{\mathbf{u}_k\}_{k=1}^{K}$, and $(d)$ results from the following inequality:

$$
\begin{aligned}
\mathbb{E}\left[|f(\boldsymbol{\theta}+\mu\mathbf{u};\xi) - F_\mu(\boldsymbol{\theta})|^2\|\mathbf{u}\|^2\right] \overset{(a)}{=}& \mathbb{E}\left[|f(\boldsymbol{\theta}+\mu\mathbf{u};\xi) - F(\boldsymbol{\theta}+\mu\mathbf{u})|^2\|\mathbf{u}\|^2\right] + \mathbb{E}\left[|F(\boldsymbol{\theta}+\mu\mathbf{u}) - F_\mu(\boldsymbol{\theta})|^2\|\mathbf{u}\|^2\right] \\
\overset{(b)}{\leq}& \sigma_\xi^2\mathbb{E}[\|\mathbf{u}\|^2] + \mathbb{E}\left[|F(\boldsymbol{\theta}+\mu\mathbf{u}) - F_\mu(\boldsymbol{\theta})|^2\|\mathbf{u}\|^2\right] \\
\overset{(c)}{\leq}& \sigma_\xi^2 d + \mathbb{E}\left[2\left(|F(\boldsymbol{\theta}+\mu\mathbf{u}) - F(\boldsymbol{\theta})|^2 + 2|F(\boldsymbol{\theta}) - \mathbb{E}_{\mathbf{u}'}[F(\boldsymbol{\theta}+\mu\mathbf{u}')]|^2\right)\|\mathbf{u}\|^2\right] \\
\overset{(d)}{\leq}& \sigma_\xi^2 d + \mathbb{E}\left[2\left(|F(\boldsymbol{\theta}+\mu\mathbf{u}) - F(\boldsymbol{\theta})|^2 + 2\mathbb{E}_{\mathbf{u}'}\left[|F(\boldsymbol{\theta}) - F(\boldsymbol{\theta}+\mu\mathbf{u}')|^2\right]\right)\|\mathbf{u}\|^2\right] \\
\overset{(e)}{\leq}& \sigma_\xi^2 d + 2L_0^2\mu^2\mathbb{E}\left[\left(\|\mathbf{u}\|^2 + \mathbb{E}_{\mathbf{u}'}\left[\|\mathbf{u}'\|^2\right]\right)\|\mathbf{u}\|^2\right] \\
\overset{(f)}{=}& \sigma_\xi^2 d + 2L_0^2\mu^2\mathbb{E}\left[\left(\|\mathbf{u}\|^2 + d\right)\|\mathbf{u}\|^2\right] \overset{(g)}{=} d\left(\sigma_\xi^2 + 4L_0^2\mu^2(d+1)\right),
\end{aligned}
\tag{25}
$$

where $(a)$ follows from (1), $(b)$ comes from Assump. 4.4, $(c)$ results from the fact that $(a+b)^2 \leq 2(a^2+b^2)$, $(d)$ follows from Jensen's inequality, $(e)$ comes from Lem. C.1, and $(c)$, $(f)$, $(g)$ results from the fact that $\mathbb{E}[\|\mathbf{u}\|^2] = d$ and $\mathbb{E}[\|\mathbf{u}\|^4] = d(d+2)$ for $\mathbf{u} \sim \mathcal{N}(0, \mathbf{I}_d)$.

Overall, combining the bias between the smoothed gradient and the true gradient with the variance of the gradient estimator,

we have:

$$\mathbb{E}\left[\left\|\hat{\nabla}F(\boldsymbol{\theta}) - \nabla F(\boldsymbol{\theta})\right\|^2\right] \overset{(a)}{\leq} \mathbb{E}\left[\left\|\hat{\nabla}F(\boldsymbol{\theta}) - F_\mu(\boldsymbol{\theta})\right\|^2\right] + \|F_\mu(\boldsymbol{\theta}) - \nabla F(\boldsymbol{\theta})\|^2 \leq \frac{Kd\left(\sigma_\xi^2 + 4L_0^2\mu^2(d+1)\right)}{(K-1)^2\mu^2} + L_1^2\mu^2 d \,, \tag{26}$$

where $(a)$ follows from (22). $\qquad\square$

The following lemma bounds the bias between the smoothed Hessian and the true Hessian:

**Lemma C.3** (Bias of Smoothed Hessian). *Let smoothed objective $F_\mu(\boldsymbol{\theta})$ be* (7), *we have:*

$$\left\|\nabla^2 F_\mu(\boldsymbol{\theta}) - \nabla^2 F(\boldsymbol{\theta})\right\|_F^2 \leq L_2^2\mu^2 d \,. \tag{27}$$

*Proof.*

$$\left\|\nabla^2 F_\mu(\boldsymbol{\theta}) - \nabla^2 F(\boldsymbol{\theta})\right\|_F^2 \overset{(a)}{\leq} \mathbb{E}_{\mathbf{u}}\left[\left\|\nabla^2 F(\boldsymbol{\theta}+\mu\mathbf{u}) - \nabla^2 F(\boldsymbol{\theta})\right\|_F^2\right] \overset{(b)}{\leq} L_2^2\mu^2\mathbb{E}_{\mathbf{u}}\left[\|\mathbf{u}\|^2\right] \overset{(c)}{=} L_2^2\mu^2 d \,, \tag{28}$$

where $(a)$ follows from Jensen's inequality, $(b)$ comes from Lem. C.1, and $(c)$ results from the fact that $\mathbb{E}_{\mathbf{u}}[\|\mathbf{u}\|^2] = d$ for $\mathbf{u} \sim \mathcal{N}(0, \mathbf{I}_d)$. $\qquad\square$

**Lemma C.4** (Bound on Second Moment of $\nu$). *Let* $\nu = (f(\boldsymbol{\theta}+\mu\mathbf{u}; \xi) - F_\mu(\boldsymbol{\theta}))/\mu^2$, *where* $\mathbf{u} \sim \mathcal{N}(0, \mathbf{I}_d)$. *Under Assump. 4.3 and Lem. C.1, we have:*

$$\begin{aligned}
\mathbb{E}[\nu^2\|\mathbf{u}\|^2] &\leq \frac{d}{\mu^4}\left(\sigma_\xi^2 + 4L_0^2\mu^2(d+1)\right) \,, \\
\mathbb{E}[\nu^2\|\mathbf{u}\|^4] &\leq \frac{d(d+2)}{\mu^4}\left(\sigma_\xi^2 + 4L_0^2\mu^2(d+2)\right) \,, \\
\mathbb{E}[\nu^2\|\mathbf{u}\|^6] &\leq \frac{d(d+2)(d+4)}{\mu^4}\left(\sigma_\xi^2 + 4L_0^2\mu^2(d+3)\right) \,.
\end{aligned} \tag{29}$$

*Proof.* First,

$$\begin{aligned}
\mathbb{E}_{\mathbf{u}}[\nu^2\|\mathbf{u}\|^2] &= \frac{1}{\mu^4}\mathbb{E}_{\mathbf{u}}\left[|f(\boldsymbol{\theta}+\mu\mathbf{u}; \xi) - F_\mu(\boldsymbol{\theta})|^2\|\mathbf{u}\|^2\right] \\
&\overset{(a)}{=} \frac{1}{\mu^4}\mathbb{E}_{\mathbf{u}}\left[\left(|f(\boldsymbol{\theta}+\mu\mathbf{u}; \xi) - F(\boldsymbol{\theta}+\mu\mathbf{u})|^2 + |F(\boldsymbol{\theta}+\mu\mathbf{u}) - F_\mu(\boldsymbol{\theta})|^2\right)\|\mathbf{u}\|^2\right] \\
&\overset{(b)}{\leq} \frac{1}{\mu^4}\left(\sigma_\xi^2\mathbb{E}_{\mathbf{u}}[\|\mathbf{u}\|^2] + \mathbb{E}_{\mathbf{u}}\left[|F(\boldsymbol{\theta}+\mu\mathbf{u}) - F_\mu(\boldsymbol{\theta})|^2\|\mathbf{u}\|^2\right]\right) \\
&\overset{(c)}{=} \frac{1}{\mu^4}\left(\sigma_\xi^2 d + \mathbb{E}_{\mathbf{u}}\left[|F(\boldsymbol{\theta}+\mu\mathbf{u}) - F_\mu(\boldsymbol{\theta})|^2\|\mathbf{u}\|^2\right]\right) \,,
\end{aligned} \tag{30}$$

where $(a)$ follows from (1), i.e. $\mathbb{E}_\xi[f(\boldsymbol{\theta}+\mu\mathbf{u}; \xi) - F(\boldsymbol{\theta}+\mu\mathbf{u})] = 0$, $(b)$ comes from Assump. 4.4, and $(c)$ results from the fact that $\mathbb{E}_{\mathbf{u}}[\|\mathbf{u}\|^2] = d$ for $\mathbf{u} \sim \mathcal{N}(0, \mathbf{I}_d)$.

Furthermore, the second term in (30) can be bounded as:

$$\begin{aligned}
\mathbb{E}_{\mathbf{u}}\left[|F(\boldsymbol{\theta}+\mu\mathbf{u}) - F_\mu(\boldsymbol{\theta})|^2\|\mathbf{u}\|^2\right] &\overset{(a)}{\leq} 2\mathbb{E}_{\mathbf{u}}\left[\left(|F(\boldsymbol{\theta}+\mu\mathbf{u}) - F(\boldsymbol{\theta})|^2 + |F(\boldsymbol{\theta}) - \mathbb{E}_{\mathbf{u}'}[F(\boldsymbol{\theta}+\mu\mathbf{u}')]|^2\right)\|\mathbf{u}\|^2\right] \\
&\overset{(b)}{\leq} 2\mathbb{E}_{\mathbf{u}}\left[\left(|F(\boldsymbol{\theta}+\mu\mathbf{u}) - F(\boldsymbol{\theta})|^2 + \mathbb{E}_{\mathbf{u}'}\left[|F(\boldsymbol{\theta}) - F(\boldsymbol{\theta}+\mu\mathbf{u}')|^2\right]\right)\|\mathbf{u}\|^2\right] \\
&\overset{(c)}{\leq} 2L_0^2\mu^2\mathbb{E}_{\mathbf{u}}\left[\left(\|\mathbf{u}\|^2 + \mathbb{E}_{\mathbf{u}'}\left[\|\mathbf{u}'\|^2\right]\right)\|\mathbf{u}\|^2\right] \\
&\overset{(d)}{=} 2L_0^2\mu^2\mathbb{E}_{\mathbf{u}}\left[\left(\|\mathbf{u}\|^2 + d\right)\|\mathbf{u}\|^2\right] \overset{(e)}{=} 4L_0^2\mu^2 d(d+1) \,,
\end{aligned} \tag{31}$$

where $(a)$ follows from the fact that $(a+b)^2 \le 2(a^2+b^2)$, $(b)$ comes from Jensen's inequality, $(c)$ results from Lem. C.1, and $(d)$, $(e)$ results from the fact that $\mathbb{E}[\|\mathbf{u}\|^2] = d$ and $\mathbb{E}[\|\mathbf{u}\|^4] = d(d+2)$ for $\mathbf{u} \sim \mathcal{N}(0, \mathbf{I}_d)$.

Overall, substituting (31) into (30), we complete the proof:

$$\mathbb{E}_{\mathbf{u}}[\nu^2 \|\mathbf{u}\|^2] \le \frac{1}{\mu^4}\left(\sigma_\xi^2 d + 4L_0^2\mu^2 d(d+1)\right) = \frac{d}{\mu^4}\left(\sigma_\xi^2 + 4L_0^2\mu^2(d+1)\right). \tag{32}$$

Similarly, we have:

$$\mathbb{E}_{\mathbf{u}}[\nu^2 \|\mathbf{u}\|^4] \le \frac{d(d+2)}{\mu^4}\left(\sigma_\xi^2 + 4L_0^2\mu^2(d+2)\right). \tag{33}$$

$$\mathbb{E}_{\mathbf{u}}[\nu^2 \|\mathbf{u}\|^6] \le \frac{d(d+2)(d+4)}{\mu^4}\left(\sigma_\xi^2 + 4L_0^2\mu^2(d+3)\right). \tag{34}$$

$\square$

Below are some useful lemmas for $N > 1$ case, which can be proved similarly as above.

**Lemma C.5** (Bias of Gradient Estimator (35) for $N > 1$). *Let smoothed objective $F_\mu(\boldsymbol{\theta})$ be (7), and the gradient estimator $\hat{\nabla}F(\boldsymbol{\theta})$ be*

$$\hat{\nabla}F(\boldsymbol{\theta}_{t-1}) \triangleq \frac{1}{NK-1} \sum_{n,k=1}^{N,K} \frac{f(\boldsymbol{\theta}_{t-n} + \mu\mathbf{u}_{t-n,k}; \xi) - b}{\mu} \mathbf{u}_{t-n,k}, \tag{35}$$

*where the averaged baseline $b \triangleq \frac{1}{NK}\sum_{n,k=1}^{N,K} f(\boldsymbol{\theta}_{t-n} + \mu\mathbf{u}_{t-n,k}; \xi)$. We then have:*

$$\mathbb{E}[\hat{\nabla}F(\boldsymbol{\theta}_{t-1})] = \frac{1}{N}\sum_{n=1}^{N} \nabla F_\mu(\boldsymbol{\theta}_{t-n}). \tag{36}$$

*Proof.* The bias of the gradient estimator $\hat{\nabla}F(\boldsymbol{\theta})$ follows from Stein's identity:

$$\mathbb{E}[\hat{\nabla}F(\boldsymbol{\theta}_{t-1})] = \mathbb{E}\left[\frac{1}{NK-1}\sum_{n,k=1}^{N,K} \frac{f(\boldsymbol{\theta}_{t-n} + \mu\mathbf{u}_{t-n,k}; \xi) - \frac{1}{NK}\sum_{n,k=1}^{N,K} f(\boldsymbol{\theta}_{t-n} + \mu\mathbf{u}_{t-n,k}; \xi)}{\mu} \mathbf{u}_{t-n,k}\right]$$

$$= \mathbb{E}\left[\frac{1}{NK-1}\sum_{n,k=1}^{N,K} \frac{\frac{NK-1}{NK}f(\boldsymbol{\theta}_{t-n} + \mu\mathbf{u}_{t-n,k}; \xi) - \frac{1}{NK}\sum_{\substack{n',k'=1 \\ (n',k')\neq(n,k)}}^{N,K} f(\boldsymbol{\theta}_{t-n} + \mu\mathbf{u}_{t-n,k'}; \xi)}{\mu} \mathbf{u}_{t-n,k}\right]$$

$$= \mathbb{E}\left[\frac{1}{NK}\sum_{n,k=1}^{N,K} \frac{f(\boldsymbol{\theta}_{t-n} + \mu\mathbf{u}_{t-n,k}; \xi)}{\mu} \mathbf{u}_{t-n,k}\right]$$

$$\quad - \mathbb{E}\left[\frac{1}{NK(NK-1)}\sum_{n,k=1}^{N,K}\sum_{\substack{n',k'=1 \\ (n',k')\neq(n,k)}}^{N,K} \frac{f(\boldsymbol{\theta}_{t-n} + \mu\mathbf{u}_{t-n,k'}; \xi)}{\mu} \mathbf{u}_{t-n,k}\right]$$

$$\overset{(a)}{=} \mathbb{E}_{\mathbf{u}}\left[\frac{1}{N}\sum_{n=1}^{N} \frac{F(\boldsymbol{\theta}_{t-n} + \mu\mathbf{u}_{t-n})}{\mu} \mathbf{u}_{t-n}\right] \overset{(b)}{=} \frac{1}{N}\sum_{n=1}^{N} \nabla F_\mu(\boldsymbol{\theta}_{t-n}),$$

$$\tag{37}$$

where $(a)$ comes from the dependence of different random directions $\{\mathbf{u}_{t-n,k}\}_{k=1}^{K}$, and $(b)$ follows from Stein's identity.

$\square$

**Lemma C.6** (Bound on Second Moment of $\nu$ for $N > 1$)**.** *Let $\nu_{t-1} = (f(\boldsymbol{\theta}_{t-1} + \mu \mathbf{u}_{t-1}; \xi) - b)/\mu^2$, where $b$ is the averaged baseline in Thm. 4.7, and $\mathbf{u}_{t-1} \sim \mathcal{N}(0, \mathbf{I}_d)$. Under Assump. 4.3 and Lem. C.1, for $N > 1$ case, we have:*

$$\sum_{n=1}^{N} \mathbb{E}\left[\nu_{t-n}^2 \|\mathbf{u}_{t-n}\|^2\right] \leq \frac{\lambda^2 N(NK-1)^2 d}{\lambda^2 \mu^4 (NK-1)^2 - L_0^2 \eta^2 \mu^2 K(N^2-1)d/3}\left(\sigma_\xi^2 + 4L_0^2 \mu^2 (d+1)\right),$$

$$\sum_{n=1}^{N} \mathbb{E}_{\mathbf{u}}\left[\nu_{t-n}^2 \|\mathbf{u}_{t-n}\|^4\right] \leq \frac{\lambda^2 N(NK-1)^2 d(d+2)}{\lambda^2 \mu^4 (NK-1)^2 - L_0^2 \eta^2 \mu^2 K(N^2-1)d(d+2)/3}\left(\sigma_\xi^2 + 4L_0^2 \mu^2 (d+2)\right), \quad (38)$$

$$\sum_{n=1}^{N} \mathbb{E}_{\mathbf{u}}\left[\nu_{t-n}^2 \|\mathbf{u}_{t-n}\|^6\right] \leq \frac{\lambda^2 N(NK-1)^2 d(d+2)(d+4)}{\lambda^2 \mu^4 (NK-1)^2 - L_0^2 \eta^2 \mu^2 K(N^2-1)d(d+2)(d+4)/3}\left(\sigma_\xi^2 + 4L_0^2 \mu^2 (d+3)\right).$$

*Proof.* First, at the iteration $t - n$, we have:

$$\mathbb{E}_{\mathbf{u}}\left[\nu_{t-n}^2 \|\mathbf{u}_{t-n}\|^2\right] = \frac{1}{\mu^4} \mathbb{E}_{\mathbf{u}}\left[\left|f(\boldsymbol{\theta}_{t-n} + \mu \mathbf{u}_{t-n}; \xi) - \frac{1}{N} \sum_{n'=1}^{N} F_\mu(\boldsymbol{\theta}_{t-n'})\right|^2 \|\mathbf{u}_{t-n}\|^2\right]$$

$$\overset{(a)}{=} \frac{1}{\mu^4} \mathbb{E}_{\mathbf{u}}\left[\left(\left|f(\boldsymbol{\theta}_{t-n} + \mu \mathbf{u}_{t-n}; \xi) - F(\boldsymbol{\theta}_{t-n} + \mu \mathbf{u}_{t-n})\right|^2 + \left|F(\boldsymbol{\theta}_{t-n} + \mu \mathbf{u}_{t-n}) - \frac{1}{N} \sum_{n'=1}^{N} F_\mu(\boldsymbol{\theta}_{t-n'})\right|^2\right) \|\mathbf{u}_{t-n}\|^2\right]$$

$$\overset{(b)}{\leq} \frac{1}{\mu^4} \left(\sigma_\xi^2 \mathbb{E}_{\mathbf{u}}[\|\mathbf{u}_{t-n}\|^2] + \mathbb{E}_{\mathbf{u}}\left[\left|F(\boldsymbol{\theta}_{t-n} + \mu \mathbf{u}_{t-n}) - \frac{1}{N} \sum_{n'=1}^{N} F_\mu(\boldsymbol{\theta}_{t-n'})\right|^2 \|\mathbf{u}_{t-n}\|^2\right]\right)$$

$$\overset{(c)}{=} \frac{1}{\mu^4} \left(\sigma_\xi^2 d + \mathbb{E}_{\mathbf{u}}\left[\left|F(\boldsymbol{\theta}_{t-n} + \mu \mathbf{u}_{t-n}) - \frac{1}{N} \sum_{n'=1}^{N} F_\mu(\boldsymbol{\theta}_{t-n'})\right|^2 \|\mathbf{u}_{t-n}\|^2\right]\right),$$

$$(39)$$

where $(a)$ follows from (1), i.e. $\mathbb{E}_\xi[f(\boldsymbol{\theta} + \mu \mathbf{u}; \xi) - F(\boldsymbol{\theta} + \mu \mathbf{u})] = 0$, $(b)$ comes from Assump. 4.4, and $(c)$ results from the fact that $\mathbb{E}_{\mathbf{u}}[\|\mathbf{u}\|^2] = d$ for $\mathbf{u} \sim \mathcal{N}(0, \mathbf{I}_d)$.

Furthermore, the second term in (39) can be bounded as:

$$\mathbb{E}_{\mathbf{u}}\left[\left|F(\boldsymbol{\theta}_{t-n} + \mu \mathbf{u}_{t-n}) - \frac{1}{N} \sum_{n'=1}^{N} F_\mu(\boldsymbol{\theta}_{t-n'})\right|^2 \|\mathbf{u}_{t-n}\|^2\right]$$

$$\overset{(a)}{=} \mathbb{E}_{\mathbf{u}}\left[\left(\left|F(\boldsymbol{\theta}_{t-n} + \mu \mathbf{u}_{t-n}) - F_\mu(\boldsymbol{\theta}_{t-n})\right|^2 + \left|F_\mu(\boldsymbol{\theta}_{t-n}) - \frac{1}{N} \sum_{n'=1}^{N} F_\mu(\boldsymbol{\theta}_{t-n'})\right|^2\right) \|\mathbf{u}_{t-n}\|^2\right]$$

$$\overset{(b)}{\leq} 4L_0^2 \mu^2 d(d+1) + \left|F_\mu(\boldsymbol{\theta}_{t-n}) - \frac{1}{N} \sum_{n'=1}^{N} F_\mu(\boldsymbol{\theta}_{t-n'})\right|^2 \cdot \mathbb{E}_{\mathbf{u}}\left[\|\mathbf{u}_{t-n}\|^2\right]$$

$$(40)$$

$$= 4L_0^2 \mu^2 d(d+1) + \left|\frac{1}{N} \sum_{n'=1}^{N} (F_\mu(\boldsymbol{\theta}_{t-n}) - F_\mu(\boldsymbol{\theta}_{t-n'}))\right|^2 \cdot d$$

$$\overset{(c)}{\leq} 4L_0^2 \mu^2 d(d+1) + \frac{dL_0^2}{N} \sum_{n'=1}^{N} \|\boldsymbol{\theta}_{t-n} - \boldsymbol{\theta}_{t-n'}\|^2$$

$$\overset{(d)}{\leq} 4L_0^2 \mu^2 d(d+1) + \frac{dL_0^2 \eta^2 \mu^2 K C_\nu}{\lambda^2 N(NK-1)^2} \cdot \sum_{n'=1}^{N} |n - n'|,$$

where $(a)$ follows from the fact that $\mathbb{E}_{\mathbf{u}}[F(\boldsymbol{\theta} + \mu \mathbf{u})] = F_\mu(\boldsymbol{\theta})$, $(b)$ comes from (31), $(c)$ results from Lem. C.1, and $(d)$

results from:

$$
\|\boldsymbol{\theta}_{t-n} - \boldsymbol{\theta}_{t-n'}\|^2 \overset{(a)}{\leq} \sum_{n''=\min(n,n')}^{\max(n,n')} \|\boldsymbol{\theta}_{t-n''} - \boldsymbol{\theta}_{t-n''+1}\|^2 \leq \eta^2 \sum_{n''=\min(n,n')}^{\max(n,n')} \left\| \widetilde{H}^{-1}(\boldsymbol{\theta}_{t-n''})\hat{\nabla}F(\boldsymbol{\theta}_{t-n''}) \right\|^2
$$
$$
\overset{(b)}{\leq} \frac{\eta^2 \mu^2 K C_\nu}{\lambda^2 (NK-1)^2} |n - n'| ,
\tag{41}
$$

where $(a)$ follows from the triangle inequality, and $(b)$ follows from the result in (92), and we define the constant upper bound $C_\nu$ such that $\sum_n \mathbb{E}\left[\nu_{t-n}^2 \|\mathbf{u}_{t-n}\|^2\right] \leq C$ for all $n$.

Overall, substituting (40) into (39), and summing over $n$, we have:

$$
\sum_{n=1}^{N} \mathbb{E}_{\mathbf{u}}[\nu_{t-n}^2 \|\mathbf{u}_{t-n}\|^2] \leq \frac{1}{\mu^4}\left( \sigma_\xi^2 dN + 4L_0^2 \mu^2 d(d+1)N + \frac{dL_0^2 \eta^2 \mu^2 K C_\nu}{\lambda^2 N(NK-1)^2} \cdot \sum_{n,n'=1}^{N} |n-n'| \right)
$$
$$
= \frac{1}{\mu^4}\left( \sigma_\xi^2 dN + 4L_0^2 \mu^2 d(d+1)N + \frac{dL_0^2 \eta^2 \mu^2 K C_\nu (N^2-1)}{3\lambda^2(NK-1)^2} \right) .
\tag{42}
$$

Note that the definition of $C_\nu$ implies that:

$$
C_\nu = \frac{1}{\mu^4}\left( \sigma_\xi^2 dN + 4L_0^2 \mu^2 d(d+1)N + \frac{dL_0^2 \eta^2 \mu^2 K C_\nu (N^2-1)}{3\lambda^2(NK-1)^2} \right) ,
\tag{43}
$$

$$
C_\nu = \frac{1}{\mu^4}\left( \sigma_\xi^2 d(d+2)N + 4L_0^2 \mu^2 d(d+2)^2 N + \frac{d(d+2)L_0^2 \eta^2 \mu^2 K C_\nu (N^2-1)}{3\lambda^2(NK-1)^2} \right) ,
\tag{44}
$$

which leads to:

$$
\sum_{n=1}^{N} \mathbb{E}\left[\nu_{t-n}^2 \|\mathbf{u}_{t-n}\|^2\right] \leq \frac{\lambda^2 N(NK-1)^2 d}{\lambda^2 \mu^4 (NK-1)^2 - L_0^2 \eta^2 \mu^2 K(N^2-1)d/3}\left(\sigma_\xi^2 + 4L_0^2 \mu^2 (d+1)\right) .
\tag{45}
$$

Similarly, we have:

$$
\sum_{n=1}^{N} \mathbb{E}_{\mathbf{u}}\left[\nu_{t-n}^2 \|\mathbf{u}_{t-n}\|^4\right] \leq \frac{\lambda^2 N(NK-1)^2 d(d+2)}{\lambda^2 \mu^4 (NK-1)^2 - L_0^2 \eta^2 \mu^2 K(N^2-1)d(d+2)/3}\left(\sigma_\xi^2 + 4L_0^2 \mu^2 (d+2)\right) ,
\tag{46}
$$

$$
\sum_{n=1}^{N} \mathbb{E}_{\mathbf{u}}\left[\nu_{t-n}^2 \|\mathbf{u}_{t-n}\|^6\right] \leq \frac{\lambda^2 N(NK-1)^2 d(d+2)(d+4)}{\lambda^2 \mu^4 (NK-1)^2 - L_0^2 \eta^2 \mu^2 K(N^2-1)d(d+2)(d+4)/3}\left(\sigma_\xi^2 + 4L_0^2 \mu^2 (d+3)\right) .
\tag{47}
$$

$\square$

## C.2. Proof of Lem. 3.1

*Proof.* We first derive the first derivative of the single-step policy optimization objective (7) by applying the Policy Gradient Thm. (Sutton et al., 1999):
$$
\nabla F_\mu(\boldsymbol{\theta}) = \mathbb{E}_{\mathbf{x} \sim \pi_{\boldsymbol{\theta}}(\mathbf{x})}\left[\nabla \ln \pi_{\boldsymbol{\theta}}(\mathbf{x}) \mathbb{E}_\xi[f(\mathbf{x};\xi)]\right] .
\tag{48}
$$

Taking the second derivative, we have:

$$
\nabla^2 F_\mu(\boldsymbol{\theta}) = \nabla \int \pi_{\boldsymbol{\theta}}(\mathbf{x})\nabla \ln \pi_{\boldsymbol{\theta}}(\mathbf{x})\mathbb{E}_\xi[f(\mathbf{x};\xi)]d\mathbf{x} \overset{(a)}{=} \int \left(\nabla \pi_{\boldsymbol{\theta}}(\mathbf{x})(\nabla \ln \pi_{\boldsymbol{\theta}}(\mathbf{x}))^\top + \pi_{\boldsymbol{\theta}}(\mathbf{x})\nabla^2 \ln \pi_{\boldsymbol{\theta}}(\mathbf{x})\right)\mathbb{E}_\xi[f(\mathbf{x};\xi)]d\mathbf{x}
$$
$$
\overset{(b)}{=} \int \pi_{\boldsymbol{\theta}}(\mathbf{x})\left((\nabla \ln \pi_{\boldsymbol{\theta}}(\mathbf{x}))(\nabla \ln \pi_{\boldsymbol{\theta}}(\mathbf{x}))^\top + \nabla^2 \ln \pi_{\boldsymbol{\theta}}(\mathbf{x})\right)\mathbb{E}_\xi[f(\mathbf{x};\xi)]d\mathbf{x}
$$
$$
= \mathbb{E}_{\mathbf{x} \sim \pi_{\boldsymbol{\theta}}(\mathbf{x})}\left[\left((\nabla \ln \pi_{\boldsymbol{\theta}}(\mathbf{x}))(\nabla \ln \pi_{\boldsymbol{\theta}}(\mathbf{x}))^\top + \nabla^2 \ln \pi_{\boldsymbol{\theta}}(\mathbf{x})\right)\mathbb{E}_\xi[f(\mathbf{x};\xi)]\right] ,
$$
$$
\tag{49}
$$
where $(a)$ comes from the product rule of differentiation, and $(b)$ follows from the fact that $\nabla \pi_{\boldsymbol{\theta}}(\mathbf{x}) = \pi_{\boldsymbol{\theta}}(\mathbf{x})\nabla \ln \pi_{\boldsymbol{\theta}}(\mathbf{x})$. $\square$

### C.3. Proof of Prop. 3.2

*Proof.* Taking the reparametrization trick $\mathbf{x} = \boldsymbol{\theta} + \mu\mathbf{u}$ with $\mathbf{u} \sim \mathcal{N}(0, \mathbf{I}_d)$, we have:

$$\nabla^2 F_\mu(\boldsymbol{\theta}) \overset{(a)}{=} \mathbb{E}_\mathbf{u}\left[\left(\mathbf{u}\mathbf{u}^\top - \mathbf{I}_d\right)\frac{\mathbb{E}_\xi[f(\boldsymbol{\theta} + \mu\mathbf{u}; \xi)]}{\mu^2}\right] , \tag{50}$$

where $(a)$ comes from $\nabla \ln \pi_{\boldsymbol{\theta}}(\mathbf{x}) = \frac{1}{\mu^2}(\mathbf{x} - \boldsymbol{\theta}) = \frac{1}{\mu}\mathbf{u}$ and $\nabla^2 \ln \pi_{\boldsymbol{\theta}}(\mathbf{x}) = -\frac{1}{\mu^2}\mathbf{I}_d$ for Gaussian policy.

Since $\mathbb{E}_\mathbf{u}[\mathbf{u}\mathbf{u}^\top] = \mathbf{I}_d$, for any constant baseline $b$ that is independent of $\mathbf{u}$, (50) can be further transformed as:

$$\nabla^2 F_\mu(\boldsymbol{\theta}) = \mathbb{E}_\mathbf{u}\left[\left(\mathbf{u}\mathbf{u}^\top - \mathbf{I}_d\right)\frac{\mathbb{E}_\xi[f(\boldsymbol{\theta} + \mu\mathbf{u}; \xi)]}{\mu^2}\right] + b \cdot \mathbb{E}_\mathbf{u}\left[\left(\mathbf{u}\mathbf{u}^\top - \mathbf{I}_d\right)\right] = \mathbb{E}_\mathbf{u}\left[\left(\mathbf{u}\mathbf{u}^\top - \mathbf{I}_d\right)\frac{\mathbb{E}_\xi[f(\boldsymbol{\theta} + \mu\mathbf{u}; \xi) - b]}{\mu^2}\right] , \tag{51}$$

which completes the proof. $\qquad\square$

### C.4. Proof of Prop. 3.3

*Proof.* Regardless of the choice of baseline $b$, since $\mathbb{E}_\mathbf{u}[\mathbf{u}\mathbf{u}^\top] = \mathbf{I}_d$, Prop. 3.2 can always be transformed as:

$$
\begin{aligned}
\nabla^2 F_\mu(\boldsymbol{\theta}) =& \mathbb{E}_\mathbf{u}\left[\frac{f(\boldsymbol{\theta} + \mu\mathbf{u}; \xi) - b}{\mu^2}\mathbf{u}\mathbf{u}^\top\right] - \mathbb{E}_\mathbf{u}\left[\mathbb{E}_\mathbf{u}\left[\frac{f(\boldsymbol{\theta} + \mu\mathbf{u}; \xi) - b}{\mu^2}\right]\mathbf{u}\mathbf{u}^\top\right] \\
=& \mathbb{E}_\mathbf{u}\left[\frac{f(\boldsymbol{\theta} + \mu\mathbf{u}; \xi) - b - \mathbb{E}_\mathbf{u}[f(\boldsymbol{\theta} + \mu\mathbf{u}; \xi) - b]}{\mu^2}\mathbf{u}\mathbf{u}^\top\right] \\
=& \mathbb{E}_\mathbf{u}\left[\frac{f(\boldsymbol{\theta} + \mu\mathbf{u}; \xi) - \mathbb{E}_\mathbf{u}[f(\boldsymbol{\theta} + \mu\mathbf{u}; \xi)]}{\mu^2}\mathbf{u}\mathbf{u}^\top\right] ,
\end{aligned}
\tag{52}
$$

which completes the proof. $\qquad\square$

### C.5. Proof of Lem. 3.5

*Proof.* Suppose there exists another baseline $b' \neq \mathbb{E}_{\mathbf{u}\sim\mathcal{N}(0,\mathbf{I}_d)}\left[\mathbb{E}_\xi\left[f(\boldsymbol{\theta} + \mu\mathbf{u}; \xi)\right]\right]$ that also satisfies the unbiasedness:

$$\mathbb{E}_\mathbf{u}\left[\frac{\mathbb{E}_\xi\left[f(\boldsymbol{\theta} + \mu\mathbf{u}; \xi) - b'\right]}{\mu^2}\mathbf{u}\mathbf{u}^\top\right] = \nabla^2 F_\mu(\boldsymbol{\theta}) = \mathbb{E}_\mathbf{u}\left[\frac{\mathbb{E}_\xi\left[f(\boldsymbol{\theta} + \mu\mathbf{u}; \xi)\right]}{\mu^2}\left(\mathbf{u}\mathbf{u}^\top - \mathbf{I}_d\right)\right] . \tag{53}$$

Take the difference on both sides, we have:

$$
\begin{aligned}
0 =& \mathbb{E}_\mathbf{u}\left[\frac{\mathbb{E}_\xi\left[f(\boldsymbol{\theta} + \mu\mathbf{u}; \xi) - b'\right]}{\mu^2}\mathbf{u}\mathbf{u}^\top\right] - \mathbb{E}_\mathbf{u}\left[\frac{\mathbb{E}_\xi\left[f(\boldsymbol{\theta} + \mu\mathbf{u}; \xi)\right]}{\mu^2}\left(\mathbf{u}\mathbf{u}^\top - \mathbf{I}_d\right)\right] \\
=& \mathbb{E}_\mathbf{u}\left[-\frac{b'}{\mu^2}\mathbf{u}\mathbf{u}^\top\right] + \mathbb{E}_\mathbf{u}\left[\frac{\mathbb{E}_\xi\left[f(\boldsymbol{\theta} + \mu\mathbf{u}; \xi)\right]}{\mu^2}\mathbf{I}_d\right] \\
=& \frac{\mathbb{E}_\mathbf{u}\left[\mathbb{E}_\xi\left[f(\boldsymbol{\theta} + \mu\mathbf{u}; \xi)\right]\right] - b'}{\mu^2}\mathbf{I}_d ,
\end{aligned}
\tag{54}
$$

which implies that $b' = \mathbb{E}_\mathbf{u}\left[\mathbb{E}_\xi\left[f(\boldsymbol{\theta} + \mu\mathbf{u}; \xi)\right]\right]$, contradicting the assumption. Therefore, the average baseline is the unique choice that guarantees the unbiasedness of the Hessian estimator in Prop. 3.3. $\qquad\square$

### C.6. Proof of Prop. 3.7

*Proof.* We rewrite the smoothed objective using the importance weight $w(\mathbf{x}) = \pi_{\boldsymbol{\theta}}(\mathbf{x})/\rho(\mathbf{x})$. The Hessian is $\nabla^2 \int f(\mathbf{x})\frac{\pi_{\boldsymbol{\theta}}(\mathbf{x})}{\rho(\mathbf{x})}\rho(\mathbf{x})d\mathbf{x}$. Since $\rho$ is independent of $\boldsymbol{\theta}$, the derivatives apply solely to $\pi_{\boldsymbol{\theta}}$. Interchanging differentiation and integration yields $\mathbb{E}_{\mathbf{x}\sim\rho}[f(\mathbf{x})\frac{1}{\rho(\mathbf{x})}\nabla^2\pi_{\boldsymbol{\theta}}(\mathbf{x})]$. The result follows by expanding $\nabla^2\pi_{\boldsymbol{\theta}} = \pi_{\boldsymbol{\theta}}(\nabla^2 \ln \pi_{\boldsymbol{\theta}} + \nabla \ln \pi_{\boldsymbol{\theta}}\nabla \ln \pi_{\boldsymbol{\theta}}^\top)$.

We start from the Hessian of the smoothed objective in (50):

$$\nabla^2 F_\mu(\boldsymbol{\theta}) = \mathbb{E}_\mathbf{u}\left[\left(\mathbf{u}\mathbf{u}^\top - \mathbf{I}_d\right)\frac{\mathbb{E}_\xi[f(\boldsymbol{\theta} + \mu\mathbf{u}; \xi)]}{\mu^2}\right] = \int \left(\mathbf{u}\mathbf{u}^\top - \mathbf{I}_d\right)\frac{\mathbb{E}_\xi[f(\boldsymbol{\theta} + \mu\mathbf{u}; \xi)]}{\mu^2}\pi_{\boldsymbol{\theta}}(\boldsymbol{\theta} + \mu\mathbf{u})d\mathbf{u} , \tag{55}$$

where we rewrite the expectation over $\mathbf{u}$ as an integral over the Gaussian policy $\pi_{\boldsymbol{\theta}}(\mathbf{x})$.

Under the assumptions $\text{supp}(\pi_{\boldsymbol{\theta}}) \subseteq \text{supp}(\rho)$ that ensures the importance sampling ratio $\mathcal{W}_{\pi/\rho}(\mathbf{u}) \triangleq \frac{\pi_{\boldsymbol{\theta}}(\boldsymbol{\theta}+\mu\mathbf{u})}{\rho(\mathbf{u})}$ is well-defined, we can express the Hessian of smoothed objective as:

$$\nabla^2 F_\mu(\boldsymbol{\theta}) = \int \left(\mathbf{u}\mathbf{u}^\top - \mathbf{I}_d\right) \frac{\mathbb{E}_\xi[f(\boldsymbol{\theta}+\mu\mathbf{u};\xi)]}{\mu^2} \mathcal{W}_{\pi/\rho}(\mathbf{u})\rho(\mathbf{u})d\mathbf{u} = \mathbb{E}_{\mathbf{u}\sim\rho}\left[\left(\mathbf{u}\mathbf{u}^\top - \mathbf{I}_d\right) \frac{\mathbb{E}_\xi[f(\boldsymbol{\theta}+\mu\mathbf{u};\xi)]}{\mu^2} \mathcal{W}_{\pi/\rho}(\mathbf{u})\right] . \tag{56}$$

Finally, similar to the derivation in (50), we have:

$$\nabla^2 F_\mu(\boldsymbol{\theta}) = \mathbb{E}_{\mathbf{u}\sim\rho}\left[\frac{\mathbb{E}_\xi[f(\boldsymbol{\theta}+\mu\mathbf{u};\xi)] - \mathbb{E}_{\mathbf{u}\sim\rho}\left[\mathbb{E}_\xi[f(\boldsymbol{\theta}+\mu\mathbf{u};\xi)]\mathcal{W}_{\pi/\rho}(\mathbf{u})\right]}{\mu^2}\mathbf{u}\mathbf{u}^\top \cdot \mathcal{W}_{\pi/\rho}(\mathbf{u})\right] . \tag{57}$$

$\square$

## C.7. Proof of Lem. 4.5

*Proof.* The gradient of the smoothed objective $F_\mu(\boldsymbol{\theta})$ is a direct result of Stein's identity:

$$\nabla F_\mu(\boldsymbol{\theta}) = \mathbb{E}_{\mathbf{u}}\left[\nabla F(\boldsymbol{\theta}+\mu\mathbf{u})\right] = \frac{1}{\mu}\mathbb{E}_{\mathbf{u}}\left[\nabla_{\mathbf{u}}F(\boldsymbol{\theta}+\mu\mathbf{u})\right] \stackrel{(a)}{=} \frac{1}{\mu}\mathbb{E}_{\mathbf{u}}\left[F(\boldsymbol{\theta}+\mu\mathbf{u})\mathbf{u}\right] , \tag{58}$$

where $(a)$ follows from Stein's identity.

Then, the Hessian of the smoothed objective $F_\mu(\boldsymbol{\theta})$ can be derived by taking another derivative:

$$\nabla^2 F_\mu(\boldsymbol{\theta}) = \left(\frac{1}{\mu}\mathbb{E}_{\mathbf{u}}\left[(\nabla F(\boldsymbol{\theta}+\mu\mathbf{u}))\mathbf{u}\right]\right) = \frac{1}{\mu^2}\mathbb{E}_{\mathbf{u}}\left[(\nabla_{\mathbf{u}}F(\boldsymbol{\theta}+\mu\mathbf{u}))\mathbf{u}^\top\right] . \tag{59}$$

Note that $\nabla_{\mathbf{u}}(F(\boldsymbol{\theta}+\mu\mathbf{u})\mathbf{u}) = (\nabla_{\mathbf{u}}F(\boldsymbol{\theta}+\mu\mathbf{u}))\mathbf{u}^\top + F(\boldsymbol{\theta}+\mu\mathbf{u})\mathbf{I}_d$, we have:

$$\nabla^2 F_\mu(\boldsymbol{\theta}) = \frac{1}{\mu^2}\mathbb{E}_{\mathbf{u}}\left[\nabla_{\mathbf{u}}(F(\boldsymbol{\theta}+\mu\mathbf{u})\mathbf{u}) - F(\boldsymbol{\theta}+\mu\mathbf{u})\mathbf{I}_d\right] \stackrel{(a)}{=} \frac{1}{\mu^2}\mathbb{E}_{\mathbf{u}}\left[F(\boldsymbol{\theta}+\mu\mathbf{u})\left(\mathbf{u}\mathbf{u}^\top - \mathbf{I}_d\right)\right] , \tag{60}$$

where $(a)$ again follows from Stein's identity. $\square$

## C.8. Proof of Thm. 4.6

*Proof.* We first simplify the Hessian estimator in (9) by:

$$\begin{aligned}
\widehat{\mathbf{H}}(\boldsymbol{\theta}) &= \frac{1}{K-1}\sum_{k=1}^{K} \frac{f(\boldsymbol{\theta}+\mu\mathbf{u}_k;\xi) - \frac{1}{K}\sum_{k'=1}^{K}f(\boldsymbol{\theta}+\mu\mathbf{u}_{k'};\xi)}{\mu^2}\mathbf{u}_k\mathbf{u}_k^\top \\
&= \frac{1}{K-1}\sum_{k=1}^{K} \frac{\frac{K-1}{K}f(\boldsymbol{\theta}+\mu\mathbf{u}_k;\xi) - \frac{1}{K}\sum_{\substack{k'=1\\k'\neq k}}^{K}f(\boldsymbol{\theta}+\mu\mathbf{u}_{k'};\xi)}{\mu^2}\mathbf{u}_k\mathbf{u}_k^\top .
\end{aligned} \tag{61}$$

Take the expectation over $\{\mathbf{u}_k\}_{k=1}^{K}$ and $\xi$ on both sides, we have:

$$
\begin{aligned}
\mathbb{E}\left[\widehat{\mathbf{H}}(\boldsymbol{\theta})\right] &= \frac{1}{K-1} \sum_{k=1}^{K} \mathbb{E}_{\mathbf{u}} \mathbb{E}_{\xi}\left[\frac{\frac{K-1}{K} f(\boldsymbol{\theta}+\mu\mathbf{u}_k;\xi) - \frac{1}{K}\sum_{\substack{k'=1\\k'\neq k}}^{K} f(\boldsymbol{\theta}+\mu\mathbf{u}_{k'};\xi)}{\mu^2} \mathbf{u}_k\mathbf{u}_k^\top\right] \\
&\overset{(a)}{=} \frac{1}{K-1}\sum_{k=1}^{K}\mathbb{E}_{\mathbf{u}}\left[\frac{\frac{K-1}{K} F(\boldsymbol{\theta}+\mu\mathbf{u}_k) - \frac{1}{K}\sum_{\substack{k'=1\\k'\neq k}}^{K} F(\boldsymbol{\theta}+\mu\mathbf{u}_{k'})}{\mu^2}\mathbf{u}_k\mathbf{u}_k^\top\right] \\
&= \frac{1}{K-1}\sum_{k=1}^{K}\frac{\frac{K-1}{K}\mathbb{E}_{\mathbf{u}}\left[F(\boldsymbol{\theta}+\mu\mathbf{u})\mathbf{u}\mathbf{u}^\top\right] - \frac{1}{K}\sum_{\substack{k'=1\\k'\neq k}}^{K}\mathbb{E}_{\mathbf{u}}[F(\boldsymbol{\theta}+\mu\mathbf{u})\mathbf{I}_d]}{\mu^2} \\
&= \frac{1}{K-1}\sum_{k=1}^{K}\frac{\frac{K-1}{K}\mathbb{E}_{\mathbf{u}}\left[F(\boldsymbol{\theta}+\mu\mathbf{u})\mathbf{u}\mathbf{u}^\top\right] - \frac{K-1}{K}\mathbb{E}_{\mathbf{u}}[F(\boldsymbol{\theta}+\mu\mathbf{u})\mathbf{I}_d]}{\mu^2} \\
&= \mathbb{E}_{\mathbf{u}}\left[\frac{F(\boldsymbol{\theta}+\mu\mathbf{u})\left(\mathbf{u}\mathbf{u}^\top - \mathbf{I}_d\right)}{\mu^2}\right] \\
&\overset{(b)}{=} \nabla^2 F_\mu(\boldsymbol{\theta}) \,,
\end{aligned} \tag{62}
$$

where $(a)$ follows from the independence of $\frac{1}{K}\sum_{\substack{k'=1\\k'\neq k}}^{K} f(\boldsymbol{\theta}+\mu\mathbf{u}_{k'};\xi)$ and $\mathbf{u}_k\mathbf{u}_k^\top$, and $\mathbb{E}_{\mathbf{u}}[uu^\top] = \mathbf{I}_d$. Step $(b)$ follows from Lem. 4.5. $\qquad\square$

## C.9. Proof of Thm. 4.7

*Proof.* Since the baseline $b$ is a free parameter in the Hessian estimator, to ensure unbiasedness, we consider the Monte Carlo implementation of Prop. 3.2:

$$
\widehat{\mathbf{H}}(\boldsymbol{\theta}) = \frac{1}{K}\sum_{k=1}^{K}\frac{f(\boldsymbol{\theta}+\mu\mathbf{u}_k;\xi) - b}{\mu^2}\left(\mathbf{u}_k\mathbf{u}_k^\top - \mathbf{I}_d\right)\,, \tag{63}
$$

where the baseline $b$ is a constant independent of the random directions $\{\mathbf{u}_k\}_{k=1}^{K}$.

We verify that the Hessian estimator in (63) is unbiased:

$$
\mathbb{E}\left[\widehat{\mathbf{H}}(\boldsymbol{\theta})\right] \overset{(a)}{=} \mathbb{E}_{\mathbf{u}}\left[\frac{F(\boldsymbol{\theta}+\mu\mathbf{u}) - b}{\mu^2}\left(\mathbf{u}\mathbf{u}^\top - \mathbf{I}_d\right)\right] \overset{(b)}{=} \mathbb{E}_{\mathbf{u}}\left[\frac{F(\boldsymbol{\theta}+\mu\mathbf{u})}{\mu^2}\left(\mathbf{u}\mathbf{u}^\top - \mathbf{I}_d\right)\right] \overset{(c)}{=} \nabla^2 F_\mu(\boldsymbol{\theta})\,, \tag{64}
$$

where $(a)$ follows from the independence of $\{\mathbf{u}_k\}$ across different queries $k$, $(b)$ uses the property $\mathbb{E}_{\mathbf{u}}[\mathbf{u}\mathbf{u}^\top] = \mathbf{I}_d$, and $(c)$ follows from Stein's identity.

Next, the variance of the Hessian estimator in (63) is given by:

$$
\begin{aligned}
\mathrm{Var}\left(\widehat{\mathbf{H}}(\boldsymbol{\theta})\right) &= \mathbb{E}\left[\left\|\widehat{\mathbf{H}}(\boldsymbol{\theta}) - \nabla^2 F_\mu(\boldsymbol{\theta})\right\|_F^2\right] = \mathbb{E}\left[\left\|\widehat{\mathbf{H}}(\boldsymbol{\theta})\right\|_F^2\right] - 2\,\mathrm{Tr}\left(\mathbb{E}\left[\widehat{\mathbf{H}}(\boldsymbol{\theta})\right]^\top \nabla^2 F_\mu(\boldsymbol{\theta})\right) + \left\|\nabla^2 F_\mu(\boldsymbol{\theta})\right\|_F^2 \\
&\overset{(a)}{=} \mathbb{E}\left[\left\|\frac{1}{K}\sum_{k=1}^{K}\frac{f(\boldsymbol{\theta}+\mu\mathbf{u}_k;\xi) - b}{\mu^2}\left(\mathbf{u}_k\mathbf{u}_k^\top - \mathbf{I}_d\right)\right\|_F^2\right] - \left\|\nabla^2 F_\mu(\boldsymbol{\theta})\right\|_F^2\,,
\end{aligned} \tag{65}
$$

where $(a)$ follows from (64).

Differentiating the variance with respect to the baseline $b$ yields:

$$
\begin{aligned}
\frac{\partial}{\partial b} \mathrm{Var}\left(\widehat{\mathbf{H}}(\boldsymbol{\theta})\right) =& 2\mathbb{E}\left[\left\langle \frac{1}{K}\sum_{k=1}^{K} \frac{f(\boldsymbol{\theta}+\mu\mathbf{u}_k;\xi)-b}{\mu^2}\left(\mathbf{u}_k\mathbf{u}_k^\top - \mathbf{I}_d\right), \frac{1}{K}\sum_{k=1}^{K}\left(-\frac{1}{\mu^2}\right)\left(\mathbf{u}_k\mathbf{u}_k^\top - \mathbf{I}_d\right)\right\rangle_F\right] \\
&\overset{(a)}{=} -\frac{2}{K^2\mu^4}\sum_{k=1}^{K}\mathbb{E}\left[\left\langle (f(\boldsymbol{\theta}+\mu\mathbf{u}_k;\xi)-b)\left(\mathbf{u}_k\mathbf{u}_k^\top - \mathbf{I}_d\right),\left(\mathbf{u}_k\mathbf{u}_k^\top - \mathbf{I}_d\right)\right\rangle_F\right] \\
&= -\frac{2}{K^2\mu^4}\sum_{k=1}^{K}\mathbb{E}_{\mathbf{u}}\left[\left\langle (F(\boldsymbol{\theta}+\mu\mathbf{u}_k)-b)\left(\mathbf{u}_k\mathbf{u}_k^\top - \mathbf{I}_d\right),\left(\mathbf{u}_k\mathbf{u}_k^\top - \mathbf{I}_d\right)\right\rangle_F\right] \\
&= -\frac{2}{K^2\mu^4}\sum_{k=1}^{K}\mathbb{E}_{\mathbf{u}}\left[(F(\boldsymbol{\theta}+\mu\mathbf{u}_k)-b)\left\|\left(\mathbf{u}_k\mathbf{u}_k^\top - \mathbf{I}_d\right)\right\|_F^2\right],
\end{aligned}
\tag{66}
$$

where $(a)$ follows from the independence of $\{\mathbf{u}_k\}$ across different queries $k$.

Setting $\frac{\partial}{\partial b}\mathrm{Var}\left(\widehat{\mathbf{H}}(\boldsymbol{\theta})\right) = 0$, we have:

$$
b^* = \frac{\sum_{k=1}^{K}\mathbb{E}_{\mathbf{u}}\left[F(\boldsymbol{\theta}+\mu\mathbf{u}_k)\left\|\left(\mathbf{u}_k\mathbf{u}_k^\top - \mathbf{I}_d\right)\right\|_F^2\right]}{\sum_{k=1}^{K}\mathbb{E}_{\mathbf{u}}\left[\left\|\left(\mathbf{u}_k\mathbf{u}_k^\top - \mathbf{I}_d\right)\right\|_F^2\right]} = \frac{\mathbb{E}_{\mathbf{u}}\left[F(\boldsymbol{\theta}+\mu\mathbf{u})\left\|\left(\mathbf{u}\mathbf{u}^\top - \mathbf{I}_d\right)\right\|_F^2\right]}{\mathbb{E}_{\mathbf{u}}\left[\left\|\left(\mathbf{u}\mathbf{u}^\top - \mathbf{I}_d\right)\right\|_F^2\right]}.
\tag{67}
$$

In particular, when $\mathbf{u} \sim \mathcal{N}(0,\mathbf{I}_d)$ as $d \to \infty$, such that $\left\|\left(\mathbf{u}_k\mathbf{u}_k^\top - \mathbf{I}_d\right)\right\|_F^2$ is effectively constant, the optimal baseline reduces to the expected function value:

$$
b^* = \mathbb{E}_{\mathbf{u}}\left[F(\boldsymbol{\theta}+\mu\mathbf{u})\right] = F_\mu(\boldsymbol{\theta}).
\tag{68}
$$

Finally, following the discussion in Appx. D.1, which establishes that Prop. 3.2 with an averaged baseline is equivalent to Def. 4.1, we conclude that the Hessian estimator in Def. 4.1 also attains the minimum variance among all constant baselines. $\qquad\square$

### C.10. Proof of Thm. 4.8

Before proving Thm. 4.8, we first present the variance analysis of the Hessian estimator with the optimal baseline in Thm. 4.7:

**Lemma C.7** (Variance of Hessian Estimator). *Consider the Hessian estimator utilizing the optimal baseline derived in Thm. 4.7. Let $\mathbf{u} \sim \mathcal{N}(\mathbf{0},\mathbf{I}_d)$. Under Assump. 4.3 and 4.4, the variance of the estimator is bounded by:*

$$
\mathbb{E}\left[\left\|\widehat{\mathbf{H}}(\boldsymbol{\theta}) - \nabla^2 F_\mu(\boldsymbol{\theta})\right\|_F^2\right] \le \frac{Kd(d+2)V}{(K-1)^2\mu^4},
$$

*where $V \triangleq \sigma_\xi^2 + 4L_0^2\mu^2(d+2)$.*

*Proof.* We adopt the decomposition of the variance of the Hessian estimator from (65), and let $b$ be the averaged baseline in

Thm. 4.7. Then we have:

$$
\begin{aligned}
\mathrm{Var}\left(\widehat{\mathbf{H}}(\boldsymbol{\theta})\right) =&\mathbb{E}\left[\left\|\frac{1}{K-1}\sum_{k=1}^{K}\frac{f(\boldsymbol{\theta}+\mu\mathbf{u}_k;\xi)-b}{\mu^2}\mathbf{u}_k\mathbf{u}_k^\top\right\|_F^2\right]-\left\|\nabla^2 F_\mu(\boldsymbol{\theta})\right\|_F^2\\
\leq&\mathbb{E}\left[\left\|\frac{1}{K-1}\sum_{k=1}^{K}\frac{f(\boldsymbol{\theta}+\mu\mathbf{u}_k;\xi)-b}{\mu^2}\mathbf{u}_k\mathbf{u}_k^\top\right\|_F^2\right]\\
\overset{(a)}{=}&\frac{1}{(K-1)^2\mu^4}\sum_{k=1}^{K}\mathbb{E}\left[|f(\boldsymbol{\theta}+\mu\mathbf{u}_k;\xi)-F_\mu(\boldsymbol{\theta})|^2\left\|\mathbf{u}_k\mathbf{u}_k^\top\right\|_F^2\right]\\
=&\frac{K}{(K-1)^2\mu^4}\mathbb{E}\left[|f(\boldsymbol{\theta}+\mu\mathbf{u};\xi)-F_\mu(\boldsymbol{\theta})|^2\|\mathbf{u}\|^4\right]\\
\overset{(b)}{\leq}&\frac{Kd(d+2)V}{(K-1)^2\mu^4}\ ,
\end{aligned}
\tag{69}
$$

where $(a)$ follows from the independence of $\{\mathbf{u}_k\}$ across different queries $k$, $(b)$ comes from Lem. C.4. Here, we denote $V \triangleq \sigma_\xi^2 + 4L_0^2\mu^2(d+2)$ for simplicity. $\qquad\square$

Now we are ready to prove Thm. 4.8.

*Proof.* The total error is bounded as:

$$
\mathbb{E}\left[\left\|\widehat{\mathbf{H}}(\boldsymbol{\theta})-\nabla^2 F(\boldsymbol{\theta})\right\|_F^2\right] \overset{(a)}{=} \mathbb{E}\left[\left\|\widehat{\mathbf{H}}(\boldsymbol{\theta})-\nabla^2 F_\mu(\boldsymbol{\theta})\right\|_F^2\right] + \left\|\nabla F_\mu(\boldsymbol{\theta})-\nabla F(\boldsymbol{\theta})\right\|_F^2 \overset{(b)}{\leq} \frac{Kd(d+2)V}{(K-1)^2\mu^4} + L_2^2\mu^2 d\ , \tag{70}
$$

where $(a)$ follows $\mathbb{E}\left[\widehat{\mathbf{H}}(\boldsymbol{\theta})-\nabla^2 F_\mu(\boldsymbol{\theta})\right] = 0$ from Thm. 4.6, and $(b)$ comes from Thm. C.7 and Lem. C.3. Here we denote $V \triangleq \sigma_\xi^2 + 4L_0^2\mu^2(d+2)$ in the last step for simplicity. $\qquad\square$

## C.11. Derivation of Def. 5.1

*Proof.* The Hessian estimator in Def. 4.1 can be rewritten as:

$$
\widehat{\mathbf{H}}(\boldsymbol{\theta}) = \frac{1}{K-1}\sum_{k=1}^{K}\frac{f(\boldsymbol{\theta}+\mu\mathbf{u}_k;\xi)-\frac{1}{K}\sum_{k'=1}^{K}f(\boldsymbol{\theta}+\mu\mathbf{u}_{k'};\xi)}{\mu^2}\mathbf{u}_k\mathbf{u}_k^\top \cdot \mathcal{W}(\boldsymbol{\theta}+\mu\mathbf{u}_k) = UDU^\top\ , \tag{71}
$$

where we denote $\mathbf{D} = \mathrm{diag}(\nu_1/(K-1),\dots,\nu_K/(K-1)) \in \mathbb{R}^{K\times K}$ with $\nu_k = \frac{f(\boldsymbol{\theta}+\mu\mathbf{u}_k;\xi)-\frac{1}{K}\sum_{k'=1}^{K}f(\boldsymbol{\theta}+\mu\mathbf{u}_{k'};\xi)}{\mu^2}\mathcal{W}(\boldsymbol{\theta}+\mu\mathbf{u}_k)$, and $\mathbf{U} = [\mathbf{u}_1,\mathbf{u}_2,\dots,\mathbf{u}_K] \in \mathbb{R}^{d\times K}$.

According to the Woodbury matrix identity (Woodbury, 1950), we have:

$$
\left(\widehat{\mathbf{H}}(\boldsymbol{\theta})+\lambda\mathbf{I}_d\right)^{-1} = \left(\lambda\mathbf{I}_d + \mathbf{U}\mathbf{D}\mathbf{U}^\top\right)^{-1} = \frac{1}{\lambda}\mathbf{I}_d - \frac{1}{\lambda^2}\mathbf{U}\left(\mathbf{D}^{-1}+\frac{1}{\lambda}\mathbf{U}^\top\mathbf{U}\right)^{-1}\mathbf{U}^\top\ . \tag{72}
$$

If we consider the diagonal approximation of $\mathbf{U}$ (i.e. assuming $\{\mathbf{u}_k\}_{k=1}^{K}$ are orthogonal), then $\mathbf{U}^\top\mathbf{U}$ is a diagonal matrix with the $k$-th diagonal element being $\|\mathbf{u}_k\|^2$. The inverse term can be calculated as:

$$
\mathbf{D}^{-1}+\frac{1}{\lambda}\mathbf{U}^\top\mathbf{U} = \mathrm{diag}\left(\frac{K-1}{\nu_k}+\frac{1}{\lambda}\|\mathbf{u}_k\|^2\right)_{k=1}^{K}\ . \tag{73}
$$

Therefore, the approximated inverse Hessian $\widetilde{\mathbf{H}}^{-1}(\boldsymbol{\theta})$ is:

$$
\widetilde{\mathbf{H}}^{-1}(\boldsymbol{\theta}) = \frac{1}{\lambda}\mathbf{I}_d - \frac{1}{\lambda^2}\sum_{k=1}^{K}\frac{\mathbf{u}_k\mathbf{u}_k^\top}{\frac{K-1}{\nu_k}+\frac{1}{\lambda}\|\mathbf{u}_k\|^2} = \frac{1}{\lambda}\mathbf{I}_d - \sum_{k=1}^{K}\frac{\nu_k}{\lambda\left(\lambda(K-1)+\nu_k\|\mathbf{u}_k\|^2\right)}\mathbf{u}_k\mathbf{u}_k^\top\ . \tag{74}
$$

$\qquad\square$

## C.12. Derivation of Def. 5.3

*Proof.* Adopt from inverse Hessian estimator in (15), we have:

$$\widetilde{\mathbf{H}}^{-1}(\boldsymbol{\theta})\hat{\nabla}F(\boldsymbol{\theta}) = \left(\frac{1}{\lambda}\mathbf{I}_d - \sum_{k=1}^{K}\frac{\mu\nu_k}{\lambda\left(\lambda(K-1)+\nu_k\|\mathbf{u}_k\|^2\right)}\mathbf{u}_k\mathbf{u}_k^\top\right)\hat{\nabla}F \ . \tag{75}$$

The first term is simple:

$$\frac{1}{\lambda}\mathbf{I}_d\hat{\nabla}F(\boldsymbol{\theta}) = \frac{1}{\lambda(K-1)}\sum_{k=1}^{K}\mu\nu_k\mathbf{u}_k \ . \tag{76}$$

While for the second term, to reduce the bias introduced by the correlation between the same random directions used in Hessian and gradient estimation, we remove the random direction that is used in the Hessian estimation from the gradient estimation, i.e., we have:

$$\left(\sum_{k=1}^{K}\frac{\mu\nu_k}{\lambda\left(\lambda(K-1)+\nu_k\|\mathbf{u}_k\|^2\right)}\mathbf{u}_k\mathbf{u}_k^\top\right)\hat{\nabla}F(\boldsymbol{\theta}) = \left(\sum_{k=1}^{K}\frac{\mu\nu_k}{\lambda\left(\lambda(K-1)+\nu_k\|\mathbf{u}_k\|^2\right)}\mathbf{u}_k\mathbf{u}_k^\top\right)\left(\frac{1}{K-2}\sum_{\substack{k'=1\\k'\neq k}}^{K}\mu\nu_{k'}\mathbf{u}_{k'}\right)$$

$$=\frac{1}{K-2}\sum_{k=1}^{K}\sum_{\substack{k'=1\\k'\neq k}}^{K}\frac{\mu\nu_k\mu\nu_{k'}}{\lambda\left(\lambda(K-1)+\nu_k\|\mathbf{u}_k\|^2\right)}\mathbf{u}_k\left(\mathbf{u}_k^\top\mathbf{u}_{k'}\right) = \sum_{k=1}^{K}\mu\nu_k\frac{\mathbf{u}_k^\top\left(\sum_{k'=1,k'\neq k}^{K}\nu_{k'}\mathbf{u}_{k'}\right)/(K-2)}{\lambda\left(\lambda(K-1)+\nu_k\|\mathbf{u}_k\|^2\right)}\mathbf{u}_k \ . \tag{77}$$

Combining the two terms together, we have:

$$\widetilde{\mathbf{H}}^{-1}(\boldsymbol{\theta})\hat{\nabla}F(\boldsymbol{\theta}) = \sum_{k=1}^{K}\mu\nu_k\left(\frac{1}{\lambda(K-1)} - \frac{\mathbf{u}_k^\top\left(\sum_{k'=1,k'\neq k}^{K}\nu_{k'}\mathbf{u}_{k'}\right)/(K-2)}{\lambda\left(\lambda(K-1)+\nu_k\|\mathbf{u}_k\|^2\right)}\right)\mathbf{u}_k \ . \tag{78}$$

$\square$

## C.13. Proof of Thm. 5.2

*Proof.* We first use Resolvent identity to decompose the error between two inverse Hessians into:

$$\mathbb{E}\left[\left\|\widetilde{\mathbf{H}}^{-1}(\boldsymbol{\theta}) - \left(\nabla^2 F_\mu(\boldsymbol{\theta}) + \lambda\mathbf{I}_d\right)^{-1}\right\|_F^2\right]$$

$$\overset{(a)}{=}\mathbb{E}\left[\left\|\widetilde{\mathbf{H}}^{-1}(\boldsymbol{\theta})\left(\nabla^2 F_\mu(\boldsymbol{\theta}) + \lambda\mathbf{I}_d - \widetilde{\mathbf{H}}(\boldsymbol{\theta})\right)\left(\nabla^2 F_\mu(\boldsymbol{\theta}) + \lambda\mathbf{I}_d\right)^{-1}\right\|_F^2\right] \tag{79}$$

$$\overset{(b)}{\leq}\mathbb{E}\left[\left\|\widetilde{\mathbf{H}}^{-1}(\boldsymbol{\theta})\right\|_2^2\left\|\nabla^2 F_\mu(\boldsymbol{\theta}) + \lambda\mathbf{I}_d - \widetilde{\mathbf{H}}(\boldsymbol{\theta})\right\|_F^2\right]\left\|(\nabla^2 F_\mu(\boldsymbol{\theta}) + \lambda\mathbf{I}_d)^{-1}\right\|_2^2 \ ,$$

where $(a)$ follows from the Resolvent identity $\mathbf{A}^{-1} - \mathbf{B}^{-1} = \mathbf{A}^{-1}(\mathbf{B} - \mathbf{A})\mathbf{B}^{-1}$ for any invertible matrices $\mathbf{A}$ and $\mathbf{B}$, and $(b)$ follows from the Cauchy-Schwarz inequality.

We next bound these three terms in (79) respectively. For the first term, we have:

$$\left\|\widetilde{\mathbf{H}}^{-1}(\boldsymbol{\theta})\right\|_2^2 = \left\|\frac{1}{\lambda}\mathbf{I}_d - \sum_{k=1}^{K}\frac{\nu_k}{\lambda\left(\lambda(K-1)+\nu_k\|\mathbf{u}_k\|^2\right)}\mathbf{u}_k\mathbf{u}_k^\top\right\|_2^2 \ . \tag{80}$$

Note that the summation $\sum_{k=1}^{K}\frac{\nu_k}{\lambda\left(\lambda(K-1)+\nu_k\|\mathbf{u}_k\|^2\right)}\mathbf{u}_k\mathbf{u}_k^\top$ is a rank-$K$ matrix. Under the orthogonality assumption of

random directions $\{u_k\}_{k=1}^K$, the eigenvalues of $\widetilde{\mathbf{H}}^{-1}(\boldsymbol{\theta})$ related to the subspace spanned by $\{\mathbf{u}_k\}_{k=1}^K$ can be computed as:

$$\lambda_k = \frac{1}{\lambda} - \frac{\nu_k\|\mathbf{u}_k\|^2}{\lambda\left(\lambda(K-1) + \nu_k\|\mathbf{u}_k\|^2\right)} = \frac{K-1}{\lambda(K-1) + \nu_k\|\mathbf{u}_k\|^2} = \frac{1}{\lambda(1+t_k)} \,, \tag{81}$$

where we denote $t_k \triangleq \nu_k\|\mathbf{u}_k\|^2/(\lambda(K-1))$ for simplicity.

Meanwhile, in the orthogonal complement subspace, the eigenvalues are all $1/\lambda$. Therefore, we have:

$$\left\|\widetilde{\mathbf{H}}^{-1}(\boldsymbol{\theta})\right\|_2^2 = \max\left\{\frac{1}{\lambda^2}, \max_{k=1,\ldots,K} \frac{1}{\lambda^2(1+t_k)^2}\right\} = \frac{1}{\lambda^2}\max\left\{1, \max_{k=1,\ldots,K} \frac{1}{(1+t_k)^2}\right\} \,. \tag{82}$$

To avoid singularity, we assume the existence of a positive constant $\rho \in (0, 1]$ such that $|t_k + 1| \geq \rho$ for all $k$. There are two cases to consider:

- For case $t_k \geq 0$ or $t_k \leq -2$, we have $1/(1 + t_k)^2 \leq 1$, thus $\left\|\widetilde{\mathbf{H}}^{-1}(\boldsymbol{\theta})\right\|_2^2 = 1/\lambda^2$.

- For case $\rho - 1 \leq t_k < 0$ or $-2 < t_k \leq -1 - \rho$, we have $1/(1 + t_k)^2 \geq 1$, thus:

$$\left\|\widetilde{\mathbf{H}}^{-1}(\boldsymbol{\theta})\right\|_2^2 = \max_{k=1,\ldots,K} \frac{1}{\lambda^2(1+t_k)^2} \leq \frac{1}{\lambda^2\rho^2} \,. \tag{83}$$

Overall, since $\rho \in (0, 1]$, we have:

$$\left\|\widetilde{\mathbf{H}}^{-1}(\boldsymbol{\theta})\right\|_2^2 \leq \frac{1}{\lambda^2\rho^2} \tag{84}$$

Then, the second term in (79) can be bounded by Thm. 4.8:

$$\mathbb{E}\left[\left\|\nabla^2 F_\mu(\boldsymbol{\theta}) + \lambda\mathbf{I}_d - \widetilde{\mathbf{H}}(\boldsymbol{\theta})\right\|_F^2\right] \leq \frac{Kd(d+2)V}{(K-1)^2\mu^4} + L_2^2\mu^2 d \tag{85}$$

For the third term in (79):

$$\left\|(\nabla^2 F(\boldsymbol{\theta}) + \lambda\mathbf{I}_d)^{-1}\right\|_2^2 \leq \left\|(\nabla^2 F(\boldsymbol{\theta}) + \lambda\mathbf{I}_d)^{-1}\right\|_F^2 = \sum_{i=1}^d \frac{1}{(\sigma_i + \lambda)^2} \leq \frac{d}{(\sigma_{\min} + \lambda)^2} \,, \tag{86}$$

where $\sigma_i$ denotes the $i$-th eigenvalue of $\nabla^2 F(\boldsymbol{\theta})$, and $\sigma_{\min}$ denotes the smallest eigenvalue respectively.

Overall, we have:

$$\mathbb{E}\left[\left\|\widetilde{\mathbf{H}}^{-1}(\boldsymbol{\theta}) - \left(\nabla^2 F_\mu(\boldsymbol{\theta}) + \lambda\mathbf{I}_d\right)^{-1}\right\|_F^2\right] \leq \frac{d}{\lambda^2\rho^2(\sigma_{\min} + \lambda)^2}\left(\frac{Kd(d+2)V}{(K-1)^2\mu^4} + L_2^2\mu^2 d\right) \,. \tag{87}$$

$\square$

### C.14. Proof of Thm. 5.4

**Theorem C.8** (Bias of Inverse Hessian-Gradient Product Estimator (Formal)). *Assume that the objective function is bounded, i.e. there exists a positive constant $G$ such that $f(\boldsymbol{\theta}; \xi) \leq G$ for all $\boldsymbol{\theta} \in \mathbb{R}^d$ and $\xi$. Under Assump. 4.3, 4.4, and the orthogonality of random directions $\{\mathbf{u}_k\}_{k=1}^K$, the bias of the inverse Hessian-gradient product approximation in (16) with respect to the true inverse Hessian-gradient product is bounded by:*

$$\mathbb{E}\left[\left\|\widetilde{\mathbf{H}}^{-1}(\boldsymbol{\theta})\hat{\nabla}F(\boldsymbol{\theta}) - \left(\nabla^2 F(\boldsymbol{\theta}) + \lambda\mathbf{I}_d\right)^{-1}\nabla F(\boldsymbol{\theta})\right\|^2\right] \leq \frac{Kd^2\left(B_1\sigma_\xi^2 + 4B_2L_0^2\mu^2\right)}{\lambda^2(\sigma_{\min} + \lambda)^2(K-1)^2} \,, \tag{88}$$

*where $B_1 \triangleq \frac{B}{\mu^6} + L_2^2 d$, and $B_2 \triangleq \frac{B}{\mu^6}(d+3) + L_2^2 d(d+1)$, and $B \triangleq \frac{4KG^2}{(K-1)^2\mu^6}(d+2)(d+4)$. Besides, $\sigma_{\min}$ denotes the minimum eigenvalue of the true Hessian $\nabla^2 F(\boldsymbol{\theta})$.*

*Proof.* We first decompose the error of the inverse Hessian-gradient product approximation into:

$$\mathbb{E}\left[\left\|\widetilde{\mathbf{H}}^{-1}(\boldsymbol{\theta})\hat{\nabla}F(\boldsymbol{\theta}) - \left(\nabla^2 F(\boldsymbol{\theta}) + \lambda\mathbf{I}_d\right)^{-1}\nabla F(\boldsymbol{\theta})\right\|^2\right]$$

$$\overset{(a)}{\leq} 2\,\underbrace{\mathbb{E}\left[\left\|\widetilde{\mathbf{H}}^{-1}(\boldsymbol{\theta})\hat{\nabla}F(\boldsymbol{\theta}) - \left(\nabla^2 F(\boldsymbol{\theta}) + \lambda\mathbf{I}_d\right)^{-1}\hat{\nabla}F(\boldsymbol{\theta})\right\|^2\right]}_{\text{①}}$$

$$+ 2\,\underbrace{\mathbb{E}\left[\left\|\left(\nabla^2 F(\boldsymbol{\theta}) + \lambda\mathbf{I}_d\right)^{-1}\hat{\nabla}F(\boldsymbol{\theta}) - \left(\nabla^2 F(\boldsymbol{\theta}) + \lambda\mathbf{I}_d\right)^{-1}\nabla F(\boldsymbol{\theta})\right\|^2\right]}_{\text{②}}, \tag{89}$$

where $(a)$ follows from the inequality $(a+b)^2 \leq 2(a^2 + b^2)$.

For the first term in (89), we have:

$$\mathbb{E}\left[\left\|\widetilde{\mathbf{H}}^{-1}(\boldsymbol{\theta})\hat{\nabla}F(\boldsymbol{\theta}) - \left(\nabla^2 F(\boldsymbol{\theta}) + \lambda\mathbf{I}_d\right)^{-1}\hat{\nabla}F(\boldsymbol{\theta})\right\|^2\right]$$

$$\overset{(a)}{=} \mathbb{E}\left[\left\|\left(\nabla^2 F(\boldsymbol{\theta}) + \lambda\mathbf{I}_d\right)^{-1}\left(\widetilde{\mathbf{H}}(\boldsymbol{\theta}) - \nabla^2 F(\boldsymbol{\theta}) - \lambda\mathbf{I}_d\right)\widetilde{\mathbf{H}}^{-1}(\boldsymbol{\theta})\hat{\nabla}F(\boldsymbol{\theta})\right\|^2\right] \tag{90}$$

$$\overset{(b)}{\leq} \mathbb{E}\left[\left\|\widetilde{\mathbf{H}}^{-1}(\boldsymbol{\theta})\hat{\nabla}F(\boldsymbol{\theta})\right\|^2\left\|\widetilde{\mathbf{H}}(\boldsymbol{\theta}) - \nabla^2 F(\boldsymbol{\theta}) - \lambda\mathbf{I}_d\right\|_F^2\right]\left\|\left(\nabla^2 F(\boldsymbol{\theta}) + \lambda\mathbf{I}_d\right)^{-1}\right\|_2^2,$$

where $(a)$ follows from the Resolvent identity $\mathbf{A}^{-1} - \mathbf{B}^{-1} = \mathbf{B}^{-1}(\mathbf{B} - \mathbf{A})\mathbf{A}^{-1}$ for any invertible matrices $\mathbf{A}$ and $\mathbf{B}$, and $(b)$ follows from $\|\mathbf{A}\mathbf{v}\|^2 \leq \|\mathbf{A}\|_2^2\|\mathbf{v}\|^2$, $\|\mathbf{A}\mathbf{B}\|_2 \leq \|\mathbf{A}\|_2\|\mathbf{B}\|_2$, and $\|\mathbf{A}\|_2 \leq \|\mathbf{A}\|_F$ for any matrices $\mathbf{A}, \mathbf{B}$ and vector $\mathbf{v}$.

We next bound the three terms in (90) respectively. For the first term, we have:

$$\left\|\widetilde{\mathbf{H}}^{-1}(\boldsymbol{\theta})\hat{\nabla}F(\boldsymbol{\theta})\right\|^2 = \left\|\sum_{k=1}^{K}\mu\nu_k\left(\frac{1}{\lambda(K-1)} - \frac{\mathbf{u}_k^\top\left(\sum_{k'=1,k'\neq k}^{K}\nu_{k'}\mathbf{u}_{k'}\right)/(K-2)}{\lambda\left(\lambda(K-1) + \nu_k\|\mathbf{u}_k\|^2\right)}\right)\mathbf{u}_k\right\|^2. \tag{91}$$

Under the assumption that $\|\mathbf{u}_k\| = 1$, and $\{\mathbf{u}_k\}_{k=1}^K$ are orthogonal, the second term in (91) reduces to zero, and we then have:

$$\left\|\widetilde{\mathbf{H}}^{-1}(\boldsymbol{\theta})\hat{\nabla}F(\boldsymbol{\theta})\right\|^2 = \left\|\sum_{k=1}^{K}\frac{\mu\nu_k}{\lambda(K-1)}\mathbf{u}_k\right\|^2 \overset{(a)}{=} \frac{\mu^2}{\lambda^2(K-1)^2}\sum_{k=1}^{K}\nu_k^2\|\mathbf{u}_k\|^2 = \frac{K\mu^2}{\lambda^2(K-1)^2}\nu^2\|\mathbf{u}\|^2, \tag{92}$$

where $(a)$ follows from the independence of $\{\mathbf{u}_k\}_{k=1}^K$.

Next, the second term in (90) can be bounded by:

$$\left\|\widetilde{\mathbf{H}}(\boldsymbol{\theta}) - \nabla^2 F(\boldsymbol{\theta}) - \lambda\mathbf{I}_d\right\|_F^2 \overset{(a)}{\leq} \left\|\hat{\mathbf{H}}(\boldsymbol{\theta}) - \nabla^2 F_\mu(\boldsymbol{\theta})\right\|_F^2 + L_2^2\mu^2 d \overset{(b)}{\leq} \frac{K}{(K-1)^2}\nu^2\|\mathbf{u}\|^4 + L_2^2\mu^2 d. \tag{93}$$

where $(a)$ results from (70), and $(b)$ follows from (69).

Combine (92) and (93), we have:

$$\mathbb{E}\left[\left\|\widetilde{\mathbf{H}}^{-1}(\boldsymbol{\theta})\hat{\nabla}F(\boldsymbol{\theta})\right\|^2\left\|\widetilde{\mathbf{H}}(\boldsymbol{\theta}) - \nabla^2 F(\boldsymbol{\theta}) - \lambda\mathbf{I}_d\right\|_F^2\right] \leq \frac{K\mu^2}{\lambda^2(K-1)^2}\mathbb{E}\left[\nu^2\|\mathbf{u}\|^2\left(\frac{K}{(K-1)^2}\nu^2\|\mathbf{u}\|^4 + L_2^2\mu^2 d\right)\right]$$

$$= \frac{K\mu^2}{\lambda^2(K-1)^2}\left(\frac{K}{(K-1)^2}\mathbb{E}[\nu^4\|\mathbf{u}\|^6] + L_2^2\mu^2 d\,\mathbb{E}[\nu^2\|\mathbf{u}\|^2]\right). \tag{94}$$

Given Lem. C.4 and the following inequality:

$$\mathbb{E}[\nu^4\|\mathbf{u}\|^6] \overset{(a)}{\leq} \mathbb{E}[\nu^2\|\mathbf{u}\|^6]\cdot\max|\nu^2| \overset{(b)}{\leq} \left(\frac{2G}{\mu^2}\right)^2\frac{d(d+2)(d+4)}{\mu^4}\left(\sigma_\xi^2 + 4L_0^2\mu^2(d+3)\right), \tag{95}$$

where $(a)$ follows from Holder's inequality $\mathbb{E}[|XY|] \leq \mathbb{E}[|X|^p]^{1/p}\mathbb{E}[|Y|^q]^{1/q}$ for $p = 1$ and $q = \infty$, and $(b)$ comes from Lem. C.4 and the bound $\max|\nu| = \max|f(\boldsymbol{\theta} + \mu\mathbf{u}; \xi) - F_\mu(\boldsymbol{\theta})|/\mu^2 = 2G/\mu^2$.

Therefore, (94) can be further bounded by:

$$\mathbb{E}\left[\left\|\widetilde{\mathbf{H}}^{-1}(\boldsymbol{\theta})\hat{\nabla}F(\boldsymbol{\theta})\right\|^2 \left\|\widetilde{\mathbf{H}}(\boldsymbol{\theta}) - \nabla^2 F(\boldsymbol{\theta}) - \lambda\mathbf{I}_d\right\|_F^2\right] \leq \frac{Kd}{\lambda^2(K-1)^2}\left(B_1\sigma_\xi^2 + 4B_2 L_0^2\mu^2\right), \tag{96}$$

where we denote $B \triangleq \frac{4KG^2}{(K-1)^2\mu^6}(d+2)(d+4)$, $B_1 \triangleq \frac{B}{\mu^6} + L_2^2 d$, and $B_2 \triangleq \frac{B}{\mu^6}(d+3) + L_2^2 d(d+1)$ for simplicity.

Adopting the results from (86), the third term in (90) can be bounded as:

$$\left\|(\nabla^2 F(\boldsymbol{\theta}) + \lambda\mathbf{I}_d)^{-1}\right\|_2^2 \leq \left\|(\nabla^2 F(\boldsymbol{\theta}) + \lambda\mathbf{I}_d)^{-1}\right\|_F^2 = \sum_{i=1}^d \frac{1}{(\sigma_i + \lambda)^2} \leq \frac{d}{(\sigma_{\min} + \lambda)^2}, \tag{97}$$

where $\sigma_i$ denotes the $i$-th eigenvalue of $\nabla^2 F(\boldsymbol{\theta})$, and $\sigma_{\min}$ denotes the smallest eigenvalue respectively.

Overall, we have:

$$\mathbb{E}\left[\left\|\widetilde{\mathbf{H}}^{-1}(\boldsymbol{\theta})\hat{\nabla}F(\boldsymbol{\theta}) - \left(\nabla^2 F(\boldsymbol{\theta}) + \lambda\mathbf{I}_d\right)^{-1}\hat{\nabla}F(\boldsymbol{\theta})\right\|^2\right] \leq \frac{Kd^2\left(B_1\sigma_\xi^2 + 4B_2 L_0^2\mu^2\right)}{\lambda^2(\sigma_{\min} + \lambda)^2(K-1)^2}. \tag{98}$$

$\square$

### C.15. Proof of Thm. 5.5

**Theorem C.9** (Convergence of ZoVH (Formal)). *Under Assump. 4.3, 4.4, let the baseline $b_t$ be the optimal baseline in Thm. 4.7, when $\eta \sim \mathcal{O}\left(\frac{4\lambda(\sigma_{\min}+\lambda)^2(K-1)^2\epsilon^2}{L_1 Kd^2\left(B_1\sigma_\xi^2+4B_2 L_0^2\mu^2\right)}\right) \sim \mathcal{O}(\epsilon^2)$, and $T \sim \mathcal{O}\left(\frac{2\lambda\Delta}{\eta\epsilon^2}\right) \sim \mathcal{O}(\epsilon^{-4})$, where $\Delta \triangleq F(\boldsymbol{\theta}_0) - F(\boldsymbol{\theta}^*)$ denotes the initial optimality gap, $\sigma_{\min}$ denotes the minimum eigenvalue of the true Hessian $\nabla^2 F(\boldsymbol{\theta})$, and $B_1, B_2$ are two constants in Thm. 5.4, the following holds:*

$$\frac{1}{T}\sum_{t=0}^{T-1}\mathbb{E}[\|\nabla F(\boldsymbol{\theta}_t)\|] \leq \epsilon + \frac{\mu L_1 d}{2}.$$

*Proof.* Due to the $L_1$-Lipschitz smoothness of $F(\boldsymbol{\theta})$ given in Lem. C.1, the smoothed objective $F_\mu(\boldsymbol{\theta})$ is also $L_1$-Lipschitz smooth:

$$\|\nabla F_\mu(\boldsymbol{\theta}) - \nabla F_\mu(\boldsymbol{\theta}')\| \overset{(a)}{\leq} \mathbb{E}_\mathbf{u}\left[\|\nabla F(\boldsymbol{\theta} + \mu\mathbf{u}) - \nabla F(\boldsymbol{\theta}' + \mu\mathbf{u})\|\right] \leq L_1\|\boldsymbol{\theta} - \boldsymbol{\theta}'\|, \tag{99}$$

where $(a)$ follows from Jensen's inequality.

Therefore, we have:

$$F_\mu(\boldsymbol{\theta}_{t+1}) - F_\mu(\boldsymbol{\theta}_t) \leq \langle \nabla F_\mu(\boldsymbol{\theta}_t), \boldsymbol{\theta}_{t+1} - \boldsymbol{\theta}_t\rangle + \frac{L_1}{2}\|\boldsymbol{\theta}_{t+1} - \boldsymbol{\theta}_t\|^2. \tag{100}$$

For the first term in the RHS of (100), we have:

$$\mathbb{E}[\langle \nabla F_\mu(\boldsymbol{\theta}_t), \boldsymbol{\theta}_{t+1} - \boldsymbol{\theta}_t\rangle] = -\eta\mathbb{E}\left[\left\langle \nabla F_\mu(\boldsymbol{\theta}_t), \widetilde{\mathbf{H}}^{-1}(\boldsymbol{\theta}_t)\hat{\nabla}F(\boldsymbol{\theta}_t)\right\rangle\right]$$

$$\overset{(a)}{=} -\eta\mathbb{E}\left[\left\langle \nabla F_\mu(\boldsymbol{\theta}_t), \sum_{k=1}^K \mu\nu_k\left(\frac{1}{\lambda(K-1)} - \frac{\mathbf{u}_k^\top\left(\sum_{k'=1, k'\neq k}^K \nu_{k'}\mathbf{u}_{k'}\right)/(K-2)}{\lambda\left(\lambda(K-1) + \nu_k\|\mathbf{u}_k\|^2\right)}\right)\mathbf{u}_k\right\rangle\right] \tag{101}$$

$$\overset{(b)}{=} -\eta\mathbb{E}\left[\left\langle \nabla F_\mu(\boldsymbol{\theta}_t), \sum_{k=1}^K \frac{\mu\nu_k}{\lambda(K-1)}\mathbf{u}_k\right\rangle\right] = -\frac{\eta}{\lambda}\mathbb{E}\left[\left\langle \nabla F_\mu(\boldsymbol{\theta}_t), \hat{\nabla}F(\boldsymbol{\theta}_t)\right\rangle\right] \overset{(c)}{=} -\frac{\eta}{\lambda}\|\nabla F_\mu(\boldsymbol{\theta}_t)\|^2,$$

where $(a)$ follows from the definition of $\widetilde{\mathbf{H}}^{-1}(\boldsymbol{\theta}_t)$, $(b)$ follows from the orthogonality of $\{\mathbf{u}_k\}_{k=1}^K$, and $(c)$ follows from Lem. C.2.

For the second term in the RHS of (100), we have:

$$\frac{L_1}{2}\mathbb{E}\Big[\|\boldsymbol{\theta}_{t+1} - \boldsymbol{\theta}_t\|^2\Big] = \frac{L_1\eta^2}{2}\mathbb{E}\Big[\big\|\widetilde{\mathbf{H}}^{-1}(\boldsymbol{\theta}_t)\hat{\nabla}F(\boldsymbol{\theta}_t)\big\|^2\Big] \overset{(a)}{\leq} \frac{L_1\eta^2 K d^2\big(B_1\sigma_\xi^2 + 4B_2 L_0^2\mu^2\big)}{2\lambda^2(\sigma_{\min} + \lambda)^2(K-1)^2},\tag{102}$$

where $(a)$ follows from the result in Thm. 5.4.

Summing all iterations from 0 to $T-1$ to both sides of (100), and denote $\Delta = F_\mu(\boldsymbol{\theta}_0) - F_\mu(\boldsymbol{\theta}^*)$, we have:

$$\frac{1}{T}\sum_{t=0}^{T-1}\mathbb{E}\Big[\|\nabla F_\mu(\boldsymbol{\theta}_t)\|^2\Big] \leq \frac{\lambda\Delta}{\eta T} + \frac{L_1\eta K d^2\big(B_1\sigma_\xi^2 + 4B_2 L_0^2\mu^2\big)}{2\lambda(\sigma_{\min} + \lambda)^2(K-1)^2}.\tag{103}$$

To simplify the expression, we choose $\eta \sim \mathcal{O}\left(\frac{4\lambda(\sigma_{\min}+\lambda)^2(K-1)^2\epsilon^2}{L_1 K d^2\big(B_1\sigma_\xi^2 + 4B_2 L_0^2\mu^2\big)}\right) \sim \mathcal{O}(\epsilon^2)$, and $T \sim \mathcal{O}\left(\frac{2\lambda\Delta}{\eta\epsilon^2}\right) \sim \mathcal{O}(\epsilon^{-4})$, such that:

$$\frac{1}{T}\sum_{t=0}^{T-1}\mathbb{E}\Big[\|\nabla F_\mu(\boldsymbol{\theta}_t)\|^2\Big] \leq \frac{1}{2}\epsilon^2 + \frac{1}{2}\epsilon^2 = \epsilon^2.\tag{104}$$

Finally, since $\|\nabla F(\boldsymbol{\theta}_t) - \nabla F_\mu(\boldsymbol{\theta}_t)\| \leq \frac{\mu L_1 d}{2}$, we have:

$$\begin{aligned}
\frac{1}{T}\sum_{t=0}^{T-1}\mathbb{E}[\|\nabla F(\boldsymbol{\theta}_t)\|] &\leq \frac{1}{T}\sum_{t=0}^{T-1}\mathbb{E}[\|\nabla F_\mu(\boldsymbol{\theta}_t)\|] + \frac{1}{T}\sum_{t=0}^{T-1}\mathbb{E}[\|\nabla F(\boldsymbol{\theta}_t) - \nabla F_\mu(\boldsymbol{\theta}_t)\|] \\
&\overset{(a)}{\leq} \sqrt{\frac{1}{T}\sum_{t=0}^{T-1}\mathbb{E}\Big[\|\nabla F_\mu(\boldsymbol{\theta}_t)\|^2\Big]} + \frac{\mu L_1 d}{2} \leq \epsilon + \frac{\mu L_1 d}{2},
\end{aligned}\tag{105}$$

where $(a)$ follows from Jensen's inequality. $\qquad\square$

### C.16. Proof of Thm. B.1

Before proving Thm. B.1, we first present the variance analysis of Hessian estimator with query reuse technique and the optimal baseline in Thm. 4.7:

**Lemma C.10** (Variance of Hessian Estimator for $N > 1$)**.** *Consider the Hessian estimator with query reuse in Def. 4.2 utilizing the optimal baseline derived in Thm. 4.7. Let $\mathbf{u} \sim \mathcal{N}(\mathbf{0}, \mathbf{I}_d)$. Under Assump. 4.3 and 4.4, the variance of the estimator is bounded by:*

$$\mathbb{E}\Big[\big\|\widehat{\mathbf{H}}(\boldsymbol{\theta}) - \nabla^2 F_\mu(\boldsymbol{\theta})\big\|_F^2\Big] \leq \frac{NKd(d+2)\lambda^2 V}{\lambda^2(NK-1)^2\mu^4 - L_0^2\eta^2(N-1)^2 K/3},$$

*where $V \triangleq \sigma_\xi^2 + 4L_0^2\mu^2(d+2)$.*

*Proof.* We adopt the decomposition of the variance of the Hessian estimator with query reuse from (10), and let $b$ be the

averaged baseline in Thm. 4.7. Then we have:

$$
\begin{aligned}
\mathrm{Var}\left(\widehat{\mathbf{H}}(\boldsymbol{\theta}_{t-1})\right) =& \mathbb{E}\left[\left\|\frac{1}{NK-1}\sum_{n,k=1}^{N,K}\frac{f(\boldsymbol{\theta}_{t-n}+\mu\mathbf{u}_{t-n,k};\xi)-b}{\mu^2}\mathbf{u}_{t-n,k}\mathbf{u}_{t-n,k}^\top\right\|_F^2\right]-\left\|\nabla^2 F_\mu(\boldsymbol{\theta}_{t-1})\right\|_F^2 \\
\leq& \mathbb{E}\left[\left\|\frac{1}{NK-1}\sum_{n,k=1}^{N,K}\frac{f(\boldsymbol{\theta}_{t-n}+\mu\mathbf{u}_{t-n,k};\xi)-b}{\mu^2}\mathbf{u}_{t-n,k}\mathbf{u}_{t-n,k}^\top\right\|_F^2\right] \\
\overset{(a)}{=}& \frac{1}{(NK-1)^2\mu^4}\sum_{n,k=1}^{N,K}\mathbb{E}\left[\left|f(\boldsymbol{\theta}_{t-n}+\mu\mathbf{u}_{t-n,k};\xi)-\frac{1}{N}\sum_{n'=1}^{N}F_\mu(\boldsymbol{\theta}_{t-n'})\right|^2\left\|\mathbf{u}_{t-n,k}\mathbf{u}_{t-n,k}^\top\right\|_F^2\right] \\
=& \frac{K}{(NK-1)^2\mu^4}\sum_{n=1}^{N}\mathbb{E}\left[\left|f(\boldsymbol{\theta}_{t-n}+\mu\mathbf{u}_{t-n,k};\xi)-\frac{1}{N}\sum_{n'=1}^{N}F_\mu(\boldsymbol{\theta}_{t-n'})\right|^2\left\|\mathbf{u}_{t-n,k}\mathbf{u}_{t-n,k}^\top\right\|_F^2\right] \\
\overset{(b)}{\leq}& \frac{NKd(d+2)\lambda^2 V}{\lambda^2(NK-1)^2\mu^4-L_0^2\eta^2\mu^2K(N^2-1)d(d+2)/3}\,,
\end{aligned}
\tag{106}
$$

where $(a)$ follows from the independence of $\{\mathbf{u}_k\}$ across different queries $k$, $(b)$ comes from Lem. C.6. Here, we denote $V \triangleq \sigma_\xi^2 + 4L_0^2\mu^2(d+2)$ for simplicity. $\qquad\square$

Now we are ready to prove Thm. B.1.

*Proof.* The total error is bounded as:

$$
\begin{aligned}
\mathbb{E}\left[\left\|\widehat{\mathbf{H}}(\boldsymbol{\theta})-\nabla^2 F(\boldsymbol{\theta})\right\|_F^2\right] &\overset{(a)}{=} \mathbb{E}\left[\left\|\widehat{\mathbf{H}}(\boldsymbol{\theta})-\nabla^2 F_\mu(\boldsymbol{\theta})\right\|_F^2\right]+\left\|\nabla F_\mu(\boldsymbol{\theta})-\nabla F(\boldsymbol{\theta})\right\|_F^2 \\
&\overset{(b)}{\leq} \frac{NKd(d+2)\lambda^2 V}{\lambda^2(NK-1)^2\mu^4-L_0^2\eta^2\mu^2K(N^2-1)d(d+2)/3}+L_2^2\mu^2 d\,,
\end{aligned}
\tag{107}
$$

where $(a)$ follows $\mathbb{E}\left[\widehat{\mathbf{H}}(\boldsymbol{\theta})-\nabla^2 F_\mu(\boldsymbol{\theta})\right]=0$ from Thm. 4.6, and $(b)$ comes from Thm. C.10 and Lem. C.3. Here we denote $V \triangleq \sigma_\xi^2 + 4L_0^2\mu^2(d+2)$ in the last step for simplicity. $\qquad\square$

# D. Additional Discussion

## D.1. Equivalence between Prop. 3.2 with Averaged Baseline and Def. 4.1

The Gaussian smoothed Hessian estimator in Prop. 3.2 with the averaged baseline $b = \mathbb{E}[f(\boldsymbol{\theta}+\mu\mathbf{u}_k;\xi)]$ is given as:

$$
\nabla^2 F_\mu(\boldsymbol{\theta}) = \mathbb{E}_{\mathbf{u}}\left[\frac{\mathbb{E}_\xi[f(\boldsymbol{\theta}+\mu\mathbf{u};\xi)]-\mathbb{E}[f(\boldsymbol{\theta}+\mu\mathbf{u}_k;\xi)]}{\mu^2}\left(\mathbf{u}\mathbf{u}^\top-\mathbf{I}_d\right)\right]\,.
\tag{108}
$$

Moreover, the practical implementation (via Monte Carlo) of the above estimator is:

$$
\widehat{\mathbf{H}}(\boldsymbol{\theta}) = \frac{1}{K-1}\sum_{k=1}^{K}\frac{f(\boldsymbol{\theta}+\mu\mathbf{u}_k;\xi)-\frac{1}{K}\sum_{k'=1}^{K}f(\boldsymbol{\theta}+\mu\mathbf{u}_{k'};\xi)}{\mu^2}\left(\mathbf{u}_k\mathbf{u}_k^\top-\mathbf{I}_d\right)\,,
\tag{109}
$$

where the $\frac{1}{K-1}$ factor is used to make the estimator unbiased (similar to Def. 4.1).

The estimator (109) can be further decomposed as:

$$\widehat{\mathbf{H}}(\boldsymbol{\theta}) = \underbrace{\frac{1}{K-1} \sum_{k=1}^{K} \frac{f(\boldsymbol{\theta} + \mu \mathbf{u}_k; \xi) - \frac{1}{K} \sum_{k'=1}^{K} f(\boldsymbol{\theta} + \mu \mathbf{u}_{k'}; \xi)}{\mu^2} \mathbf{u}_k \mathbf{u}_k^{\top}}_{\text{Def. 4.1}}$$

$$\underbrace{- \frac{1}{K-1} \sum_{k=1}^{K} \frac{f(\boldsymbol{\theta} + \mu \mathbf{u}_k; \xi) - \frac{1}{K} \sum_{k'=1}^{K} f(\boldsymbol{\theta} + \mu \mathbf{u}_{k'}; \xi)}{\mu^2} \mathbf{I}_d}_{=0} , \tag{110}$$

where the first term is exactly the Hessian estimator in Def. 4.1, and the second term equals zero since:

$$\frac{1}{K-1} \sum_{k=1}^{K} \frac{f(\boldsymbol{\theta} + \mu \mathbf{u}_k; \xi) - \frac{1}{K} \sum_{k'=1}^{K} f(\boldsymbol{\theta} + \mu \mathbf{u}_{k'}; \xi)}{\mu^2} \mathbf{I}_d$$

$$= \left( \frac{1}{K-1} \sum_{k=1}^{K} \frac{f(\boldsymbol{\theta} + \mu \mathbf{u}_k; \xi)}{\mu^2} - \frac{K}{K-1} \frac{1}{K} \sum_{k'=1}^{K} \frac{f(\boldsymbol{\theta} + \mu \mathbf{u}_{k'}; \xi)}{\mu^2} \right) \mathbf{I}_d \tag{111}$$

$$= \left( \frac{1}{K-1} \sum_{k=1}^{K} \frac{f(\boldsymbol{\theta} + \mu \mathbf{u}_k; \xi)}{\mu^2} - \frac{1}{K-1} \sum_{k=1}^{K} \frac{f(\boldsymbol{\theta} + \mu \mathbf{u}_k; \xi)}{\mu^2} \right) \mathbf{I}_d = 0 .$$

Overall, we show that the Gaussian smoothed Hessian estimator in Prop. 3.2 with the averaged baseline is equivalent to the one in Def. 4.1.

### D.2. Smoothing Bias vs. Smoothing Radius $\mu$

The Hessian estimator in Thm. 4.6 is unbiased for the Gaussian-smoothed Hessian $\nabla^2 F_\mu(\boldsymbol{\theta})$ rather than the exact Hessian $\nabla^2 F(\boldsymbol{\theta})$. The resulting smoothing bias between $\nabla^2 F_\mu(\boldsymbol{\theta})$ and $\nabla^2 F(\boldsymbol{\theta})$ is controlled directly by the smoothing radius $\mu$. Below we present the formal bound on the smoothing bias in terms of $\mu$:

**Proposition D.1** (Bias of Smoothed Hessian)**.** *Under Assump. 4.3, for* $\mathbf{u} \sim \mathcal{N}(\mathbf{0}, \mathbf{I}_d)$ *and* $F_\mu(\boldsymbol{\theta}) = \mathbb{E}_{\mathbf{u}}[F(\boldsymbol{\theta} + \mu \mathbf{u})]$, *the Gaussian-smoothed Hessian satisfies:*

$$\left\| \nabla^2 F_\mu(\boldsymbol{\theta}) - \nabla^2 F(\boldsymbol{\theta}) \right\|_F \leq L_2 \mu \sqrt{d} . \tag{112}$$

**Remark.** Prop. D.1 gives the unsquared version of Lem. C.3, thus the proof is similar. The bound in Prop. D.1 explicitly characterizes the dependence of the smoothing bias on the smoothing radius $\mu$. Thus, decreasing $\mu$ reduces the gap between the smoothed and true Hessians linearly. At the same time, the finite-sample variance terms (Thm. 4.8) in these bounds contain inverse powers of $\mu$, so $\mu$ should be interpreted as a bias-variance tuning parameter rather than simply set as small as possible in noisy finite-sample estimation.

**Empirical Validation.** We also empirically evaluate the Frobenius norm gap $\left\| \nabla^2 F_\mu(\boldsymbol{\theta}) - \nabla^2 F(\boldsymbol{\theta}) \right\|_F$ under different smoothing radii on Rosenbrock and Styblinski-Tang functions to validate Prop. D.1.

As shown in Fig. 4, the bias (Frobenius norm gap) proportionally increases with $\mu$ in a log-log scale, with slopes approximately around 1, matching the linear dependence predicted by Prop. D.1. These results indicate that the smoothing bias is well controlled for sufficiently small $\mu$.

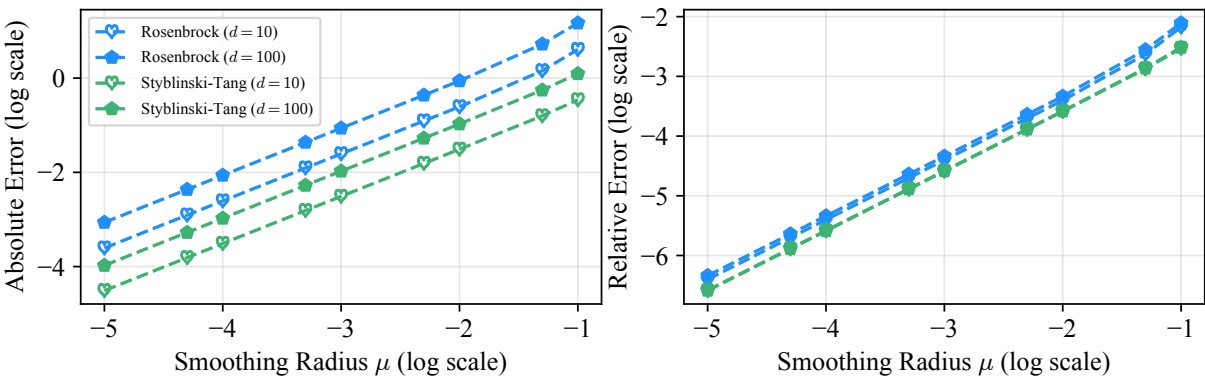

*Figure 4.* Frobenius norm error $\left\|\nabla^2 F_\mu(\boldsymbol{\theta}) - \nabla^2 F(\boldsymbol{\theta})\right\|_F$ under different smoothing radii $\mu$. The log-log slopes are 1.04 for Rosenbrock and 1.01 for Styblinski-Tang. The errors are averaged over 3 independent runs. Since the standard deviations are small, we omit the error bars for better visualization.

### D.3. Connection Between Optimal Baseline and Control Variates

The optimal baseline result in Thm. 4.7 can be interpreted as a direct analogue of the classical control variate principle (Asmussen & Glynn, 2007). For the Gaussian Hessian identity in Prop. 3.2, define the random matrix variables:

$$\mathbf{A}(\mathbf{u}) \triangleq \frac{F(\boldsymbol{\theta} + \mu\mathbf{u})}{\mu^2}\left(\mathbf{u}\mathbf{u}^\top - \mathbf{I}_d\right), \qquad \mathbf{B}(\mathbf{u}) \triangleq \frac{1}{\mu^2}\left(\mathbf{u}\mathbf{u}^\top - \mathbf{I}_d\right). \tag{113}$$

Since $\mathbb{E}_\mathbf{u}[\mathbf{B}(\mathbf{u})] = \mathbf{0}$, the baseline estimator can be written as $\mathbf{A}(\mathbf{u}) - b\mathbf{B}(\mathbf{u})$ and remains unbiased for any constant $b$ that is independent of $\mathbf{u}$:

$$\mathbb{E}_\mathbf{u}[\mathbf{A}(\mathbf{u}) - b\mathbf{B}(\mathbf{u})] = \mathbb{E}_\mathbf{u}[\mathbf{A}(\mathbf{u})] = \nabla^2 F_\mu(\boldsymbol{\theta}) . \tag{114}$$

Thus, the baseline term is a zero-mean control variate whose coefficient is chosen only to reduce variance, not to change the target expectation.

Because the estimator is matrix-valued, the natural scalar criterion is the Frobenius variance,

$$\min_{b \in \mathbb{R}} \mathbb{E}_\mathbf{u}\left[\left\|\mathbf{A}(\mathbf{u}) - b\mathbf{B}(\mathbf{u}) - \nabla^2 F_\mu(\boldsymbol{\theta})\right\|_F^2\right] . \tag{115}$$

Differentiating with respect to $b$ gives

$$b^* = \frac{\mathbb{E}_\mathbf{u}\left[\langle\mathbf{A}(\mathbf{u}) - \nabla^2 F_\mu(\boldsymbol{\theta}), \mathbf{B}(\mathbf{u})\rangle_F\right]}{\mathbb{E}_\mathbf{u}\left[\|\mathbf{B}(\mathbf{u})\|_F^2\right]} = \frac{\mathbb{E}_\mathbf{u}[\langle\mathbf{A}(\mathbf{u}), \mathbf{B}(\mathbf{u})\rangle_F]}{\mathbb{E}_\mathbf{u}\left[\|\mathbf{B}(\mathbf{u})\|_F^2\right]} = \frac{\mathbb{E}_\mathbf{u}\left[F(\boldsymbol{\theta} + \mu\mathbf{u})\left\|\mathbf{u}\mathbf{u}^\top - \mathbf{I}_d\right\|_F^2\right]}{\mathbb{E}_\mathbf{u}\left[\left\|\mathbf{u}\mathbf{u}^\top - \mathbf{I}_d\right\|_F^2\right]} , \tag{116}$$

where the second equality uses $\mathbb{E}_\mathbf{u}[\mathbf{B}(\mathbf{u})] = \mathbf{0}$. This is exactly the optimal baseline in (13). When $\left\|\mathbf{u}\mathbf{u}^\top - \mathbf{I}_d\right\|_F^2$ concentrates in high dimensions, the coefficient reduces to $F_\mu(\boldsymbol{\theta})$, which is estimated in practice by the averaged baseline.

### D.4. Exact Woodbury Inverse vs. Approximate Inverse Hessian

The inverse Hessian approximation in Def. 5.1 replaces the full Gram matrix $\mathbf{U}^\top\mathbf{U}$ in the exact Woodbury inverse with its diagonal part. This step is exact when the random directions are pairwise orthogonal. For Gaussian directions, it is also accurate in high dimensions because the diagonal terms $\|\mathbf{u}_k\|^2$ are of order $d$, while the off-diagonal terms $\mathbf{u}_i^\top\mathbf{u}_j$ are only of order $\sqrt{d}$ for $i \neq j$. The following proposition gives the resulting approximation error.

**Proposition D.2** (Exact and Approximate Inverse Error). *Let* $\mathbf{U} = [\mathbf{u}_1, \ldots, \mathbf{u}_K] \in \mathbb{R}^{d \times K}$ *with* $\mathbf{u}_k \stackrel{\text{i.i.d.}}{\sim} \mathcal{N}(\mathbf{0}, \mathbf{I}_d)$. *Define* $\mathbf{G} \triangleq \mathbf{U}^\top\mathbf{U}$, $\Delta \triangleq \mathrm{diag}(\mathbf{G})$, *and* $\mathbf{E} \triangleq \mathbf{G} - \Delta$. *Let*

$$\mathbf{M} \triangleq \mathbf{D}^{-1} + \frac{1}{\lambda}\mathbf{G}, \quad \widetilde{\mathbf{M}} \triangleq \mathbf{D}^{-1} + \frac{1}{\lambda}\Delta , \tag{117}$$

where $\mathbf{D}$ is defined as in Def. 5.1. Assume $\lambda > 0$ is fixed, $\left\|\mathbf{D}^{-1}\right\|_2 = \mathcal{O}_p(1)$. Then,

$$\left\|\left(\widehat{\mathbf{H}}(\boldsymbol{\theta}) + \lambda \mathbf{I}_d\right)^{-1} - \widetilde{\mathbf{H}}^{-1}(\boldsymbol{\theta})\right\|_F = \mathcal{O}_p\left(\frac{K}{\sqrt{d}}\right). \tag{118}$$

*Proof.* From (14) and (15), the only difference between the exact inverse and the approximation is whether $\mathbf{M}^{-1}$ or $\widetilde{\mathbf{M}}^{-1}$ is used in the low-dimensional Woodbury correction:

$$\left(\widehat{\mathbf{H}}(\boldsymbol{\theta}) + \lambda \mathbf{I}_d\right)^{-1} - \widetilde{\mathbf{H}}^{-1}(\boldsymbol{\theta}) = -\frac{1}{\lambda^2}\mathbf{U}\left(\mathbf{M}^{-1} - \widetilde{\mathbf{M}}^{-1}\right)\mathbf{U}^{\top}. \tag{119}$$

For Gaussian directions, standard concentration gives $\|\mathbf{U}\|_2^2 = \|\mathbf{G}\|_2 = \mathcal{O}_p(d), \|\Delta\|_2 = \Theta_p(d)$, and $\|\mathbf{E}\|_F = \mathcal{O}_p(K\sqrt{d})$, since the off-diagonal entries $\mathbf{u}_i^{\top}\mathbf{u}_j$ have variance $d$ for $i \neq j$. Because $\left\|\mathbf{D}^{-1}\right\|_2 = \mathcal{O}_p(1)$ and $\lambda$ is fixed, the diagonal matrix $\widetilde{\mathbf{M}}$ has entries of order $d$, hence $\left\|\widetilde{\mathbf{M}}^{-1}\right\|_2 = \mathcal{O}_p(d^{-1})$.

By the assumed stability of the exact Woodbury inverse, $\left\|\mathbf{M}^{-1}\right\|_2 = \mathcal{O}_p(d^{-1})$. Then, by the resolvent identity,

$$\mathbf{M}^{-1} - \widetilde{\mathbf{M}}^{-1} = -\frac{1}{\lambda}\mathbf{M}^{-1}\mathbf{E}\widetilde{\mathbf{M}}^{-1}. \tag{120}$$

Therefore,

$$\begin{aligned}
\left\|\left(\widehat{\mathbf{H}}(\boldsymbol{\theta}) + \lambda \mathbf{I}_d\right)^{-1} - \widetilde{\mathbf{H}}^{-1}(\boldsymbol{\theta})\right\|_F &\leq \frac{1}{\lambda^3}\|\mathbf{U}\|_2^2\left\|\mathbf{M}^{-1}\right\|_2\|\mathbf{E}\|_F\left\|\widetilde{\mathbf{M}}^{-1}\right\|_2 \\
&= \mathcal{O}_p(d) \cdot \mathcal{O}_p(d^{-1}) \cdot \mathcal{O}_p(K\sqrt{d}) \cdot \mathcal{O}_p(d^{-1}) = \mathcal{O}_p\left(\frac{K}{\sqrt{d}}\right),
\end{aligned} \tag{121}$$

which completes the proof. $\qquad\square$

**Remark.** Prop. D.2 formalizes the intuition behind the diagonal Gram approximation in (15). The result keeps the dependence on the number of query directions as $\mathcal{O}_p(K/\sqrt{d})$. For the small fixed $K$ used in our experiments, this reduces to the $d^{-1/2}$ decay observed empirically. Thus, the approximation becomes increasingly accurate in the high-dimensional regimes targeted by ZoVH, while avoiding the need to manipulate dense $d \times d$ inverse matrices.

**Empirical Validation.** We further empirically measure the Frobenius norm error between the exact Woodbury inverse in (14) and the approximation in (15).

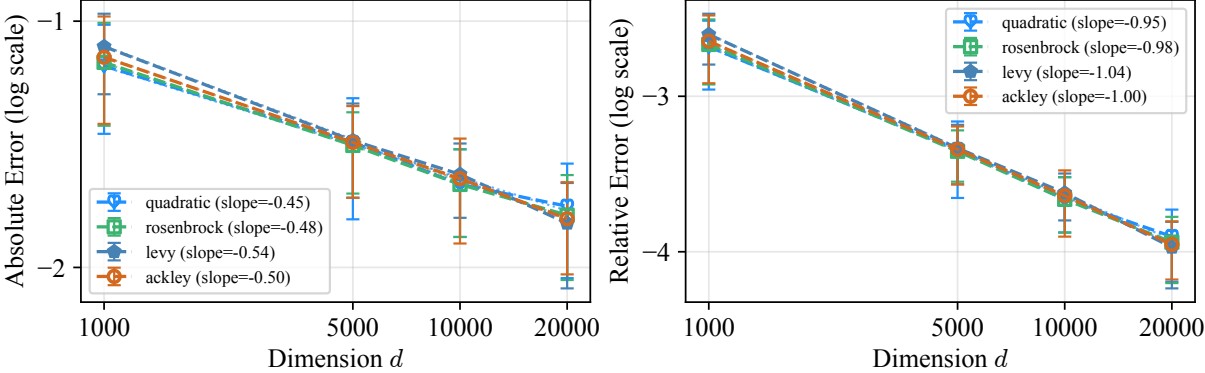

*Figure 5.* Frobenius norm error between the exact Woodbury inverse and the approximate inverse as the dimension $d$ increases. We use $K = 3$ and average the result over 50 random seeds.

As shown in Fig. 5, the error decreases consistently as the dimension increases. The fitted slopes are close to $-0.5$ across four synthetic functions, matching the $d^{-1/2}$ dependence in Prop. D.2. The result supports using the approximate inverse for the small $K$ setting used by ZoVH, especially in LLM fine-tuning where individual layers contain millions of parameters.

## D.5. Inverse Hessian-Gradient Product Approximation with Query Reuse

**Definition D.3** (Shared Product with Query Reuse). Assume $\widetilde{\mathbf{H}}^{-1}(\boldsymbol{\theta})$ and $\hat{\nabla}F(\boldsymbol{\theta})$ are estimated using the same set of standard Gaussian directions $\{\mathbf{u}_k\}_{k=1}^{K}$. The bias-corrected inverse Hessian-gradient product $\widetilde{\mathbf{H}}^{-1}(\boldsymbol{\theta})\hat{\nabla}F(\boldsymbol{\theta})$ with query reuse is given by:

$$\widetilde{\mathbf{H}}^{-1}(\boldsymbol{\theta}_{t-1})\hat{\nabla}F(\boldsymbol{\theta}_{t-1}) \triangleq \sum_{i,k=1}^{N,K} \frac{\mu\nu_{t-i,k}}{\lambda} \left( \frac{1}{NK-1} - \frac{\mathbf{u}_{t-i,k}^{\top} \left( \sum_{(j,k')\neq(i,k)} \nu_{t-j,k'}\mathbf{u}_{t-j,k'} \right)/(NK-2)}{\lambda(NK-1) + \nu_{t-i,k}\|\mathbf{u}_{t-i,k}\|^2} \right) \mathbf{u}_{t-i,k}. \quad (122)$$

where the averaged baseline $\hat{b}_t \triangleq \frac{1}{NK}\sum_{i,k=1}^{N,K} y_{t-i,k}$, and $\nu_{t-i,k} \triangleq \left( y_{t-i,k} - \hat{b}_t \right)/\mu^2$.

**Remark.** The definition above extends Def. 5.3 to incorporate query reuse across $N$ iterations. By reusing the same set of $K$ random directions over $N$ consecutive iterations, we effectively increase the sample size to $NK$, which enhances the accuracy of both the Hessian and gradient estimations. This approach leverages historical query information, reducing the need for additional function evaluations while maintaining robust optimization performance.

## D.6. Comparison with HiZOO (Zhao et al., 2025)

HiZOO (Zhao et al., 2025) is a recently proposed curvature-aware zeroth-order optimization method that utilizes a randomized central-difference Hessian estimator (5) for memory-efficient LLMs fine-tuning. However, to avoid the computational complexity of inverting the full Hessian estimator, HiZOO only approximates the diagonal of the Hessian estimator to facilitate inversion. Besides, recent studies (Zhang et al., 2024) have shown that the practical Hessian of LLMs exhibits significant block-diagonal structures, indicating that simple diagonal approximations may overlook important curvature information.

In contrast, our proposed method ZoVH addresses this by approximating the full Hessian matrix and captures more comprehensive curvature information than diagonal approximations. This leads to more accurate inverse Hessian-gradient products, which are crucial for effective optimization steps. A detailed comparison of HiZOO and ZoVH in terms of Hessian approximation and its inverse is summarized in Table 2.

*Table 2.* Comparison between HiZOO (Zhao et al., 2025) and ZoVH.

| Method | HiZOO | ZoVH |
|---|:---:|:---:|
| **Gradient** | $\hat{\nabla}F(\boldsymbol{\theta}) \triangleq \frac{1}{2K}\sum_{k=1}^{K} \frac{f(\boldsymbol{\theta}+\mu\mathbf{u}_k;\xi)-f(\boldsymbol{\theta}-\mu\mathbf{u}_k;\xi)}{\mu}\mathbf{u}_k$ | Eq. (6) |
| **Hessian** | $\widehat{\mathbf{H}}_{\text{diag}}(\boldsymbol{\theta}) \triangleq \frac{1}{2K}\sum_{k=1}^{K} \frac{f(\boldsymbol{\theta}_k^+;\xi)-2f(\boldsymbol{\theta};\xi)+f(\boldsymbol{\theta}_k^-;\xi)}{\mu^2}\mathbf{u}_k^{\top}\mathbf{u}_k$ | Def. 4.1 |
| **Inverse Hessian** | $1/\widehat{\mathbf{H}}_{\text{diag}}(\boldsymbol{\theta})$ | Def. 5.1 |
| **Inv-Hess-Grad Opt.** | $1/\widehat{\mathbf{H}}_{\text{diag}}(\boldsymbol{\theta}) \cdot \hat{\nabla}F(\boldsymbol{\theta})$ | Def. D.3 |

# E. Additional Experiments

## E.1. Curvature-Aware Zeroth-Order LLM Fine-Tuning

Recent advancements in memory-efficient fine-tuning for large language models (LLMs) have begun to leverage zeroth-order optimization (Malladi et al., 2023) and curvature-aware techniques (Zhao et al., 2025). However, conventional ZO Hessian approximations often suffer from high variance, which can hamper convergence during fine-tuning. ZoVH addresses this limitation by reducing variance through a provably averaged baseline and the reuse of historical query information. In this section, we apply ZoVH to curvature-aware ZO fine-tuning of LLMs, comparing its performance against MeZO (Malladi et al., 2023) and HiZOO (Zhao et al., 2025) (detailed in Appx. F.4). Specifically, we fine-tune OPT-1.3B and OPT-13B (Zhang et al., 2022) on five downstream tasks from the GLUE (Wang et al., 2018) and SuperGLUE (Wang et al., 2020) benchmarks, including SST2, COPA, SQuAD, WSC, and WIC.

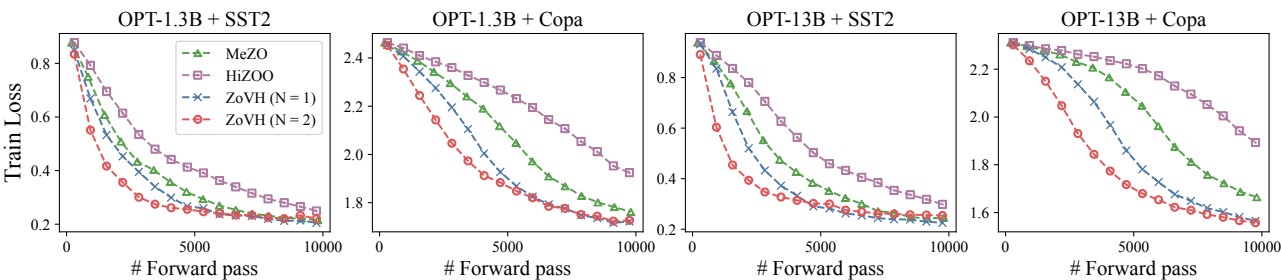

*Figure 6.* Convergence comparison of different methods for curvature-aware ZO fine-tuning on OPT-1.3B and OPT-13B.

**Results.** As shown in Fig. 6, ZoVH exhibits faster convergence than both MeZO and HiZOO across distinct tasks and model sizes. Furthermore, ZoVH with $N = 2$ (incorporating query reuse) accelerates convergence further while achieving test accuracy comparable to, or even exceeding, that of ZoVH with $N = 1$ (without query reuse). The test accuracy for all methods is summarized in Tab. 3. ZoVH secures the highest accuracy on the majority of tasks, as well as the highest average accuracy across both models. Notably, ZoVH with $N = 2$ results in lower accuracy on certain tasks compared to $N = 1$. This reflects the bias-variance tradeoff inherent in query reuse, where bias may eventually dominate the total error near convergence. In practice, this issue can be alleviated by reducing the reuse parameter $N$ as the optimization approaches convergence.

*Table 3.* Test accuracy of curvature-aware ZO methods on LLM fine-tuning tasks. All methods are limited to 10,000 forward passes. Results are averaged over 3 runs, with standard deviations in parentheses. Best results are highlighted in bold.

| Model | Method | SST2 | Copa | SQuAD | WSC | WIC | Avg |
|-------|--------|------|------|-------|-----|-----|-----|
| OPT-1.3B | MeZO | 90.41 (0.58) | 75.33 (0.58) | 75.17 (0.48) | 54.17 (0.56) | 57.26 (2.76) | 70.47 |
| | HiZOO | 85.02 (1.79) | 76.00 (0.00) | 70.25 (1.16) | 54.17 (0.56) | 57.05 (1.39) | 68.50 |
| | ZoVH ($N = 1$) | 90.29 (0.66) | **76.67** (1.53) | **75.77** (1.08) | **60.26** (6.40) | **58.73** (0.45) | **72.34** |
| | ZoVH ($N = 2$) | **90.63** (0.65) | 73.67 (1.53) | 72.86 (0.90) | 57.69 (5.35) | 56.64 (1.60) | 70.30 |
| OPT-13B | MeZO | 91.13 (0.46) | **86.00** (2.00) | 81.88 (0.40) | 51.28 (2.94) | 54.08 (0.83) | 72.87 |
| | HiZOO | 81.92 (2.62) | 80.67 (1.53) | 74.45 (0.79) | 46.47 (6.40) | 54.75 (0.18) | 67.65 |
| | ZoVH ($N = 1$) | **91.55** (0.52) | 85.67 (1.15) | **82.06** (0.85) | 59.29 (8.18) | 54.34 (0.89) | **74.58** |
| | ZoVH ($N = 2$) | 90.90 (0.52) | 83.33 (1.15) | 80.70 (1.62) | **62.18** (3.09) | **55.64** (0.41) | 74.55 |

It is worth noting that HiZOO demonstrates inferior performance compared to MeZO in both convergence speed and test accuracy. This underperformance is likely due to the Hessian smoothing factor $\alpha$ in HiZOO, which is set to a small value (e.g., $\alpha = 10^{-8}$, refer to Appx. F.4) to guarantee numerical stability when inverting the diagonal Hessian estimator (while higher $\alpha$ values leads to divergence issues in our experiments). Consequently, such a small $\alpha$ may provide negligible regularization, leading to behavior that is practically indistinguishable from MeZO. Additionally, HiZOO requires 3 function queries per step, whereas MeZO uses only 2. Since we evaluate performance under a fixed query budget (as shown in Fig. 6 and Tab. 3), HiZOO completes fewer optimization steps, further contributing to its degraded performance.

**Runtime and Memory Overhead.** We also measure wall-clock running time and peak GPU memory on the OPT-13B fine-tuning experiments to evaluate the practical overhead of the inverse Hessian-gradient product in Def. D.3. Tab. 4 reports the results.

As shown in Tab. 4, ZoVH is close to HiZOO in both running time and peak memory usage, while being only slightly slower than MeZO. Compared with HiZOO, ZoVH increases running time by about 5% and peak memory by only 0.04 GB. The wall-clock overhead remains small because forward evaluations dominate the runtime, and the additional inverse-Hessian computation only uses scalar coefficients and random directions reconstructed from their seeds. The memory overhead is also limited, since the history buffer stores only $NK$ scalar function values and their corresponding random seeds, rather than the full perturbation vectors. Its persistent memory cost is therefore $\mathcal{O}(NK)$ and does not grow with the model dimension.

*Table 4.* Running time and peak memory usage for OPT-13B fine-tuning.

| Metric | MeZO | HiZOO | ZoVH |
|---|---|---|---|
| Running Time (min) | 29.25 | 32.18 | 33.80 |
| Peak Memory (GB) | 27.14 | 27.12 | 27.16 |

### E.2. Ablation Study for Averaged Baseline and Query Reuse

We further isolate the individual contributions of the averaged baseline and query reuse in ZoVH. The ablation is conducted on four synthetic optimization benchmarks with $d = 5000$. We compare four variants that differ only in the baseline choice and whether query reuse is enabled. The anchor baseline variants use $b = F(\boldsymbol{\theta})$, while the averaged baseline variants use (8). The query reuse variants use the same history setting as the synthetic optimization experiments with $N = 4$.

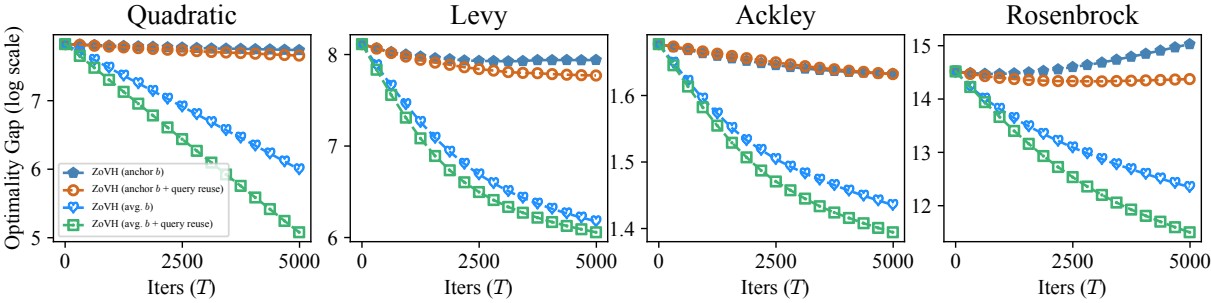

*Figure 7.* Ablation of the averaged baseline and query reuse on synthetic optimization benchmarks with $d = 5000$ and $K = 3$. Results are averaged over 5 independent runs. Lower optimality gap is better.

As shown in Fig. 7, most of the gain comes from replacing the anchor baseline with the averaged baseline, regardless of whether query reuse is used. The averaged baseline curves descend much faster and reach substantially lower optimality gaps than their anchor baseline counterparts. This supports the variance-minimization role of the averaged baseline established in Thm. 4.7. Query reuse provides an additional gain by enlarging the effective sample size without adding new function evaluations, especially in the early and middle stages of optimization. The best performance is achieved when the averaged baseline and query reuse are combined. This is consistent with the bias-variance discussion in Thm. B.1, where a moderate history length reduces variance while the iterate drift remains small enough that the additional reuse bias does not dominate.

## F. Experiments Setup

### F.1. Empirical Error Analysis of Hessian Approximation

**Baselines.** The compared estimators include the two-point (3), three-point (4) Stein Estimators, the randomized central-difference estimator (5), and our proposed ZoVH w/ and w/o the query reuse technique. The one-point Stein estimator (2) is excluded from the comparison due to its significantly higher error than other methods.

**Synthetic Functions.** All experiments are conducted in $d = 5000$ dimensions, and averaged over 500 test points Hessian errors (20 random initializations with 25 test points collected along the optimization trajectory). The analytical forms of the synthetic functions used in our experiments are as follows:

- **Quadratic Function:**

$$F(\boldsymbol{\theta}) = \frac{1}{2}\boldsymbol{\theta}^T \mathbf{A}\boldsymbol{\theta} + \boldsymbol{b}^T \boldsymbol{\theta} + c \,, \tag{123}$$

  where $\mathbf{A}$ is a symmetric positive-definite matrix, $b$ is a $d$-dimensional vector, and $c$ is a scalar constant.

- **Rosenbrock Function:**

$$F(\boldsymbol{\theta}) = \sum_{i=1}^{d-1} \left[ 100(\theta_{i+1} - \theta_i^2)^2 + (1 - \theta_i)^2 \right] \,. \tag{124}$$

- **Styblinski-Tang Function:**

$$F(\boldsymbol{\theta}) = \frac{1}{2} \sum_{i=1}^{d} \left( \theta_i^4 - 16\theta_i^2 + 50\theta_i \right) . \tag{125}$$

**Neural Network Hessian.** We further evaluate the Hessian approximation error on a small convolutional neural network (CNN), which consists of two convolutional layers, followed by two fully connected layers. In Fig. 3, we present the empirical Hessian approximation errors across the first convolutional layer ($d = 96$), the second convolutional layer ($d = 6144$), and the second fully connected layer ($d = 1280$). This model is trained on the MNIST dataset (LeCun et al., 1998) for digit classification. All experiments are averaged over 5625 test points Hessian errors (3 independent runs with 1875 test points collected along each optimization trajectory).

### F.2. Synthetic Function Optimization

**Baselines.** We compare ZoVH with several representative ZO optimization methods as baselines:

- **Vanilla ZOO (Nesterov & Spokoiny, 2017).** This is the standard stochastic finite-difference gradient estimation, whose analytical form is similar to (6), except the averaged baseline $b$ replaced with anchor baseline $b = f(\boldsymbol{\theta}; \xi)$.

- **HiZOO (Zhao et al., 2025).** This is a recent curvature-aware ZO optimization method that utilizes a randomized central-difference Hessian estimator (5) with diagonal approximation for LLMs fine-tuning. The complete comparison between HiZOO and ZoVH is detailed in Appx. D.6. We do not include ZOHA (Ye et al., 2025) as a separate baseline because HiZOO already covers this case. When the scaling factor is set to 1, HiZOO reduces to ZOHA.

- **ZoAR (Qiu et al., 2025).** This is a variance-reduced ZO optimization method that incorporates averaged baseline and query reuse techniques to improve gradient estimation.

**Hyperparameter Settings.** All experiments are conducted in $d = 10000$ dimensions, and averaged over independent 10 runs. To ensure fair comparison, the hyperparameters of all methods are chosen to achieve the best performance (summarized in Tab. 5). The analytical forms of the synthetic functions used in our experiments are as follows:

- **Quadratic Function:**

$$F(\boldsymbol{\theta}) = \frac{1}{2} \sum_{i=1}^{d} \theta_i^2 . \tag{126}$$

- **Levy Function:**

$$F(\boldsymbol{\theta}) = \sin^2(\pi w_1) + (w_d - 1)^2(1 + \sin^2(2\pi w_d)) + \sum_{i=1}^{d-1} (w_i - 1)^2(1 + 10\sin^2(\pi w_i + 1)) , \tag{127}$$

where $w_i = 1 + \frac{\theta_i - 1}{4}$.

- **Rosenbrock Function:**

$$F(\boldsymbol{\theta}) = \sum_{i=1}^{d-1} \left[ 100(\theta_{i+1} - \theta_i^2)^2 + (1 - \theta_i)^2 \right] . \tag{128}$$

- **Ackley Function:**

$$F(\boldsymbol{\theta}) = -20\exp\left(-0.2\sqrt{\frac{1}{d}\sum_{i=1}^{d}\theta_i^2}\right) - \exp\left(\frac{1}{d}\sum_{i=1}^{d}\cos(2\pi\theta_i)\right) + 20 + e . \tag{129}$$

### F.3. Black-Box Adversarial Attack

**Baselines.** The compared methods are the same as in the synthetic experiments in Sec. F.2.

*Table 5.* Hyper-parameter settings for synthetic function optimization.

| Hyper-parameter | Value |
|---|---|
| Number of queries per iteration $K$ | 3 |
| Smoothing radius $\mu$ | 0.1 |
| Hessian smooth factor $\alpha$ (For HiZOO) | $1e-8$ |
| Hessian regularization factor $\lambda$ (For ZoVH) | 0.1 |
| Learning rate $\eta$ (Quadratic) | $\{5e-6, 1e-5, 2e-5, 5e-5, 1e-4\}$ |
| Learning rate $\eta$ (Levy) | $\{5e-6, 1e-5, 2e-5, 5e-5, 1e-4\}$ |
| Learning rate $\eta$ (Ackley) | $\{1e-3, 5e-3, 1e-2, 5e-2, 1e-1\}$ |
| Learning rate $\eta$ (Rosenbrock) | $\{5e-9, 1e-8, 2e-8, 5e-8, 1e-7\}$ |

**Hyperparameter Settings.** For the black-box adversarial attack, we use the same model as in (Shu et al., 2025): a simple two-layer CNN trained on the MNIST dataset. All experiments are averaged over 10 independent runs. To ensure fair comparison, all hyperparameters are chosen to achieve the best performance of each method (summarized in Tab. 6).

*Table 6.* Hyperparameter settings for black-box adversarial attack.

| Hyperparameter | Value |
|---|---|
| Number of queries per iteration $K$ | 3 |
| Smoothing parameter $\mu$ | 0.5 |
| Learning rate $\eta$ | $\{0.01, 0.02, 0.05, 0.1\}$ |
| Hessian smooth factor $\alpha$ (For HiZOO) | $1e-8$ |
| Hessian regularization factor $\lambda$ (For ZoVH) | 0.1 |

## F.4. Curvature-Aware Zeroth-Order LLM Fine-tuning

**Baselines.** We compare ZoVH with two representative ZO optimization methods as baselines:

- **MeZO (Malladi et al., 2023).** A memory-efficient ZO fine-tuning method for LLMs that employs a central-difference gradient estimator.

- **HiZOO (Zhao et al., 2025).** A recent curvature-aware ZO optimization method that leverages a randomized central-difference Hessian estimator (5) with a diagonal approximation for LLM fine-tuning. A comprehensive comparison between HiZOO and ZoVH is provided in Appx. D.6. We do not include ZOHA (Ye et al., 2025) as a separate baseline because HiZOO already covers this case. When the scaling factor is set to 1, HiZOO reduces to ZOHA.

**Hyperparameter Settings.** All experiments are averaged over 3 independent runs. To ensure fair comparison, all hyperparameters are chosen to achieve the best performance of each method (summarized in Tab. 7).

*Table 7.* Hyperparameter settings for curvature-aware ZO LLM fine-tuning.

| Hyperparameter | Value |
|---|---|
| Number of queries per iteration $K$ | 2 (for MeZO), 3 (for HiZOO and ZoVH) |
| Smoothing parameter $\mu$ | $1e-2$ |
| Learning rate $\eta$ | $\{1e-4, 5e-5, 1e-5\}$ |
| Hessian smooth factor $\alpha$ (For HiZOO) | $1e-8$ |
| Hessian regularization factor $\lambda$ (For ZoVH) | 0.5 |

