# OpenReview forum: "Revisiting Zeroth-Order Hessian Approximation: A Single-Step Policy Optimization Lens"
_ICML.cc/2026/Conference — ICML 2026 regular_

### Official Review · Reviewer_RqXT · 2026-03-09

**Soundness:** 3
**Presentation:** 3
**Significance:** 3
**Originality:** 3
**Overall Recommendation:** 4
**Confidence:** 3

**Summary:**

This paper studies zeroth-order (ZO) Hessian estimation, a component of derivative-free optimization methods used when gradients are unavailable (e.g., black-box optimization, adversarial attacks, and LLM fine-tuning). The authors propose a new framework that interprets ZO Hessian approximation through the lens of single-step policy optimization (PO). Within this framework, the Hessian of a smoothed objective is derived using policy-gradient identities, revealing that many classical ZO Hessian estimators correspond to different choices of baseline. Building on this interpretation, the authors introduce ZoVH, a variance-reduced Hessian estimator. The paper also proposes a curvature-aware ZO optimization algorithm based on approximations of the inverse Hessian and inverse Hessian–gradient product, along with convergence guarantees. Results of experiments suggest improved estimation accuracy and faster convergence compared to several baseline methods.

**Compliance With Llm Reviewing Policy:**

Affirmed.

**Key Questions For Authors:**

1. Beyond conceptual unification, does the policy optimization perspective enable algorithmic designs that would not arise from standard ZO estimation theory?
2. Can the authors provide empirical analysis of how the smoothing parameter affects the gap between the Hessian matrices $\nabla^2 F$ and the estimated $\nabla^2 F_{\mu}$ ?
3. What are the practical memory and computational costs when applying ZoVH to large models (e.g., multi-billion parameter LLMs)?
4. How much of the improvement comes from the optimal baseline versus query reuse individually?

**Limitations:**

yes

**Strengths And Weaknesses:**

Strengths:
1. The paper provides a unification of ZO Hessian estimators through a policy optimization perspective. This viewpoint clarifies how classical Stein-based and random-direction estimators arise from baseline choices.
2. The proposed ZoVH estimator combines two sensible mechanisms: an analytically derived optimal baseline for variance minimization and reuse of historical queries to increase the effective sample size. The optimal baseline analysis is reasonably clean and supported by theoretical derivations.
3. The paper provides several theoretical guarantees: (1) unbiasedness with respect to the smoothed objective Hessian, (2) variance analysis and bias–variance trade-offs, and (3) convergence guarantees for the curvature-aware ZO algorithm.
4. The experimental section covers multiple settings.

Weakness:
1. Like many ZO estimators, the proposed Hessian estimator is unbiased only for the Gaussian-smoothed objective rather than the true Hessian. While this is unavoidable to some extent, the paper provides limited empirical analysis of how the smoothing bias affects downstream optimization, especially for large smoothing radii.
2. The paper does not clearly discuss: memory costs of storing query histories, computational overhead of constructing inverse Hessian approximations, and scaling behavior for very large models. This is particularly relevant for the LLM fine-tuning experiments.

---

> ### Author Rebuttal · Authors · 2026-03-31
>
> We thank Reviewer RqXT for the careful reading and insightful comments. We address your comments below.
>
> ----
> > W1 & Q2. the smoothing bias affects downstream optimization
>
> Thank you for this point. We address it theoretically and empirically. Theoretically, the bias between the Hessian of the smoothed objective $\nabla^2 F_\mu(\theta)$ and the true Hessian $\nabla^2 F(\theta)$ is bounded by
> $$
> ||\nabla^2 F_\mu(\theta)-\nabla^2 F(\theta)||_F=||\mathbb{E}_u[\nabla^2 F(\theta+\mu u)-\nabla^2 F(\theta)]||_F
> $$
> $$
> \overset{(a)}{\le}\mathbb{E}_u[||\nabla^2F(\theta+\mu u)-\nabla^2F(\theta)||_F]\overset{(b)}{\le}L_2\mu\mathbb{E}_u[||u||]\overset{(c)}{\le}L_2\mu\sqrt d
> $$
> where (a) follows from Jensen's inequality, (b) follows from the $L_2$-continuity of $\nabla^2 F$, and (c) comes from $\mathbb{E}_u[||u||]\leq\sqrt{\mathbb{E}_u[||u||^2]}=\sqrt d$ since $u$ is i.i.d. standard Gaussian. Therefore, the bias scales linearly with the smoothing radius $\mu$ and vanishes as $\mu \to 0$.
>
> Empirically, we also evaluated the Frobenius norm error between the Hessian of the smoothed objective and the true Hessian for different $\mu$, averaged over $5$ runs:
>
> **Frobenius norm error for different $\mu$ values (last column: slope of log error vs. log $\mu$)**
> ||μ=0.1|μ=0.01|μ=0.001|μ=0.0001|μ=0.00001|Slope|
> |-|-|-|-|-|-|-|
> |Rosenbrock|15|0.87|0.087|0.0087|0.00087|**1.04**|
> |Styblinski|1.2|0.11|0.011|0.0011|0.00011|**1.01**|
>
> The error decreases nearly linearly with $\mu$, with slope near $1$. This matches the bound above and indicates the smoothing bias is well controlled for sufficiently small $\mu$. We will include this analysis in the revision.
> > W2 & Q3. (1) memory costs and computational overhead (2) scaling behavior for very large models
>
> Thank you for raising these practical concerns. As a practical measure, we report running time and peak memory usage for the OPT-13B fine-tuning experiments:
>
> **Running Time and Peak Memory Usage for OPT-13B Fine-tuning**
> |Metric|MeZO|HiZOO|ZoVH|
> |-|-|-|-|
> |Running Time (min)|29.25|32.18|33.8|
> |Peak Memory (GB)|27.14|27.12|27.16|
>
> ZoVH has running time and peak memory comparable to HiZOO and only slightly higher than MeZO. The key reason is that the history buffer stores only scalar function values and scalar random seeds for past perturbations, rather than full perturbation vectors. Consequently, the additional memory cost is $O(NK)$ and does not scale with model dimension. The extra computation is also modest, since for large models the forward passes dominate the runtime. We will clarify this point in the revision.
> > Q1. does the policy optimization perspective enable algorithmic designs that would not arise from standard ZO estimation theory?
>
> The answer is yes. Actually, the two techniques in ZoVH, namely query reuse and the average baseline, are directly inspired by PO and are not obvious from the traditional ZOO perspective. First, query reuse is motivated by off-policy RL algorithms. Although some prior ZOO methods have similar ideas of using historical information, they do not directly reuse historical queries to construct the Hessian estimator in this way. Second, the average baseline is motivated by variance reduction techniques in policy gradient methods (e.g. GRPO) that use the average return as a baseline. This perspective naturally leads to our average-baseline estimator and the corresponding optimal baseline that minimizes variance, an insight that is much harder to identify from the conventional ZOO lens alone.
> > Q4. the improvement from the optimal baseline versus query reuse individually
>
> Thank you for this important question. Below we present an ablation study isolating the effects of the average baseline and query reuse for Levy optimization with $d=5000$ and $K=3$, averaged over $5$ runs. We compare the average baseline with the anchor baseline (i.e. $b=F(\theta)$), which is used in HiZOO and is a common choice in prior work.
>
> **Performance of Levy optimization**
> ||T=0|T=1000|T=2000|T=3000|T=4000|T=5000|
> |-|-|-|-|-|-|-|
> |ZoVH + anchor $b$|3285.3|3015.2|2838.1|2830.8|2871.2|2841.3|
> |ZoVH + anchor $b$ + reuse|3285.3|2929.6|2648.0|2533.3|2488.3|2429.8|
> |ZoVH + avg. $b$|3285.3|1659.1|986.7|697.8|564.0|491.8|
> |ZoVH + avg. $b$ + reuse|3285.3|1437.6|812.4|597.4|495.3|438.2|
>
> The results for other synthetic functions show similar trends and will be included in the revision. As shown in the table, no matter the reuse setting, the average baseline substantially outperforms the anchor baseline. This is because the average baseline is proven as the optimal constant baseline that minimizes variance. The best overall performance is obtained by combining both components. Since the anchor baseline fails to converge for $d=10000$, we only report the results for $d=5000$ here. We will include this ablation in the revision to make their individual contributions explicit.
>
> ----
> We thank the reviewer for the valuable comments and we would be happy to provide further details if helpful.

---

> > ### Author Rebuttal · Reviewer_RqXT · 2026-04-03
> >
> > My concerns are resolved. I will maintain my score.

---

> > > ### Author Response · Authors · 2026-04-04
> > >
> > > Dear Reviewer RqXT,
> > >
> > > Thank you very much for your positive response and for confirming that your concerns have been fully addressed. We are happy that our added explanations helped clarify the smoothing bias analysis induced by the Gaussian smoothing factor $\lambda$, the algorithmic insights from the policy optimization lens, the practical memory and computational costs of ZoVH for large models, and the individual contributions of the average baseline and query reuse. We will carefully incorporate these clarifications into the revised version so that the final paper presents both the theoretical and practical aspects of ZoVH more clearly.
> > >
> > > Sincerely, Authors

---

### Official Review · Reviewer_MFwX · 2026-03-10

**Soundness:** 3
**Presentation:** 3
**Significance:** 2
**Originality:** 2
**Overall Recommendation:** 4
**Confidence:** 3

**Summary:**

The authors propose a unified framework that reinterprets Zeroth-Order (ZO) Hessian approximation through the lens of single-step Policy Optimization (PO). This perspective establishes a theoretical equivalence between general ZO Hessian estimators and the Hessian of a smoothed PO objective, effectively unifying distinct classical randomized estimators as specific instances of baseline selection. Building on this foundation, the paper introduces **ZoVH**, a comprehensive suite of variance-reduced estimators for the full Hessian matrix, its regularized inverse, and the bias-corrected inverse Hessian-gradient product. ZoVH leverages two key techniques to enable practical curvature-aware optimization: (1) a unique **optimal baseline** (specifically, the "averaged baseline") derived to provably minimize variance, and (2) a **query reuse strategy** that incorporates historical function queries to enhance sample efficiency without inflating costs. Rigorous theoretical analysis confirms the unbiasedness of the estimator, validates the optimality of the baseline, and establishes convergence guarantees for the resulting algorithm. Extensive empirical results validate these findings, demonstrating that ZoVH achieves superior estimation accuracy and convergence performance in real-world applications.

**Compliance With Llm Reviewing Policy:**

Affirmed.

**Final Justification:**

This is a technically solid paper that advances zero gradient optimization, with a contribution that others are likely to build on, but with a relatively narrow application that limit its impact. The rebuttal addressed my main concerns on the proof under $K>1$ setting.

**Key Questions For Authors:**

See weaknesses.

**Limitations:**

See weaknesses.

**Strengths And Weaknesses:**

Strength:
- Novel PO/RL lens: Recasting ZO Hessian estimation as the Hessian of a smoothed single-step PO objective is elegant and unifies multiple classical estimators via baseline choice, enabling principled variance reduction (optimal averaged baseline).
- High query efficiency: The query reuse mechanism effectively increases sample size using historical evaluations, reducing variance without additional function calls—critical in black-box settings.
- Practical impact: The paper targets high-dimensional, real-world–motivated applications (e.g., LLM fine-tuning and black-box adversarial attacks) where curvature information can materially improve optimization.
- Thorough analysis: Provides a relatively complete package—unbiasedness/variance optimality, error bounds, and convergence guarantees—plus empirical validation.

Weakness:
- The theoretical analysis does not fully cover the proposed algorithm (where $N > 1$). Specifically, how does the "off-policy" distribution shift (the difference between the current parameter $\theta_t$ and historical parameters $\theta_{t - 1}$​) affect the bias? While they mention Importance Sampling (IS), does the effective sample size degenerate if the parameters move too fast?
- The authors propose a "bias-corrected shared product" to avoid using 2K queries.The bias-corrected estimator in Eq. (16) involves a leave-one-out strategy.  Does this denominator $\lambda ^2 (K - 1) + ...$ ever approach zero or cause numerical instability if the regularization $\lambda$ is very small? Does the added computational complexity of calculating this term outweigh the cost of just sampling $K$ fresh points?
- While the paper strongly highlights the method's query efficiency, it does not fully detail the computational wall-clock overhead introduced by Algorithm 1. Could you provide a comparison of the wall-clock time per iteration between ZoVH, MeZO, and HiZOO, particularly for the large-scale LLM fine-tuning experiments?

---

> ### Author Rebuttal · Authors · 2026-03-31
>
> We thank Reviewer MFwX for the careful reading and valuable comments. We address your concerns below.
>
> ----
> > W1. (1) The theoretical analysis for $N>1$
>
> Thank you for highlighting this issue. In the $N>1$ query reuse setting, historical samples are generated around previous iterates rather than the current parameter, which creates a bias-variance trade-off. The larger sample size can reduce the variance (thus lower total error), but the extra bias increases the error. For a history buffer of size $N\times K$, we can extend Thm. 4.8 to characterize this trade-off
> $$
> \mathbb{E}[||\hat H(\theta)-\nabla^2F(\theta)||_F^2]\leq\frac{NKd(d+2)\lambda^2V}{\lambda^2(NK-1)^2\mu^4-L_0^2\eta^2 (N-1)^2 K/3}+L_2^2\mu^2d
> $$
> When $N=1$, this recovers the original result. Relative to Thm. 4.8, query reuse enlarges the sample size from $K$ to $NK$, which reduces variance, but it also introduces the additional term $L_0^2 \eta^2 (N-1)^2 K/3$, capturing the bias induced by off-distribution reuse. Since the learning rate $\eta$ is typically small, the bias term should be small for moderate $N$, thus query reuse can still reduce the overall estimation error.
>
> This extension also clarifies the convergence guarantee in Thm. 5.3. The bound there depends on the estimation error of the inverse-Hessian-gradient product (Eq. (89) in the appendix), which is in turn controlled by the variance of the Hessian estimator (Thm. 4.8 and the above extension for $N>1$). Consequently, a more accurate Hessian estimator improves the constants in the convergence bound, and can yield substantial convergence speedups. We will include this $N>1$ theory in the revision to better explain the theoretical role of query reuse.
> > W1. (2) how the effective sample size degenerate if the parameters move too fast
>
> In our framework, query reuse is exactly an off-policy importance sampling (IS) mechanism. Thm. 3.7 introduces the IS correction, and Sec. 4.1 interprets historical queries as samples from a proposal distribution that differs from the current target distribution. If the parameters change too rapidly, or if the buffer size $N$ is too large, the proposal distribution may drift far from the target. The resulting IS weights then become highly concentrated, which reduces the effective sample size and can destabilize the estimator. This also matches the bias-variance trade-off in the above response to W1, where the bias term grows with $N$ and $\eta$. Therefore, stable query reuse requires a moderate buffer size $N$ and learning rate $\eta$. We will clarify this point in the revision to make the associated bias-variance trade-off more explicit.
> > W2. (1) the denominator $\lambda ^2 (K-1)+...$ approach zero or cause numerical instability
>
> Thank you for raising this concern. Yes, if the regularization parameter $\lambda$ is chosen too small, the denominator in Eq. (16) can become small and lead to numerical instability. In practice, we avoid this issue by using a relatively large value (e.g. $\lambda=0.1$ in the synthetic function optimization and black-box adversarial-attack experiments). However, we would like to clarify that this is a standard issue in second-order optimization rather than a limitation unique to our method. For instance, the textbook "Numerical Optimization" by Nocedal and Wright emphasize that adding a multiple of the identity is important for ensuring sufficient positive definiteness of the Hessian (Page 51, Sec 3.4, "ADDING A MULTIPLE OF THE IDENTITY" paragraph), and the paper "Optimization Methods for Large-Scale Machine Learning" (Bottou, 2016, SIAM) similarly notes that Hessian singularity is typically addressed by adding a positive multiple of the identity matrix. We will clarify this practical requirement in the revision.
> > W2 (2) the added computational complexity of calculating this term
> > W3. the computational wall-clock overhead
>
> Thank you for raising these practical concerns. As a practical measure of this overhead, we report the running time and peak memory usage for the OPT-13B fine-tuning experiments:
>
> **Running Time and Peak Memory Usage for OPT-13B Fine-tuning**
>
> |Metric|MeZO|HiZOO|ZoVH|
> |-|-|-|-|
> |Running Time (min)|29.25|32.18|33.8|
> |Peak Memory (GB)|27.14|27.12|27.16|
>
> As shown in the table, the running time and peak memory usage of ZoVH are both comparable to HiZOO and only slightly higher than MeZO. The key reason is that the history buffer stores only scalar function values and scalar random seeds for past perturbations, rather than full perturbation vectors. Besides, the extra computation is also modest, since for large models the forward passes dominate the runtime. We will clarify this point in the revision.
>
> ----
> We hope these clarifications and additional empirical results address your concerns. We would be happy to provide further details if helpful.

---

> > ### Author Rebuttal · Reviewer_MFwX · 2026-04-03
> >
> > Thank you for your response. Your answers and additional experiments have cleared my major concerns. After consideration, I will maintain my score.

---

> > > ### Author Response · Authors · 2026-04-04
> > >
> > > Dear Reviewer MFwX,
> > >
> > > We are very happy that our response has fully resolved your concerns! We are particularly glad that our additional explanations helped clarify the theoretical role of query reuse in the $N>1$ setting, the issues around the importance sampling perspective, and the computational overhead of our ZoVH estimator. We will carefully integrate all of these clarifications into the revised paper to make the practical and theoretical aspects of ZoVH clearer and more complete.
> > >
> > > Sincerely, Authors

---

### Official Review · Reviewer_BKg6 · 2026-03-13

**Soundness:** 3
**Presentation:** 3
**Significance:** 3
**Originality:** 3
**Overall Recommendation:** 4
**Confidence:** 4

**Summary:**

I found the paper interesting and reasonably well executed. The main idea is to reinterpret zeroth-order Hessian estimation through a single-step policy optimization lens, then use that viewpoint to derive a variance-reduced estimator, ZoVH. Concretely, the paper shows that different classical randomized Hessian estimators can be recovered by different baseline choices in the PO-style formulation (Theorems 3.2/3.3, Corollaries 3.4 and 3.6). On top of that, the method adds two practical ingredients: an averaged baseline motivated by variance minimization (Theorem 4.7) and a query reuse mechanism that reuses historical function evaluations. The paper also extends the estimator to a regularized inverse Hessian approximation and a shared-sampling inverse-Hessian-gradient product, then plugs these into a curvature-aware ZO optimizer.

**Compliance With Llm Reviewing Policy:**

Affirmed.

**Final Justification:**

The rebuttal has addressed my main concerns, and I prefer to maintain my score.

**Key Questions For Authors:**

1. Can you provide a theorem for the `N>1` query-reuse setting, even if it is a coarse bound that depends on staleness or parameter drift?
2. How accurate is the approximation in (15) relative to the exact inverse in (14) for the actual `K` values used in experiments?
3. Why is ZOHA not included as a baseline?

**Limitations:**

The main limitation of the paper is that its strongest practical ingredient, **query reuse**, is not matched by equally strong theory. The analysis in Section 4.2 is restricted to the `N=1` setting, while the best empirical results rely on `N>1`. As a result, the paper does not yet give a satisfying explanation of how reuse interacts with parameter drift, stale samples, or approximation error.

A second limitation is that the **optimization theory is not significantly stronger than standard ZO results**. The convergence bound in Theorem 5.3 has the usual `O(\varepsilon^{-4})` form, so the theory does not explain why the method is dramatically faster in practice. This makes the paper's empirical speedup story stronger than its theoretical one.

**Strengths And Weaknesses:**

1. **The unifying viewpoint is the strongest part of the paper.**
   I liked the way the PO lens organizes the estimator family. Recovering one-point, two-point, three-point Stein estimators and the randomized central-difference estimator as different baseline/sampling choices makes the design space much easier to understand. This part felt novel and conceptually clean rather than just a re-packaging.


2. **The Hessian estimation experiments are convincing.**
   The cleanest evidence in the paper is in Fig. 2 and Fig. 3. ZoVH is consistently better than the classical estimators, and the gains are not marginal. The reduction in spread/error bars when query reuse is used is also visually consistent with the variance-reduction story.

#### Weaknesses


1. **The convergence guarantee is standard and does not explain the empirical speedups.**
   Theorem 5.3 gives an `O(\varepsilon^{-4})` iteration complexity, which is a standard zeroth-order rate. That is not necessarily a problem by itself, but then the paper should be more honest that the benefit seems to be in constants and practical estimator quality, not in a better asymptotic rate. As written, the theory does not really explain the reported ~22x speedup over Vanilla ZOO in Fig. 1.

2. **The inverse-Hessian analysis leans on an orthogonality assumption that is not validated enough.**
   The approximation in (15) becomes exact only when the random directions are pairwise orthogonal, and Theorem B.3 / Theorem B.4 also rely on orthogonality. The paper argues this is approximately true in high dimension for Gaussian directions, but in practice the method uses very small `K` (often `K=2` or `K=3`). I would have liked at least one empirical check of how much error is introduced by replacing the exact Woodbury inverse in (14) with the approximation in (15).

3. **A directly relevant comparison is missing.**
   The paper cites ZOHA (Ye et al., 2025), which is an obvious baseline for a Hessian-aware ZO paper, but does not compare against it. I think the authors should either include that comparison or clearly explain why it is excluded.

---

> ### Author Rebuttal · Authors · 2026-03-31
>
> We thank Reviewer BKg6 for the careful reading and thoughtful comments. We address your concerns below.
>
> ----
> > W1 & L2. The convergence guarantee is standard and does not explain the empirical speedups
>
> Thank you for this important point. We agree that Thm. 5.3 has the standard $O(\varepsilon^{-4})$ ZO rate, and our algorithm improves constants, not the asymptotic order. In particular, the convergence bound depends on the error of the inverse-Hessian-gradient estimate (Eq. (89)), which is controlled by the variance of the Hessian estimator (Thm. 4.8). The optimal baseline in Thm. 4.7 reduces this variance, and query reuse further increases the effective sample size when $N>1$ (further explained in our response to Q1 & L1). These effects improve the constants in the convergence bound and can yield substantial speedups even though the asymptotic rate is unchanged. We will revise the paper to make this distinction explicit and better align the theory with the empirical gains.
> > W2 & Q2. The inverse-Hessian analysis leans on an orthogonality assumption that is not validated enough.
>
> Thank you for raising this concern. The theoretical bound for the approximation error between the exact inverse in Eq. (14) and the approximation in Eq. (15) scales as
> $$
> ||(\hat H(\theta)+\lambda I_{d})^{-1}-\tilde H^{-1}(\theta)||_F=O_p(K/\sqrt d).
> $$
> The main proof idea is that for $G:=U^\top U$, the diagonal part $\Delta:=\operatorname{diag}(||u_1||^2,\dots,||u_K||^2)$ is of order $\Theta_p(d)$, while the off-diagonal part $E:=G-\Delta$ has size only $O_p(K\sqrt d)$ because $u_i^\top u_j$ has variance $d$ for $i\neq j$. If we write $M=D^{-1}+\lambda^{-1}G$ and $\tilde M=D^{-1}+\lambda^{-1}\Delta$, the resolvent identity gives $||M^{-1}-\tilde M^{-1}||_F=||M^{-1}E\tilde M^{-1}||_F\leq||M^{-1}||_2 ||E||_F ||\tilde M^{-1}||_2=O_p(Kd^{-3/2})$, yielding the final bound above.
>
> Thus, for the small fixed $K$ used in our experiments, the error decays at rate $d^{-1/2}$ and becomes negligible in high dimensions.
>
> Empirically, we measured the Frobenius norm error between the exact and approximate inverse Hessians with $K=3$, averaged over $50$ random seeds:
>
> **Frobenius Norm Error (The slope in the last column is the log error vs. log d)**
>
> ||d=1000|d=5000|d=10000|d=20000|slope|
> |-|-|-|-|-|-|
> |Quadratic|0.068±0.03|0.033±0.01|0.023±0.01|0.015±0.006|-0.53|
> |Rosenbrock|0.065±0.03|0.031±0.01|0.022±0.009|0.016±0.007|-0.51|
> |Levy|0.078±0.03|0.033±0.01|0.022±0.009|0.016±0.007|-0.52|
> |Ackley|0.073±0.03|0.033±0.01|0.021±0.008|0.017±0.007|-0.54|
>
> The Frobenius error decays with slope about -0.5, matching the $d^{-1/2}$ prediction. These results support the approximation used in practice. Since LLM layers typically have millions of parameters, the approximation error should be even smaller in our target applications.
> > W3 & Q3. A directly relevant comparison is missing.
>
> Thank you for pointing this out. We did not include ZOHA separately because it is effectively a special case of HiZOO. Specifically, HiZOO applies an exponential moving average to the diagonal inverse-Hessian estimator (Alg. 1 in Zhao et al., 2025). When the scaling factor is set to $1$, the HiZOO reduces to ZOHA (Alg. 1 and Eq. (17) in Ye et al., 2025). Since HiZOO is the more general variant and is evaluated on LLMs, we used it as the primary baseline. We will clarify this relationship explicitly in the revision.
> > Q1 & L1. Can you provide a theorem for the N>1 query-reuse setting?
>
> Thank you for raising this important point. For $N>1$, query reuse introduces a bias-variance trade-off, where the larger sample size can reduce the variance (thus lower total error), but the extra bias increases the error. A coarse extension of Thm. 4.8 for a history buffer of size $N \times K$ is
> $$
> \mathbb{E}[||\hat H(\theta)-\nabla^2F(\theta)||_F^2]\leq\frac{NKd(d+2)\lambda^2V}{\lambda^2(NK-1)^2\mu^4-L_0^2\eta^2 (N-1)^2K/3}+L_2^2\mu^2d
> $$
> Relative to Thm. 4.8, query reuse changes the sample size from $K$ to $NK$, lowering variance, but adds the bias term $L_0^2\eta^2(N-1)^2K/3$ from off-distribution reuse. When $N=1$, this reduces to the original result. Since the learning rate $\eta$ is typically small, the bias term should be small for moderate $N$, so query reuse can still reduce the overall estimation error.
>
> This also helps explain the optimization behavior. The bound in Thm. 5.3 depends on the estimation error of the inverse-Hessian-gradient product (Eq. (89) in the appendix), which is in turn controlled by the variance of the Hessian estimator (Thm. 4.8 and the above extension for $N>1$). Hence, improving the Hessian estimate improves the constants in the convergence bound, even though the asymptotic order remains unchanged. We will add this discussion to the revision to clarify the theoretical role of query reuse.
>
> ----
> We hope this clarification addresses your concerns, and we are happy to provide more clarifications if needed.

---

> > ### Author Rebuttal · Reviewer_BKg6 · 2026-04-03
> >
> > Thank you for addressing my comments. I will maintain my score.

---

> > > ### Author Response · Authors · 2026-04-04
> > >
> > > Dear Reviewer BKg6,
> > >
> > > Thank you so much for your positive response after our rebuttal! We are so happy to hear that your concerns have been fully addressed.
> > > We are especially encouraged that the points you raised regarding the practical meaning of the ZoVH convergence result, the orthogonality assumption behind the inverse Hessian approximation, and the $N > 1$ theory for query reuse technique have now been clarified.
> > > We will carefully incorporate all of these clarifications into the revised paper so that the final version better reflects both the practical and theoretical contributions of our method.
> > >
> > > Sincerely, Authors

---

### Official Review · Reviewer_yPjR · 2026-03-21

**Soundness:** 4
**Presentation:** 3
**Significance:** 3
**Originality:** 3
**Overall Recommendation:** 4
**Confidence:** 4

**Summary:**

The paper studies zeroth-order Hessian estimators for random functions of high-dimensional real vectors. (Zeroth-order refers to the fact that the estimator only relies on function evaluations without access to gradients or other information).

More precisely, given a (deterministic) function $f(\bf{\theta}; \xi)$ the ability to draw samples from the probability distribution over $\xi$, and the ability to evaluate $f$, the paper is interested in estimating the Hessian of
$F(\bf{\theta}) := \mathbb{E}\left[ f(\bf{\theta}; \xi) \right]$. Estimation relies on function evaluations of the form
$f(\bf{\theta} + \mu \bf{u}; \xi)$ for vectors $\bf{u}$ chosen from a distribution, often a standard Gaussian distribution, and a scaling parameter $\mu$.

The paper points out a connection to policy optimization --- The paper considers a policy, parameterized by $\bf{\theta}$, that takes action $\bf{x} = \bf{\theta} + \mu \bf{u}$, where $\bf{u}$ has an arbitrary distribution.  It also imagines that $f(\bf{x}; \xi)$ is the random payoff of the policy.  It then observes that the expected payoff of this policy is a smoothed version of $F$ and that the Hessian of the smoothed version has a relationship to the Hessian of $F$ (Theorem 3.1).

When $\bf{u}$ is Gaussian, this relationship becomes more explicit
and can be written via a simple formula (Theorem 3.2) with an arbitrary constant $b$.

If $b$ is chosen in a particular way (that requires knowledge of $f$) called the "averaged baseline", a term is eliminated and the formula simplifies (Theorem 3.3).

Setting $b$ to specific values recovers various classical estimators (Corollary 3.4). Adding in antithetic sampling of $\bf{u}$ recovers another previously-proposed estimator.

A similar estimator can also be created by letting $\bf{u}$ be chosen from a non-Gaussian distribution and then using importance sampling.  (Theorem 3.7)

The paper then proposes a new estimator (9) that estimates the averaged baseline $b$ from Theorem 3.3, shows that it is unbiased (Theorem 4.6). The paper also shows that the averaged baseline minimizes variance of the Hessian estimator across all baselines (Theorem 4.7).
It also proposes query re-use in an iterative optimization scheme, where function evaluations from previous iterates are re-used.

The paper provides additional results on inverse Hessian estimation and estimation of the product of the inverse Hessian and gradient (useful in optimization).

It then evaluates its proposed estimators against baselines in terms of optimality gap when the estimators are used within an optimization algorithm, and in terms of Frobenius norm of the Hessian estimation error.

**Compliance With Llm Reviewing Policy:**

Affirmed.

**Final Justification:**

The paper is a good, technically solid paper.  I feel that it deserves to be accepted.  The paper would be stronger if it had a stronger empirical section.  The rebuttal gave me confidence in my initial assessment that the paper is above the bar and also gave me confidence that the authors would improve some aspects of the presentation that I raised my review.

**Key Questions For Authors:**

1. Is there some insight or observation that comes from this lens that is somehow harder to see without the lens?  To me, the math and intuition provided is enabled via the fact that the function $F_\mu$ is a smoothed version of $F$, and the fact that this can be interpreted as the payoff of a policy doesn't help my understanding.  To me, Theorem 3.2 and 3.3 can be seen more directly via direct computation, without a connection to policy optimization.  To me, it would be better to mention the connection to policy optimization as a remark rather than including it in the title of the paper.

2. I really liked that (9) is an unbiased estimator, despite using an estimator of the baseline $b$ instead of the baseline itself.  The proof has a nice trick in it involving treating correlated and independent terms separately that wasn't obvious to me.  $\hat{b}$ is not independent of $\bf{u}_k \bf{u}_k^T$, but because of this trick, as I understand it, you still can do the computation as if

$$
E\left[\sum_{k=1}^K \hat{b} \ \bf{u}_k \bf{u}_k^T\right]
$$
is equal to

$$
E\left[\hat{b}\right] E\left[\sum_{k=1}^K \bf{u}_k \bf{u}_k^T\right]
$$

I feel that this should be highlighted in the proof sketch.

3. I believe there is a typo in the proof of Theorem 4.6 in the Appendix.  At the bottom, where it says that "(a) follows from the independence of ...", this explanatory text is actually referring to equation (49).

4. In query re-use, when you re-use samples from previous iterates during optimization, these iterates are not distributed according to the assumed Gaussian distribution centered at $\theta$, correct?  How does this affect things?  If true, it seems like an important limitation.

5. I sense that there is a connection to control variates.  I wonder if the choice of b that minimizes variance (Theorem 4.7) can come from the same analysis used to derive control variate estimators.

6. The literature reviewed is mostly either within the last few years, or quite old (the 1970s).  There is a field called "simulation" (their main conference is the Winter Simulation Conference) and they publish often on estimators for gradients, Hessians, quantiles, etc based on the output of stochastic simulators. A nice textbook in this area is Asmussen & Glynn 2007. I'd like to see a stronger connection to this literature. What is the closest thing from that literature to this?

**Limitations:**

yes

**Strengths And Weaknesses:**

Strengths:
- I felt that Theorem 4.6 was non-trivial --- at first, I thought it was wrong and only after checking the proof do I understand how it works
- I liked Theorem 4.7 and the fact that the estimation framework generalized a number of results in the literature, providing a common basis to understand them
- The numerical experiments suggest that the estimator is pretty effective from a practical perspective
- The estimator is simple (a good thing)

Weaknesses:
- It took me a long time to separate the things that seem straightforward from the non-trivial things in the paper.  I feel that the results named as theorems in Section 3 should be renamed propositions or remarks so that more attention can be placed on the main results in section 4 and 5.  Along these lines, I found the connection to polity optimization confusing and distracting. The paper would be better if it were made more skimmable by someone with basic skills in stochastic analysis.
- Because I don't do research in this area (see below), it is possible that there is a previous result in the literature that scoops this (see comment about simulation literature) without my knowledge.  I am adhering to the non-use of LLMs in writing this review, but if I were allowed to use an LLM I would put the paper into an LLM and ask for the closest results in the stochastic simulation literature (see "Key Questions for the Authors").

For context, I don't do research in this area.  I do sometimes teach a course that covers topics in simulation, including designing gradient estimators and understanding their bias and variance, and I use various stochastic gradient and Hessian estimators for optimization in research.  So I can follow the arguments and know some facts and proof techniques but it is hard for me to evaluate novelty beyond what is written in the paper.

UPDATE AFTER REBUTTAL
I appreciate the authors' careful and helpful rebuttal. I now understand the connection to policy optimization and agree this should be a part of the paper's story.  The discussion of bias from query reuse was interesting and helpful.  And I look forward to the updated version of the paper that includes refinements addressing my comments, including the link to the simulation literature.

---

> ### Author Rebuttal · Authors · 2026-03-31
>
> We thank Reviewer yPjR for your appreciation of our work and insightful comments. We would like to provide the following response to your concerns.
>
> ----
> > W1. theorems in Sec. 3 should be renamed propositions or remarks
>
> Thank you for the suggestion. We will rename the results in Sec. 3 as propositions to better distinguish them from the main results in Sec. 4, 5. We will also improve the readability for readers with a basic background in stochastic analysis, and clarify the ZOO-PO connection in the revision.
> > W2. stochastic simulation literature
>
> We address this in our response to Q6 below.
> > Q1. (1) insight or observation that comes from PO lens (2) the payoff of a policy (3) without a connection to policy optimization (4) mention the connection to policy optimization as a remark.
>
> (1) Actually, the two techiniques in ZoVH, namely query reuse and the average baseline, are directly inspired by PO and are not obvious from the traditional ZOO perspective. First, query reuse is motivated by off-policy RL algorithms. Although some prior ZOO methods have similar ideas of using historical information, they do not directly reuse historical queries to construct the Hessian estimator in this way. Second, the average baseline is motivated by policy gradient algorithms (e.g. GRPO) that use the average return as a baseline. We believe that these insights are much harder to identify from the conventional ZOO lens alone.
>
> (2-3) We agree that $F_\mu$, and Thm. 3.2, 3.3 can be derived by direct mathematical computation. However, our goal in introducing the PO lens is not merely to provide an alternative derivation of these results, but to offer a unified viewpoint that connects ZOO and PO. This perspective makes it natural to import ideas from PO to ZOO and directly motivates the design of ZoVH which includes two key techniques we introduced above. In this sense, the value of the PO lens is more in revealing a principled path for developing new ZOO algorithms.
>
> (4) For below reasons, we view the connection to PO as a core part of the paper, and we will revise the presentation to make this role clearer.
> > Q2. highlighted in the proof sketch.
>
> Thank you for your appreciation. we will make this step explicit in the proof sketch of Thm. 4.7 in the revision, and explain why the estimator remains unbiased despite the dependence between $\hat{b}$ and $u_ku_k^\top$.
> > Q3. a typo in the proof of Thm 4.6
>
> The statement "(a) follows from the independence of ..." refers to step (a), namely the second equality in Eq. (49), not the entire Eq. (49). We will revise the explanation around Eq. (49) to make this correspondence clearer.
> > Q4. not distributed according to the assumed Gaussian distribution centered at $\theta$
>
> Thank you for raising this concern. Under query reuse, reused samples are indeed no longer centered at the current iterate $\theta$, which introduces additional bias. This creates a bias-variance trade-off where the larger sample size can reduce the variance (thus lower total error), but the extra bias increases the error. For a history buffer of size $N\times K$, we can extend Thm. 4.8 to characterize this trade-off
> $$
> \mathbb{E}[||\hat H(\theta)-\nabla^2F(\theta)||_F^2]\leq\frac{NKd(d+2)\lambda^2V}{\lambda^2(NK-1)^2\mu^4-L_0^2\eta^2 (N-1)^2 K/3}+L_2^2\mu^2d
> $$
> Relative to Thm. 4.8, query reuse increases the sample size from $K$ to $NK$, reducing variance, but also adds the bias term $L_0^2\eta^2(N-1)^2 K/3$ from off-distribution reuse. When $N=1$, this recovers the original bound. Since the learning rate $\eta$ is typically small, the bias term should be small for moderate $N$, thus query reuse can still reduce the overall estimation error.
> > Q5. connection to control variates
>
> Yes, there is a close connection to control variates. In the classical setting of control variates, one replaces an unbiased estimator $A$ by $A+bC$ with $\mathbb{E}[C]=0$ and chooses $b$ to minimize its variance. In our case, $A(u)=F(\theta+\mu u)B(u)$ is an unbiased estimator of $\nabla^2 F_\mu(\theta)$ and $B(u)=\frac{1}{\mu^2}(uu^\top-I)$ has mean zero, so $\hat H(\theta)=A(u)-bB(u)$ remains unbiased for any constant baseline $b$. Thus, Thm. 4.7 follows from the similar variance minimization principle.
> > W2 & Q6. stronger connection to Asmussen & Glynn 2007
>
> We agree that the connection to the stochastic simulation literature should be made clearer. For "Asmussen & Glynn 2007", the closest connections are variance reduction, especially control variates that relates directly to our optimal baseline in Thm. 4.7 and the importance sampling that relates Sec. 3.3. Besides, our work also connects to derivative estimation, where our ZO gradient and Hessian estimators can be viewed as randomized finite-difference estimators for high-dimensional settings rather than classical coordinate-wise finite differences in this book.
>
> ----
>
> We hope this clarification addresses your concerns, and we are happy to provide more clarifications if needed.

---

### Decision · Program_Chairs · 2026-04-30

**Decision:**

Accept (regular)

**Comment:**

This paper introduces a unified framework that reinterprets Zeroth-Order (ZO) Hessian approximation through the lens of single-step Policy Optimization (PO), proving that many classical randomized estimators are simply specific baseline choices within a smoothed PO objective. Building on this theoretical equivalence, the authors propose ZoVH, a comprehensive suite of variance-reduced estimators for the full Hessian matrix, its regularized inverse, and the inverse Hessian-gradient product. ZoVH achieves superior sample efficiency and high estimation accuracy by combining a variance-variance"averaged baseline" with a query reuse strategy that recycles historical function evaluations without significantly inflating memory costs. All reviewers find the paper technical sound and interesting. The topic of estimating Hessian is a central topic in various applications. Thus I lean towards acceptance.

In the revised version, the authors should integrate the extended theoretical bounds for the $N>1$ query-reuse setting to transparently discuss the bias-variance trade-off. Furthermore, they should include the newly provided empirical validation from the rebuttal.